# Macrophages excite muscle spindles with glutamate to bolster locomotion

Yuyang Yan[1,2], Nuria Antolin[3], Luming Zhou[1], Luyang Xu[4], Irene Lisa Vargas[3], Carlos Daniel Gomez[3], Guiping Kong[1], Ilaria Palmisano[1], Yi Yang[1], Jessica Chadwick[1], Franziska Müller[1], Anthony M. J. Bull[4], Cristina Lo Celso[2], Guido Primiano[5,6], Serenella Servidei[5,6], Jean François Perrier[3], Carmelo Bellardita[3,7 ✉] & Simone Di Giovanni[1,7 ✉]

The stretch reflex is a fundamental component of the motor system that orchestrates the coordinated muscle contractions underlying movement. At the heart of this process lie the muscle spindles (MS), specialized receptors finely attuned to fluctuations in tension within intrafusal muscle fibres. The tension variation in the MS triggers a series of neuronal events including an initial depolarization of sensory type Ia afferents that subsequently causes the activation of motoneurons within the spinal cord[1,2]. This neuronal cascade culminates in the execution of muscle contraction, underscoring a presumed closed-loop mechanism between the musculoskeletal and nervous systems. By contrast, here we report the discovery of a new population of macrophages with exclusive molecular and functional signatures within the MS that express the machinery for synthesizing and releasing glutamate. Using mouse intersectional genetics with optogenetics and electrophysiology, we show that activation of MS macrophages (MSMP) drives proprioceptive sensory neuron firing on a millisecond timescale. MSMP activate spinal circuits, motor neurons and muscles by means of a glutamate-dependent mechanism that excites the MS. Furthermore, MSMP respond to neural and muscle activation by increasing the expression of glutaminase, enabling them to convert the uptaken glutamine released by myocytes during muscle contraction into glutamate. Selective silencing or depletion of MSMP in hindlimb muscles disrupted the modulation of the stretch reflex for force generation and sensory feedback correction, impairing locomotor strategies in mice. Our results have identified a new cellular component, the MSMP, that directly regulates neural activity and muscle contraction. The glutamate-mediated signalling of MSMP and their dynamic response to sensory cues introduce a new dimension to our understanding of sensation and motor action, potentially offering innovative therapeutic approaches in conditions that affect sensorimotor function.

The stretch reflex is the fastest motor response of the nervous system to mechanical stimuli, identified at the end of the nineteenth century by a series of studies from Sherrington[2–8]. The response is mediated by a peripheral receptor, the so-called muscle spindle (MS), located in parallel with extrafusal muscle fibres and delimited by a connective tissue capsule[1,9]. The stretch of the receptor activates the proprioceptive sensory Ia fibres that convey information about the degree and the final length of the homologous muscle. The Ia fibres enter the spinal cord where they synapse directly onto the alpha motor neurons to evoke muscle contraction. Disruption of this neuromuscular circuit leads to sensorimotor impairments in proprioception, irregular muscle tone affecting motor control, imbalanced posture and stepping that limit motor rehabilitation[10,11].

This underscores the role of MS in sensorimotor function. Despite decades of study, key aspects of this crucial neural element for motor control remain unclear. In particular, the molecular mechanisms underpinning their function as well as the nature of the communication between the components of the circuitry remain undetermined.

Therefore, we aimed to investigate the molecular signatures and mechanisms specifically associated with MS physiology. Contrary to the prevailing model, which emphasizes exclusive skeletal muscle–neuronal interaction in MS activity, our investigation revealed an unanticipated role for tissue-resident macrophages within the neuromuscular circuitry, offering an unexpected dimension to the neural dynamics governing sensorimotor function.

[1]Faculty of Medicine, Department of Brain Sciences, Imperial College London, London, UK. [2]Faculty of Natural Sciences, Department of Life Sciences, Imperial College London, London, UK. [3]Department of Neuroscience, University of Copenhagen, Copenhagen, Denmark. [4]Faculty of Engineering, Department of Bioengineering, Imperial College London, London, UK. [5]Dipartimento di Neuroscienze, Organi di Senso e Torace, Fondazione Policlinico Universitario Agostino Gemelli IRCCS, Rome, Italy. [6]Dipartimento di Neuroscienze, Università Cattolica del Sacro Cuore, Rome, Italy. [7]These authors contributed equally: Carmelo Bellardita, Simone Di Giovanni. ✉e-mail: cbellardita@sund.ku.dk; s.di-giovanni@imperial.ac.uk

## MS molecular signatures reveal MSMP

Initially, we investigated the molecular signatures associated with MS, where intrafusal myofibers reside, by performing RNA sequencing (RNA-seq) after laser microdissection of the MS microenvironment versus non-MS (NMS) extrafusal muscle fibres (Extended Data Fig. 1a,b). Principal component analysis (PCA) and differential gene expression revealed a very clear separation of the MS versus the extrafusal microenvironment (Extended Data Fig. 1c–e and Supplementary Table 1), including the exclusive expression of well-established MS markers within the MS microenvironment (Extended Data Fig. 1f and Supplementary Table 1)[9,12,13]. Gene ontology (GO) analysis of differential gene expression between MS and NMS areas showed significant enrichment for axon guidance, synaptic transmission, ion transport, cell adhesion and Wnt signalling, in keeping with the MS neuronal identity and function (Extended Data Fig. 1g and Supplementary Table 1). However, we observed immune-related functional categories, suggesting the unexpected presence of immune cells within the MS microenvironment (Extended Data Fig. 1g and Supplementary Table 1). Therefore, we performed systematic immunostaining for immune cell types including dendritic, natural killer, B and T cells; neutrophils and macrophages, which revealed the presence of Iba1 and F4/80 positive macrophages in proximity to MS (Fig. 1a–c, Extended Data Figs. 1h and 2a,b and Supplementary Videos 1–3).

Next, by using CX3CR1[GFP] transgenic mice, we found that all CX3CR1[+] macrophages are F4/80 positive and localize within the MS capsule (Fig. 1g), suggesting the possibility of tissue residency (Fig. 1d–f)[14]. CX3CR1[+] macrophages were found in extrafusal skeletal muscle only sporadically (Fig. 1d–f). We wondered whether macrophages also inhabited the MS microenvironment in human skeletal muscle. Indeed, we found CD68-positive macrophages in proximity to human MS (Fig. 1h), whereas these were absent in NMS skeletal muscle.

As intrafusal myofibers are innervated by γ-motoneurons, we asked whether they were also surrounded by MSMP. However, F4/80 and bungarotoxin immunostaining showed a lack of MSMP in proximity to γ-motoneurons (Extended Data Fig. 2c,d). Similarly, macrophages were not enriched in proximity to neuromuscular junctions (NMJ) (Extended Data Fig. 2e,f), which allows muscle contraction following activation of α-motoneurons.

Together, these data indicate the presence of tissue specific CX3CR1[+] macrophages residing in close proximity to MS.

## MSMP express glutamate-pathway-related genes

The presence of macrophages in the MS and their tissue-specificity led us to wonder whether they might play a role in MS physiology. First, we investigated the specific molecular signatures of MSMP. We performed RNA-seq from MSMP isolated by fluorescence-activated cell sorting (FACS) by using anti-CD45, CD11b, F4/80 and CX3CR1 antibodies to strongly enrich for MSMP from mouse skeletal muscles (Extended Data Fig. 3a–d). The purity of isolated muscle macrophages was confirmed by flow cytometry (Extended Data Fig. 3e,f). We also isolated tissue-resident macrophages from highly innervated organs such as the lung alveoli and heart and performed RNA-seq[15,16]. Next, we compared the gene expression profiles of MS, lung and heart-associated macrophages as well as sciatic nerve-associated macrophages (SNMP) by using a previously published RNA-seq dataset[17]. PCA (Fig. 1i), sample-to-sample distance heatmap (Fig. 1j) and differential gene expression analysis (Fig. 1k and Supplementary Table 2) all showed a clear distinction among the four datasets, indicating that MSMP have unique molecular signatures. GO analysis revealed that MSMP have fewer gene expression profiles associated with the immune response as compared to lung- and heart-associated macrophages, and SNMP (Extended Data Fig. 4b,d–f and Supplementary Table 2). Furthermore, compared to lung and heart macrophages, MSMP showed an upregulation of GO categories associated with muscle contraction, neuronal projection and synaptic activity (Extended Data Fig. 4a,c and Supplementary Table 2). Although both MSMP and SNMP reside in a neuronal environment, specific GO categories enriched in MSMP versus SNMP also showed enrichment for neuronal and synaptic molecular pathways (Fig. 1l). GO of these neuronal-specific pathways identified glutamate-related genes (Fig. 1m). Further GO analysis from genes involved in glutamate mechanisms indicated a possible role in excitatory neurotransmission, synaptic exocytosis, control of neuromuscular balance and glutamatergic synapse (Fig. 1n), which are typical functions of neurons rather than macrophages. Moreover, RNA-seq revealed the expression of genes involved in glutamate transport, synthesis and release (Fig. 1o)[18–20].

Together, these data indicate that MSMP have unique molecular signatures and they possess the transport, synthesis and release gene expression machinery for the neurotransmitter glutamate. These molecular and cellular features may enable MSMP to modulate neural activity in MS and ultimately influence muscle contraction.

## MSMP modulate muscle contraction in milliseconds

We proposed that MSMP regulate neural activity and muscle responses by using the main excitatory neurotransmitter glutamate. Optogenetic stimulation of CX3CR1[+] heart macrophages and spinal microglia has recently been used to explain the membrane properties and modulatory roles of macrophages[16,21]. Therefore, we decided to use CX3CR1[Cre]::ChR2[YFP] transgenic mice to express Channel rhodopsin-2 (ChR2) in MSMP to optically stimulate MSMP in selected muscles in the hindlimb and simultaneously record sensory and muscle responses in vivo (Fig. 2a,b). Patch-clamp experiments from FACS-sorted MSMP (Extended Data Fig. 3b–f) confirmed that ChR2 activation induces their depolarization (Extended Data Fig. 5a–e).

Strikingly, optical stimulation of MSMP elicited muscle activation in a millisecond timescale (Fig. 2c,d) and was proportional to the stimulation features (Fig. 2e). Simultaneous recordings of sensory fibres and muscles indicated that optical stimulation of MSMP recruited sensory neurons that in turn activated motor neurons leading to muscle contraction (Fig. 2f–h). This sequence of events was further supported by the observation that high-frequency optical stimulation of MSMP or dorsal root transection abolished the motor responses, indicating a monosynaptic activation of sensory neurons and polysynaptic activation of motor neurons (Extended Data Fig. 6a–d). Last, optical illumination of distinct muscles evoked response only in the optically targeted muscle, indicating high spatial-temporal selectivity of neural and muscular responses (Extended Data Fig. 6e,f). Together, these findings suggest that optical stimulation of MSMP modulates neural activity and muscle response throughout the reflex arc of the stretch reflex.

To investigate whether MSMP may directly regulate the most rapid sensorimotor response of the mammalian motor system, we performed the first of a series of loss of function experiments in which we perturbed the activity of MSMP. Given the particularly fast nature of the stretch reflex response, we used again an optical actuator to accurately inhibit MSMP activity. Patch-clamp experiments after optical stimulation (593 nm wavelength) on cultured CX3CR1[Cre+]/NpHR3[YFP+] cells hyperpolarized the MSMP membrane potential for prolonged time (Extended Data Fig. 5f–j). Thus, we examined whether MSMP-dependent activation of sensory neurons played a role in regulating the degree of the stretch reflex after progressively greater degrees of muscle elongation (Fig. 2i–m). To this end, we anaesthetized mice expressing the inhibitory opsin (NpHR3[+/−]) or their negative littermates (NpHR3[−/−]) and initiated stretch reflexes in the gastrocnemius muscle in response to increasing degrees of muscle elongation (Fig. 2i–k). As expected, we observed responses with larger amplitudes in response to greater degrees of muscle extension (Fig. 2j) that did not change in response to light in the negative littermates

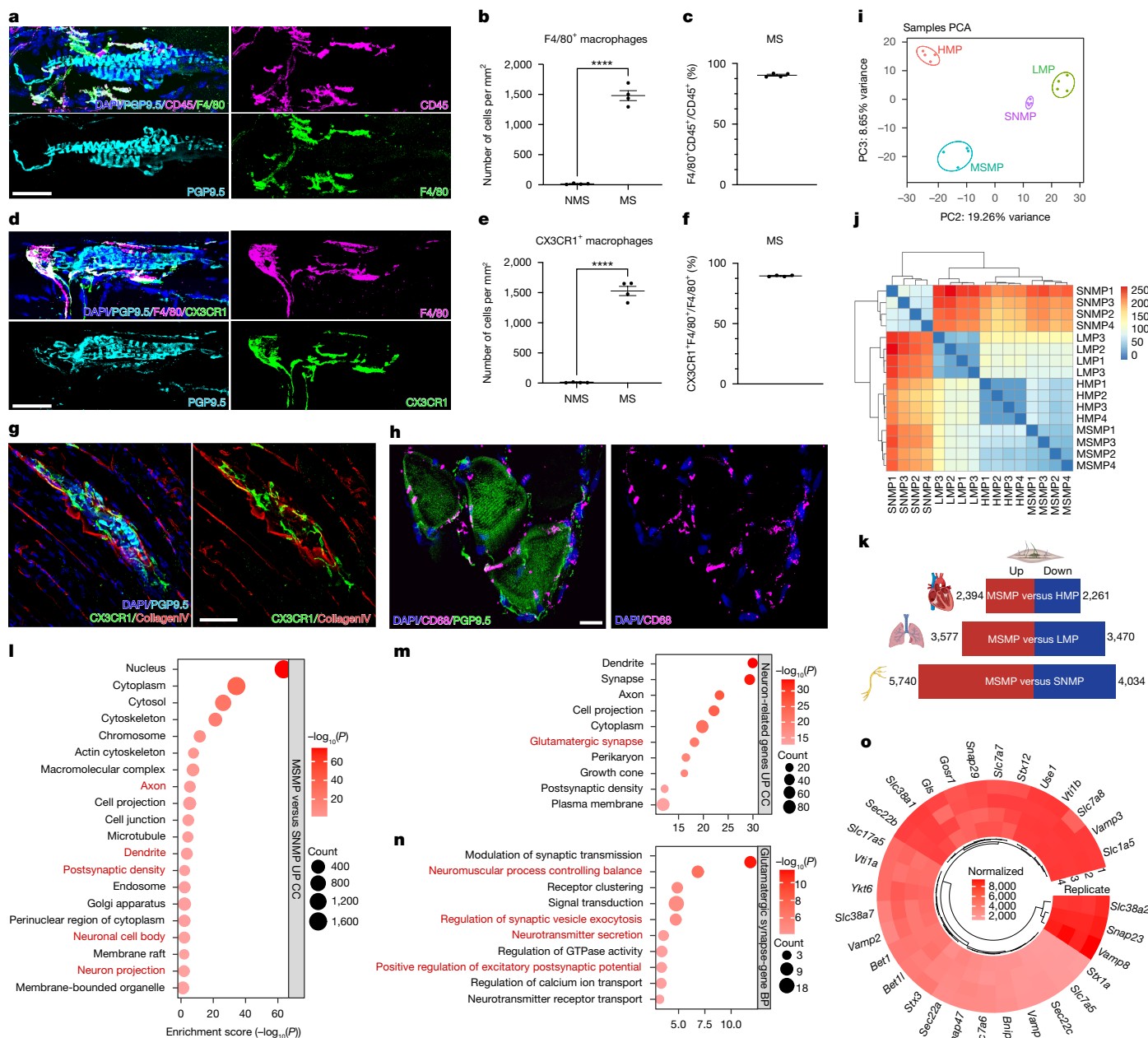

**Fig. 1 | CX3CR1⁺ macrophages in proximity to MS express a distinct molecular signature enriched in neuronal genes for glutamatergic transmission.**
**a**, Immunostaining for CD45/F4/80⁺ macrophages in mouse hindlimb skeletal muscle. MS are labelled with PGP9.5. **b**, F4/80⁺ macrophages in MS and NMS areas ($n = 4$ biological independent mice; two-tailed unpaired Student's $t$-test; mean ± s.e.m.; ****$P < 0.0001$). **c**, Percentage of CD45⁺/F4/80⁺ macrophages in CD45⁺ cells in MS ($n = 4$ biological independent mice; mean ± s.e.m.). **d**, Immunostaining for F4/80⁺/CX3CR1⁺ macrophages in proximity to PGP9.5⁺ MS. **e**, CX3CR1⁺ macrophages in MS and NMS areas ($n = 4$ biological independent mice; two-tailed unpaired Student's $t$-test; mean ± s.e.m.; ****$P < 0.0001$). **f**, Percentage of F4/80⁺/CX3CR1⁺ in F4/80⁺ MSMP ($n = 4$ biological independent mice; mean ± s.e.m.). **g**, Immunostaining of CX3CR1⁺ cells inside the capsule of MS (labelled by Collagen IV). **h**, Immunostaining showing enrichment for CD68⁺ macrophages in PGP9.5⁺ MS from quadriceps muscle of a normal subject. **i**, PCA of FACS-sorted MSMP versus FACS-sorted heart (HMP), lung (LMP) macrophages and SNMP. **j**, Correlation-based sample-to-sample distance heatmap of MSMP, HMP, LMP and SNMP. **k**, Differential expression analysis (FDR < 0.05, fold change greater than 1.5) of MSMP versus HMP, LMP and SNMP. **l**, GO of upregulated genes in MSMP versus SNMP (FDR < 0.05, fold change greater than 1.5). GO categories highlighted in red text are neuron-related pathways. **m**, GO analysis of 336 neuron-related upregulated genes in MSMP versus SNMP (FDR < 0.05, fold change greater than 1.5). **n**, GO of 77 'Glutamatergic synapse' category upregulated genes in MSMP versus SNMP (FDR < 0.05, fold change greater than 1.5). **o**, Expression of glutamine transporters, glutamate synthesis and glutamate release-related genes by MSMP. Scale bars, 50 μm (**a**,**d**,**g**), 100 μm (**h**). Illustrations in **k** created using BioRender (https://biorender.com).

NpHR3⁻ᐟ⁻ (black dots in Fig. 2k–m). By contrast, optical stimulation to the gastrocnemius muscle of NpHR3⁺ᐟ⁻ mice during progressive muscle elongations resulted in the amplitude reduction of the stretch reflexes (orange dots in Fig. 2k–m). Notably, this decrease was more pronounced with greater degrees of muscle elongation, reaching approximately 35% of reduction compared to control condition with the greatest stretch (Fig. 2l,m).

These series of experiments identify MSMP as a new component of the stretch reflex circuit, demonstrating the direct modulation of the reflex by dynamically engaging spinal circuits and regulating motor activity.

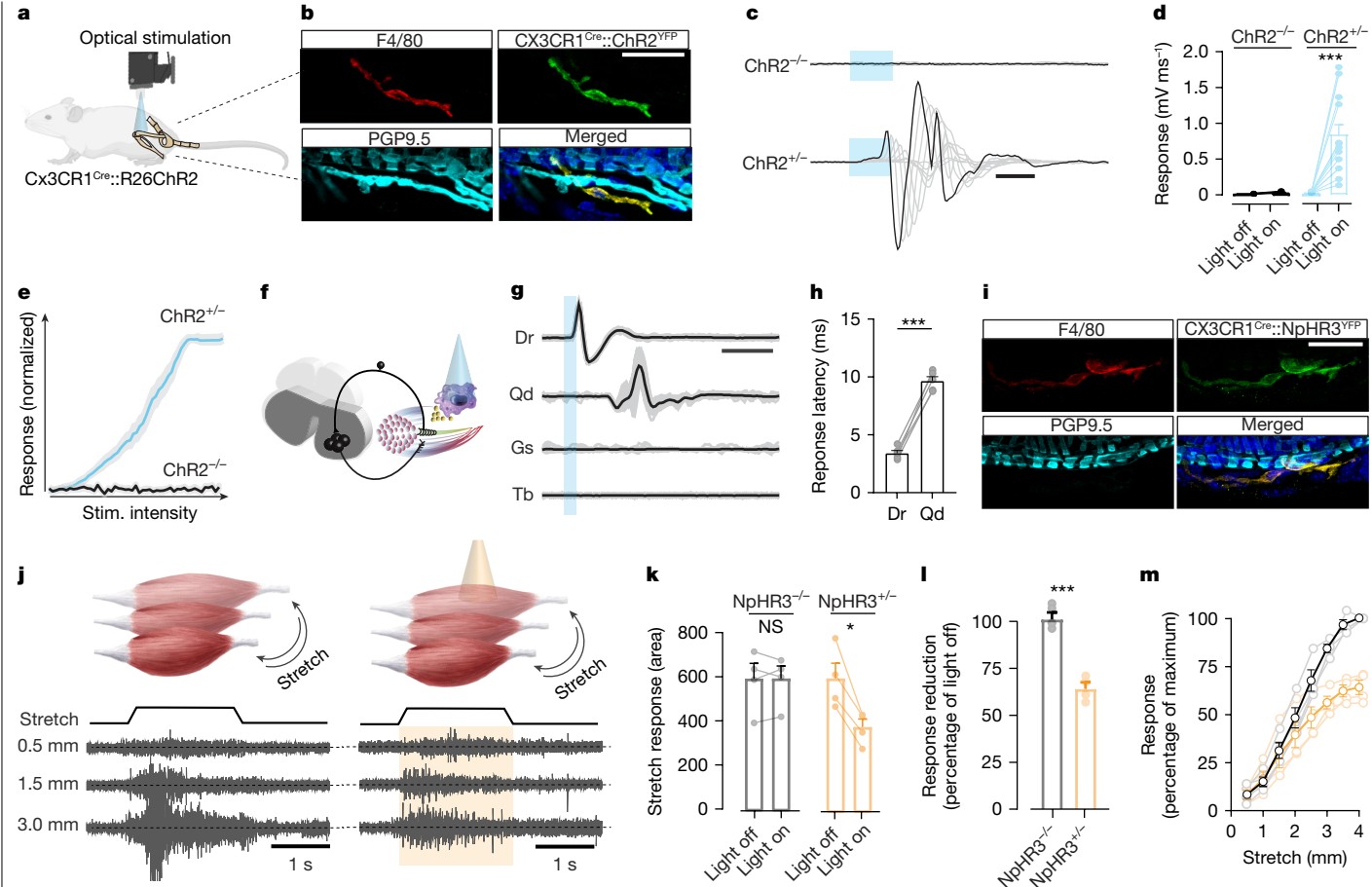

**Fig. 2 | MSMP activation or inhibition elicit rapid neural activity with muscle contraction or impair muscle activity in response to stretch, respectively. a**, Schematic of the experimental design. Optogenetic stimulation a specific hindlimb muscle in CX3CR1^Cre::ChR2^YFP mice to activate CX3CR1^+ macrophages and simultaneously record sensory and muscle responses animals while illuminating. **b**, Immunostaining for the Cre-driven expression of opsin (ChR2^YFP) in F4/80^+ cells located in the PGP9.5^+ MS. **c–e**, Traces and graphs showing the response of the gastrocnemius muscle in a CX3CR1^Cre::ChR2^YFP mouse (ChR2^+/−) and its negative littermate (ChR2^−/−) on optical stimulation (stim.) (**c**), with quantification of the response amplitude (**d**) and the stimulus–response curve (**e**) in both groups (*n* = 7 biological independent mice; Wilcoxon test, mean ± s.e.m.). **f–h**, Schematic of the experiment in CX3CR1^cre::ChR2^YFP mice (**f**) with representative traces of response from the quadriceps (Qd), Tibialis (Tb) and Gastrocnemius (Gs) muscles with the sensory root response (Dr) during short optical stimulation (2 ms duration, **g**) with quantification

of latencies for sensory (Dr) and muscle (Qd) responses (**h**, *n* = 5 biological independent mice, paired Student's *t*-test, mean ± s.e.m.). **i**, Immunostaining showing colocalization of NpHR3^YFP and F4/80 in CX3CR1^Cre::NpHR3^YFP mice in PGP9.5^+ MS. **j**, Schematic of the experiment with representative muscle responses to greater stretches of the gastrocnemius muscle in mice with and without light stimulation. **k**, Absolute amplitude responses in CX3CR1^Cre mice expressing halorhodopsin (NpHR3^+/−) and their negative littermates (NpHR3^−/−). *n* = 4 biological independent mice per condition, unpaired Student's *t*-test, mean ± s.e.m. **l**, Amplitude response as percentage to the control condition (no light) (*n* = 4 biological independent mice per condition, paired Student's *t*-test, mean ± s.e.m.). **m**, The stimulus–response in relation to stretch in NpHR3^+/− mice without (black) and with (orange) inactivation of MSMP (*n* = 4 biological independent mice). Significance levels: NS, not significant; *^P* < 0.05, ***^P* < 0.001. Scale bars, 30 μm (**b**,**i**), 5 ms (**c**), 10 ms (**g**), 1 s (**j**). Illustration in **a** created using BioRender (https://biorender.com).

## MSMP modulate neural activity via glutamate

Next, we investigated the nature of the fast communication between MSMP and sensory neurons. Recently published RNA-seq data from dorsal root ganglia (DRG) proprioceptive Ia sensory neurons suggested that they express the glutamate receptors AMPA and *N*-methyl-*D*-aspartate (NMDA)[22]. Indeed, immunostaining for NMDA and AMPA receptors showed that they are expressed within the MS afferents around the intrafusal muscle fibres (Fig. 3a), supporting the model in which MSMP might activate the stretch reflex through glutamate release. To test this hypothesis, we optically stimulated MSMP in anaesthetized CX3CR1^Cre::ChR2^YFP mice while simultaneously recording the sensory root and the gastrocnemius muscle (Fig. 3b) as previously (Fig. 2a–e). As expected, the optical stimulation elicited sensory and motor responses that were instantaneously abolished by local intramuscular injection of specific blockers for AMPA/kainate and NMDA receptors (CNQX and AP-5, Fig. 3c,d). Direct electric stimulation of the

sensory root restored both the sensory and motor responses even in the presence of the blockers (Fig. 3c,d), indicating a monosynaptic, glutamate-dependent excitation of sensory neurons by optical stimulation of macrophages. Next, we decided to perturb the presynaptic component of the circuit by preventing SNARE vesicle release from the MSMP using cell-type specific silencing based on Cre-dependent expression of the tetanus toxin light chain (TeLC)[23–28]. To this end, we injected a Cre-dependent viral vector expressing either the TeLC (adeno-associated virus (AAV)-Flex-TxTeLC-mcherry, TeLC, Fig. 3e) or a fluorophore as control (AAV-Flex-mCherry) into the gastrocnemius of CX3CR1^Cre::ChR2^YFP transgenic mice to achieve specific expression in MSMP. Animals from both groups underwent optical stimulation and recording of the tibialis (control) and gastrocnemius muscles (TeLC) in both hindlimbs (Fig. 3f). As expected, all muscles from the control groups responded to the site-specific stimulation with large amplitude and short latency responses (Fig. 3f,g). By contrast, the amplitude of the gastrocnemius response was strongly reduced or absent in TeLC

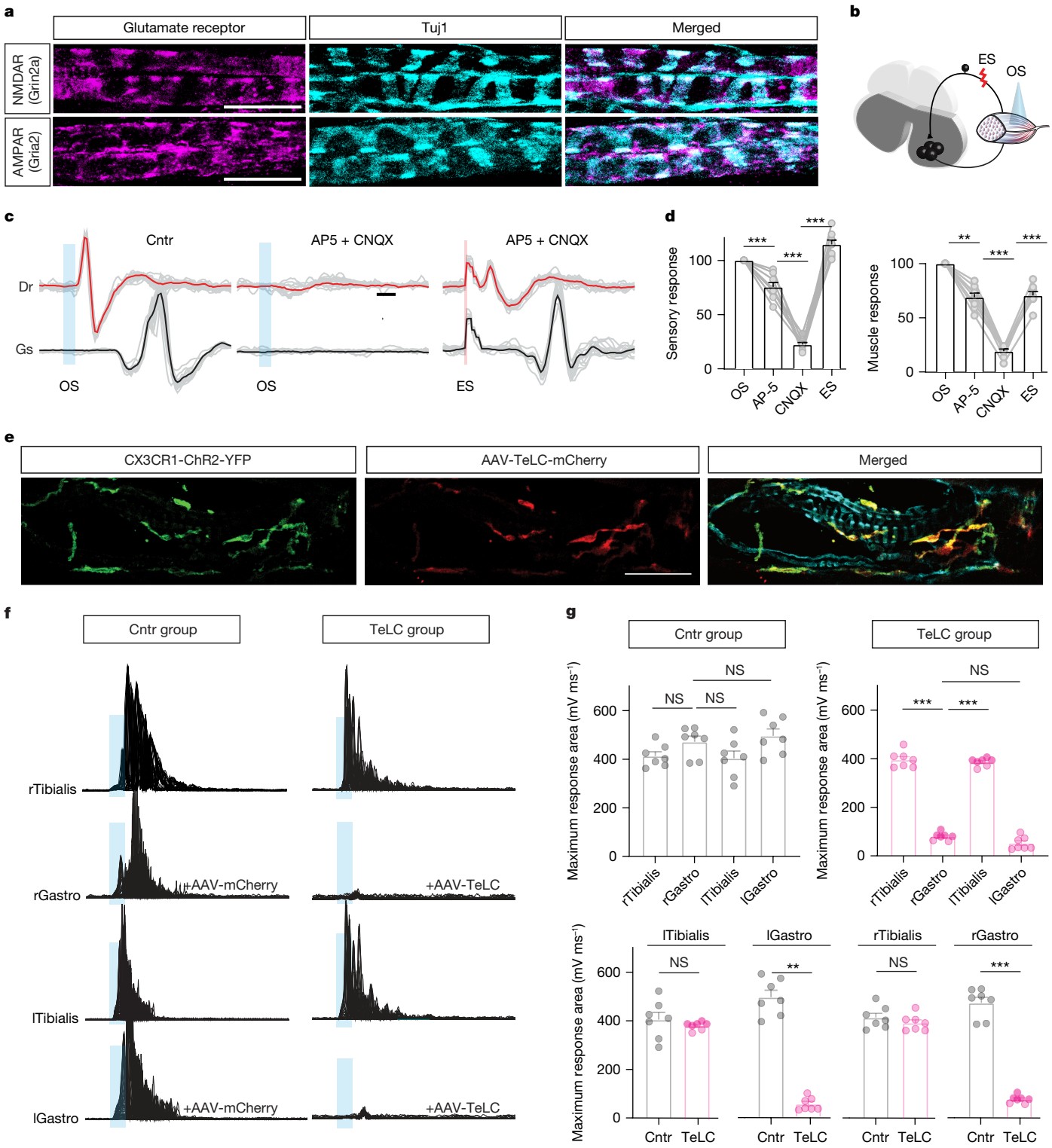

**Fig. 3 | MSMP modulate neural activity and muscle contraction by means of fast glutamatergic transmission. a**, Representative immunostaining of glutamate receptors (Grin2a, NMDA and Gria2, AMPA receptors) in MS afferents (labelled with Tuj1). **b**,**c**, Schematic (**b**) and representative traces (**c**) for sensory and muscle responses in CX3CR1$^{cre}$::ChR2$^{YFP}$ mice induced by optical stimulation (OS) (left panel, **c**), optical stimulation with AP-5 and CNQX (middle panel, **c**) and electric stimulation (ES) of the dorsal root (50 ms, 1 Hz, right panel, **c**). **d**, Bar graphs showing normalized sensory (left graph) and muscle response (right panel) ($n = 7$ biological independent mice; one-way ANOVA, Tukey's multiple comparisons test; mean ± s.e.m.) in **c**. **e**, Representative immunostaining for the expression of TeLC-mCherry (red) in CX3CR1$^{cre}$::ChR2$^{YFP}$ macrophages in mouse gastrocnemius MS (labelled with PGP9.5, cyan) with

AAV9/2-FLEX-mCherry-TeLC (referred to as TeLC) intramuscular injection. **f**–**g**, Representative traces (**f**) and quantification of response amplitude (**g**) after short optical stimulation (5 ms) in anaesthetized CX3CR1$^{cre}$::ChR2$^{YFP}$ mice that received a viral injection of either a control fluorescent marker mCherry (control group (cntr), $n = 7$ biological independent mice; one-way ANOVA, Tukey's multiple comparisons test; mean ± s.e.m.) or TeLC-mCherry (TeLC, $n = 7$ biological independent mice; one-way ANOVA, Tukey's multiple comparisons test; mean ± s.e.m.) in the gastrocnemius muscles. Tibialis muscles were not injected with either control virus (AAV-mCherry) or TeLC virus (they were used as negative controls in both control and TeLC mice). Significance levels: NS, not significant, **$P < 0.01$, ***$P < 0.001$. Scale bars, 20 μm (**a**), 2 ms (**c**), 50 μm (**e**).

animals (Fig. 3f,g), indicating that MSMP mediated the sensory-motor activation of the spinal cord underlying muscle contraction.

These findings collectively reveal that MSMP exert swift control over neural activity and muscle contraction through glutamatergic signalling. The fast chemical connection between MSMP and the neural system allows MSMP to operate at a millisecond timescale and modulate spinal activity and muscle contraction with high spatiotemporal precision.

## MSMP uptake and convert glutamine into glutamate

As MSMP modulate neural activity, and in particular sensory feedback, driving muscle contraction, we asked whether MSMP are in turn modulated by neuronal activity. A first positive indication on this direction came from our RNA-seq that MSMP express sodium, potassium and calcium voltage-gated channels (Supplementary Table 2), a repertoire of channels required for active membrane responses to inputs of neuronal origins. Thus, we investigated whether activation of the neural components of the stretch reflex arc may induce activation of the MSMP. We carried out single-cell transcriptomics of sorted MSMP after optogenetic stimulation of DRG proprioceptive sensory neurons by using Pv-ChR2 transgenic mice (Pv-ChR2). This experimental design allowed us to bypass the MS and activate directly the sensory fibres generating the electric analogue of the stretch reflex, the so-called 'H reflex'[29]. Analysis of the transcriptomics data identified five main clusters of cells that expressed tissue residency markers (Fig. 4a and Extended Data Fig. 7a–c); immunofluorescence experiments by using antibodies against the marker proteins Il1rn, Hspa1a and Pecam1 showed similar spatial sublocalization within the MS (Extended Data Fig. 7d,e). Differential gene expression analysis revealed that MSMP respond to optogenetic stimulation of DRG proprioceptive sensory neurons leading to muscle contraction (Fig. 4b,c, Extended Data Figs. 7c and 8a and Supplementary Table 3). The MSMP molecular response is underpinned by the increased expression of genes involved in macrophage activation, differentiation and chemokine signalling as well as by immediate early genes such as *Fos*, *Jun*, *Ier3/5*, *Egr1* and *Atf4* (Fig. 4b, Extended Data Fig. 8a and Supplementary Table 3), which are typically induced in activated neurons[30–34].

We next focused our attention on glutamate metabolism: the single-cell data confirmed that MSMP clusters express the full repertoire of glutamate uptake, synthesis and release[18–20,35] (Fig. 4c,f and Extended Data Fig. 7f); however, among these, only glutaminase, which converts glutamine into glutamate, was increased after optogenetic stimulation of proprioceptive afferents (Fig. 4c and Supplementary Table 3). The MSMP gene expression of glutaminase and of the early response genes *Egr1*, *JunB*, *Atf4* and *Ier5* was also induced following a physiological task such as swimming (Extended Data Fig. 8b). Because it is well known that glutamine is transiently synthesized and released during muscle contraction[36–38], we proposed that MSMP might convert glutamine into glutamate to activate sensory neurons by means of NMDA and AMPA glutamate receptors. To model this, we exposed cultured MSMP to glutamine or vehicle and measured glutamate concentration in the culture medium 2 or 8 h later. As gene expression of glutaminase was increased after optogenetic stimulation and swimming, we also tested whether glutaminase is required for glutamine to glutamate conversion in MSMP by adding either vehicle or the glutaminase inhibitor CB-839 (Fig. 4c,d). Indeed, we found that glutamine delivery led to an increase of glutamate in the medium, which was blocked by the glutaminase inhibitor CB-839 (Fig. 4d,e).

As previous studies have shown that glutamine transporters can induce membrane depolarization, calcium mobilization and glutamate release[39–43] and our RNA-seq showed the expression of glutamine transporters (*Slc1a5*, *Slc7a5-a8*, *Slc38a1*, *Slc38a2*, *Slc38a7*) in MSMP (Figs. 1o and 4f), we investigated whether glutamine could directly induce membrane depolarization and lead to glutamate release.

Indeed, glutamine activated MSMP by increasing calcium mobilization and glutamate release, as shown by a rise in the Fluo-4 signal (Fig. 4g,h) and glutamate concentration (Fig. 4i). Inhibiting glutamine transporters (MeAIB and V9302) significantly reduced both MSMP activation and glutamate levels (Fig. 4h,i). Finally, adding $Ca^{2+}$ chelators (BAPTA-AM and EGTA) fully inhibited MSMP activity and glutamate release, indicating dependence on calcium signalling (Fig. 4h,i).

Together, these experiments show that MSMP respond to sensory modulation and muscle contraction induced by both optical stimulation of Ia sensory neurons and by an ethological relevant motor task such as swimming. Furthermore, MSMP uptake glutamine that is converted into glutamate and released through a glutamine transporter's calcium-dependent mechanism.

## MSMP modulate locomotion-related gene expression

To begin investigating the physiological relevance of MSMP we first asked whether MSMP depletion would result in impairment of sensory function. Thus, we examined the changes in molecular signatures of DRG proprioceptive sensory neurons after macrophage depletion. Initially, we depleted macrophages with a combination of clodronate liposomes and the colony stimulating factor 1 receptor (CSF-1R) inhibitor BLZ945. As shown by immunostaining, this led to the almost complete depletion of MSMP at both 4 and 11 days after treatment and to a roughly 40% reduction of SNMP (Extended Data Fig. 9a–d). Subsequently, we performed single-nucleus RNA-seq in DRG from control or macrophage-depleted mice (Extended Data Fig. 9e). As expected, we identified several clusters corresponding to the diversity of DRG sensory neuron subclasses (Extended Data Fig. 9f and Supplementary Table 4). GO analysis of differentially expressed genes within proprioceptive sensory neurons revealed that macrophage depletion led to the downregulation of molecular mechanisms that support the function of proprioceptive MS afferents. These included axonal, synaptic, neurotransmitter and ion transport pathways as well as locomotion (Extended Data Fig. 9g and Supplementary Table 4). This promoted us to assess whether locomotor behaviour would be compromised in macrophage-depleted mice. After depleting macrophages, locomotor activity was assessed by behavioural tests including grid walk and treadmill 7 days (day 11) after depletion versus control mice. We found that macrophage depletion led to a significantly increased number of missteps and alterations of spatiotemporal gait and coordination parameters at low (20 cm s⁻¹) and high (40 cm s⁻¹) speeds (Extended Data Fig. 9h–j and Supplementary Tables 5 and 6).

Together, these data indicate that macrophages exert influence over neuronal activity regulating key aspects of the sensory-motor function driving movements.

## Selective disruption of MSMP impairs locomotion

To specifically investigate how MSMP modulate movement and locomotor behaviours we reasoned that stretch reflexes represent a fundamental building block of the sensory-motor system. They are crucial to perform accurate movements and provide rapid corrections for unpredictable perturbations in stereotyped movements such as walking and swimming. Especially in the case of the gastrocnemius, the stretch reflexes are engaged during locomotion to produce force during the extensor phase of the step/stroke cycle, supporting both the vertical and horizontal components of the limb movement[44,45]. Similarly, proprioceptive inputs represent one of the major sources of sensory feedback to the nervous system during swimming[46]. Thus, initially we decided to examine the locomotion of mice before and after the injection of either TeLC for genetic silencing of MSMP (TeLC group) or fluorophore (mCherry control group) in the gastrocnemius muscle during swimming (Fig. 5a–h). Detailed kinematic analysis of

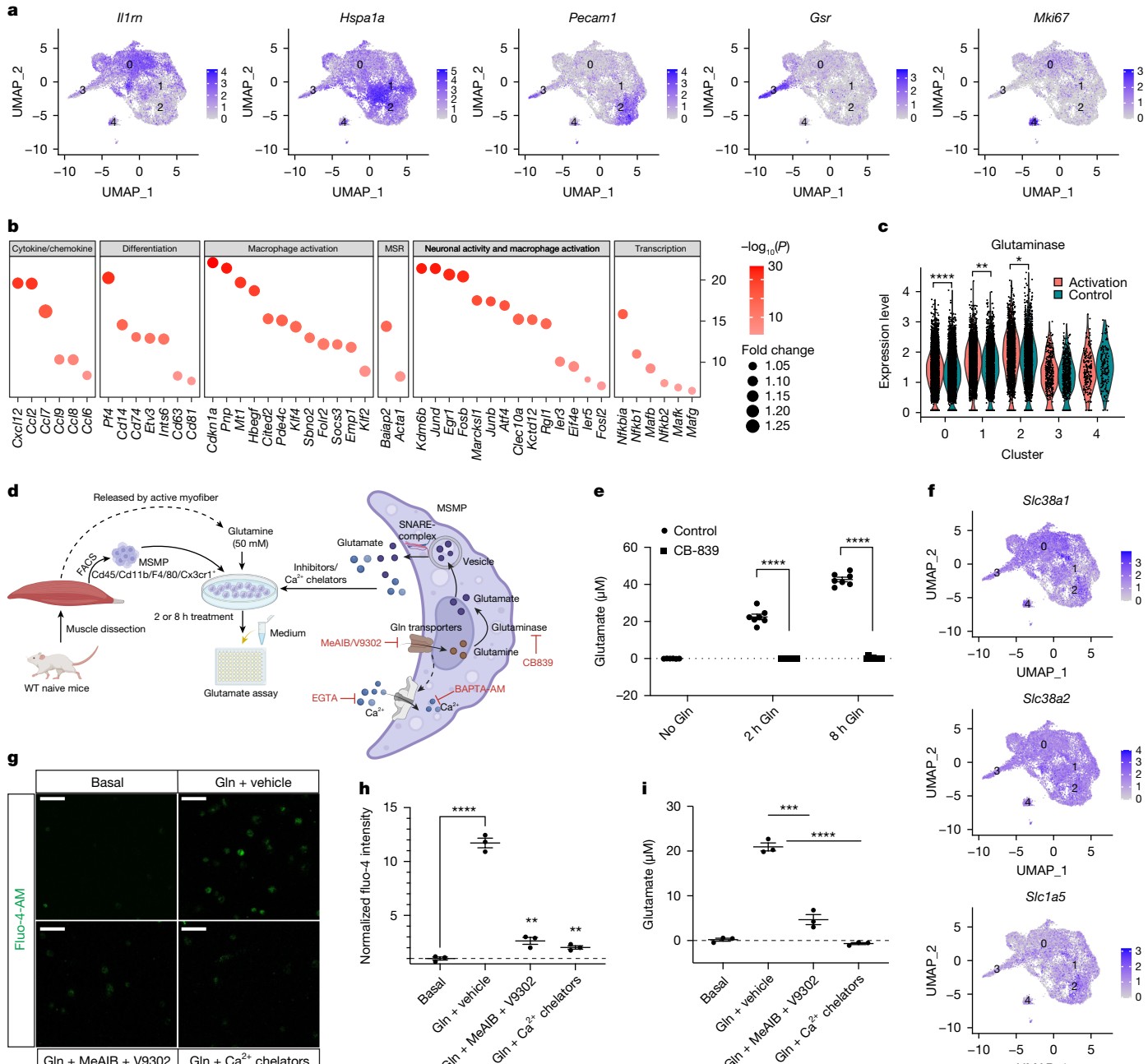

**Fig. 4 | MSMP respond to proprioceptive neuronal activation and convert glutamine into glutamate by means of glutaminase. a**, UMAP results show the expression of marker genes for five clusters of 18,538 FACS-sorted MSMP. **b**, DE analysis from identified macrophage clusters shows increased expression of genes involved in cytokine or chemokine signalling, macrophage or neuron activation, transcription, differentiation and macrophage shape remodel (MSR) (FDR < 0.05, fold change greater than 1). **c**, DE analysis shows significantly increased expression of glutaminase in response to Pv neuron activation in MSMP cluster 0, 1 and 2 (FDR < 0.05, fold change greater than 1). **d**, Experimental design of in vitro MSMP culture medium glutamate assay with FACS-sorted MSMP treated with vehicle or glutamine or glutaminase inhibitor (CB-839) or glutamine transporter inhibitors (MeAIB and V9302) or $Ca^{2+}$ chelators (BAPTA-AM and EGTA). **e**, Glutamate was detected in MSMP culture medium 2 or 8 h after 50 mM glutamine treatment. The release of glutamate was diminished by glutaminase inhibitor (CB-839) (n = 6 biological replicates; one-way ANOVA,

Tukey's multiple comparisons test; mean ± s.e.m.). **f**, Expression of glutamine transporters SNAT1 (*Slc38a1*), SNAT2 (*Slc38a2*) and ASCT2 (*Slc1a5*) by MSMP. **g**,**h**, Immunostaining (**g**) and bar graphs (**h**) showing elevated MSMP intracellular $Ca^{2+}$ level (Fluo-4-AM signal) induced by 50 mM glutamine treatment that was significantly reduced by glutamine transporter inhibitors (MeAIB + V9302) and $Ca^{2+}$ chelators (BAPTA-AM and EGTA) (n = 3 biological replicates; one-way ANOVA, Tukey's multiple comparisons test; mean ± s.e.m.). **i**, Glutamate was detected in MSMP culture medium after 2 h of 50 mM glutamine treatment. The medium glutamate level was significantly reduced by glutamine transporter inhibitors (MeAIB + V9302) and $Ca^{2+}$ chelators (BAPTA-AM and EGTA) (n = 3 biological replicates; one-way ANOVA, Tukey's multiple comparisons test; mean ± s.e.m.). Significance levels: NS, not significant, *$P < 0.05$, **$P < 0.01$, ***$P < 0.001$; ****$P < 0.0001$. Scale bar, 50 μm. Illustrations in **d** created using BioRender (https://biorender.com).

gait and PCA indicated that TeLC injected mice showed a change in the hindlimb movement supporting the swimming stroke with an increase maximal angular excursion of the ankle and hind paw, the two joints

controlled by the gastrocnemius, increasing the facto the stretch of this muscle (Fig. 5d–g and Supplementary Video 4). However, a reduction in the angular excursion of the hip and the hindfinger prevented any

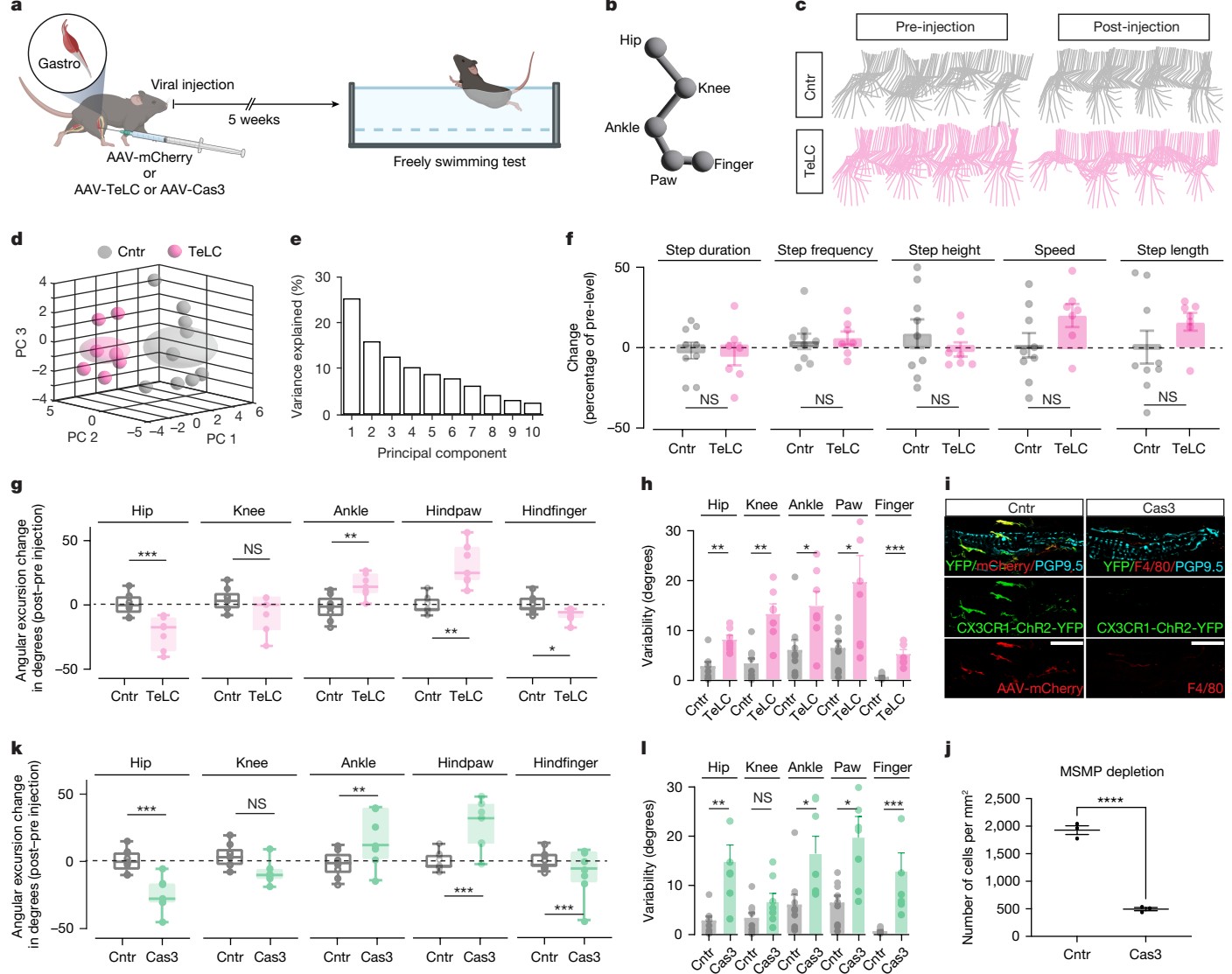

**Fig. 5 | Selective disruption of MSMP activity alters locomotor strategies in freely swimming mice. a**, Experimental design of swimming tests after selective disrupt MSMP neurotransmitter release (AAV-TeLC) and MSMP depletion (AAV-Cas3). **b**, Schematic representation of key hindlimb points. **c**, Stick diagram decomposition of the hindlimb stroke cycle in CX3CR1$^{cre}$::ChR2$^{YFP}$ mice before and after viral delivery of TeLC (magenta) or a control fluorescent marker (grey) into both gastrocnemius muscles. **d,e**, PCA of gait features clustered the control (grey) and TeLC (magenta) groups based on the first three principal components with individual animals represented by small circles and group centroids by large sphere (**d**) and the first ten components explained variability (**e**). **f–h**, Graphs showing changes in gait features (**f**, two-tailed unpaired Student's *t*-test, mean ± s.e.m.), angular excursion (**g**, Welch's *t*-test, mean ± s.e.m.) and angular excursion variability

(**h**, Mann–Whitney test, mean ± s.e.m.) of the control group (grey, *n* = 9 biological independent mice) and TeLC group (magenta, *n* = 7 biological independent mice). **i,j**, Representative confocal images of gastrocnemius MS from a CX3CR1$^{cre}$::ChR2$^{YFP}$ mouse injected with either AAV-DIO-mCherry (control) or AAV-DIO-Caspase3 to deplete MSMP (Cas3 group) (**i**), with the relative count of MSMP 5 weeks after the viral injection (**j**) (*n* = 3 biological independent mice; two-tailed unpaired Student's *t*-test; mean ± s.e.m.). **k,l**, Graphs showing changes in angular excursion (**k**, Welch's *t*-test; mean ± s.e.m) and angular excursion variability (**l**, Mann–Whitney test; mean ± s.e.m.) of the control group (grey, *n* = 9 biological independent mice) and caspase group (green, *n* = 7 biological independent mice). Significance levels: NS, not significant, *\*P* < 0.05, \*\**P* < 0.01, \*\*\**P* < 0.001; \*\*\*\**P* < 0.0001. Illustrations in **a** created using BioRender (https://biorender.com).

modification in the resultant spatiotemporal gait parameters, including step duration, frequency, height, speed and length (Fig. 5f,g). Furthermore, angular excursion variability was significantly impaired in the TeLC group, particularly at the ankle and hind paw (Fig. 5h). A similar set of experiments with cell-specific depletion of CX3CR1$^{Cre+}$ MSMP in the gastrocnemius muscles by the muscle selective injection of a Cre-dependent AAV-DIO-caspase3 (Cas3 group) (Fig. 5i,j) confirmed that depletion of MSMP leads to an abnormal stretch of the gastrocnemius (Fig. 5k) without modifications of strength-related spatiotemporal gait parameters (Extended Data Fig. 10a–c). The MSMP depleted mice also showed a major variability in the hindlimb movements, indicating

a reduced ability of rapidly correcting movement in response to perturbation, in keeping with alteration of the proprioceptive system and stretch reflex (Fig. 5l). Similar kinematic abnormalities after TeLC-mediated MSMP inactivation or Cas3-dependent MSMP depletion were observed in mice tested in a skilled walking test over a ladder (Extended Data Fig. 10d–f).

These findings collectively indicate that MSMP play a critical role in modulating the sensory and motor components underlying locomotion. Disruption of MSMP activity leads to significant alterations in motor coordination and sensory feedback, essential components for maintaining smooth and effective locomotion.

## Discussion

The stretch reflex, a fundamental neural response to external mechanical stimuli, is crucial for maintaining coordinated motor function[2–8,47]. The present study has unearthed a new dimension to the stretch reflex, revealing the involvement of tissue-resident macrophages and glutamate-dependent signalling in this fundamental neuromuscular circuit. This previously unrecognized macrophage population is resident within the capsule of MS in close proximity to the sensory proprioceptive afferents. Tissue-resident macrophages are highly heterogenous and pleiotropic immune cells with both homeostatic and reactive functions in response to signals present in their local microenvironment[16,17,48,49]. However, their presence and role within the MS microenvironment have remained undetermined until now. How and when MSMP migrate into the skeletal muscle during development remains to be investigated.

RNA-seq revealed that MSMP have unique molecular signatures, indicating a role in the communication between the skeletal muscle and the nervous system. We found that MSMP possess the transport, synthesis and release gene expression machinery for the neurotransmitter glutamate, whereas MS sensory afferents express AMPA and NMDA glutamate receptors, suggesting cross-talk between MSMP and the MS[18–20,22,35]. Single-cell MSMP transcriptomics after optogenetic stimulation of DRG proprioceptive sensory neurons leading to muscle contraction, revealed that MSMP respond to activation of the MS circuitry, including by elevating the expression of glutaminase, which converts glutamine into glutamate. Exposure of cultured MSMP to glutamine, which is synthesized and released during muscle contraction[20,36–38], and manipulation of glutaminase activity and glutamine transport revealed that production of glutamate requires glutaminase and its release requires glutamine transport inducing macrophage depolarization. Further, single-nucleus transcriptomic of MS sensory neurons after macrophage depletion showed that MSMP regulate molecular mechanisms that support the physiological function of proprioceptive MS afferents.

The optogenetic activation of MSMP has uncovered a new regulatory mechanism for motor control. Similarly, the impairment in locomotor performance after macrophage depletion or inactivation highlights the involvement of MSMP in motor function following the modulation of proprioceptive activity. Optical stimulation of MSMP confirmed their involvement in driving the rapid activation of Ia sensory afferents and subsequent muscle contraction, in keeping with glutamate timescales in this process. The observed swift activation of Ia sensory afferents through MSMP stimulation not only implies their contribution to the stretch reflex but also emphasizes their significance in shaping the immediate motor response to external disturbances. In fact, the demonstration that NMDA and AMPA glutamate receptor blockage in the skeletal muscle inhibits both sensory and motor responses provides a direct link between MSMP-driven activity and glutamate signalling, integrating glutamate-mediated mechanisms into the reflex arc for rapid motor execution.

The ability of MSMP to convert glutamine into glutamate to modulate muscle contraction based on metabolic demands could reveal a crucial regulatory mechanism coupling metabolism with activity. By releasing glutamate, MSMP might contribute to energy-efficient muscle function, optimizing contraction strength in accordance with the muscular energy resources. This metabolic regulation could be particularly relevant during sustained or strenuous activities, preventing excessive energy consumption and promoting endurance. The impairment in sensorimotor tasks after skeletal muscle MSMP silencing, inhibition of SNARE vesicle release or depletion, indicate that MSMP contribute to the coordination and accuracy of motor responses, enabling more refined adjustments to posture, movement and muscle tone for adaptive swimming and walking.

This could be crucial for tasks requiring delicate motor control, balance and coordination. Overall, the ability of the MSMP to integrate metabolic signals with proprioceptive feedback could provide a mechanism for the neuromuscular system to adapt to changing physiological conditions and environmental demands. In line with this, the optogenetic inhibition of MSMP activation during muscle stretch at increasing degrees of muscle extension identified macrophages as a new component of the stretch reflex circuit, revealing their dynamic contribution to muscle response in relation to the degree of muscle elongation.

In summary, this paper describes the identification of MSMP and their functional characterization in contributing to the stretch reflex, and in bolstering muscle contraction and locomotion by a fast glutamatergic transmission between MSMP and sensory afferents. However, whether MSMP convert glutamine into glutamate in vivo and whether they respond to mechanical stretch remains to be determined.

This discovery has long-ranging implications for the cellular and molecular rules that control immune–neuronal interactions in neuromuscular physiology. It might also imply the possibility that immunosuppression and diseases of the innate immune system affecting monocyte-macrophages might impair MS neuromuscular function. Last, it might bear direct relevance for conditions in which proprioception, muscle tone and contraction are affected including spinal cord injuries, myotonic muscle dystrophy, peripheral neuropathies, nerve injuries and diabetes.

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

## Methods

### Mice

Animal work was conducted according to UK Home Office licence legislation under the Animals (Scientific Procedures) Act 1986, with Local Ethical Review by the Imperial College London Animal Welfare and Ethical Review Body Standing Committee or, under the EU and Danish legislation to ensure the animals' well-being. Mice were maintained under standard housing conditions on a 12 h light/dark cycle with food and water provided, at a constant room temperature and humidity (21–24 °C and 45–65%, respectively). Details of the wild-type and transgenic mouse lines are described:

C57BL6/J mice (Charles River Laboratories); B6.129P2(Cg)-Cx3cr1 tm1Litt/J (Jackson laboratories) known as CX3CR1$^{GFP}$ mice were provided by M. Malcangio; B6J.B6N(Cg)-Cx3cr1tm1.1(cre)Jung/J), referred to as CX3CR1$^{Cre}$ (code 025524, Jackson Laboratories); B6;129S-Gt(ROSA)26Sortm32(CAG-COP4*H134R/EYFP)Hze/J), referred to as ChR2$^{YFP}$ (code 012569, Jackson Laboratories); B6;129S-Gt(ROSA)26 Sortm39(CAG-hop/EYFP)Hze/J), known as NpHR3$^{YFP}$ (code 014539, Jackson Laboratories) and B6;129P2-Pvalbtm1(cre)Arbr/J), known as Pv$^{Cre}$ (code 008069, Jackson Laboratories).

All male and female mice were used between 8 and 12 weeks of age were used for all experiments. Transgenic lines were screened by PCR analysis for the presence of the transgenes in genomic DNA purified from ear biopsies. Anaesthesia by isoflurane inhalation or intraperitoneal injection of ketamine (100 mg kg$^{-1}$)/xylazine (10 mg kg$^{-1}$) was used to kill the mice to minimize suffering. Cervical dislocation was performed following anaesthesia to ensure death. Experimenters were blinded to experimental groups during scoring and quantifications.

### Immunofluorescence

Mice were anaesthetized by receiving an intraperitoneal injection of ketamine (100 mg kg$^{-1}$) and xylazine (10 mg kg$^{-1}$). Once unconscious, the thoracic cavity was opened to expose the heart and a perfusion needle was directly inserted into the lower ventricle. The mouse was perfused with 10 ml of Dulbecco's phosphate-buffered saline (DPBS, pH 7.4, Sigma) followed by 20 ml of 4% paraformaldehyde (PFA). Mouse hindlimb skeletal muscles including extensor digitorum longus (EDL), soleus, tibialis anterior and gastrocnemius (Gastro) muscles, sciatic nerves and DRGs were fixed in 4% PFA overnight at 4 °C, then transferred to 30% sucrose for 3 days. Fresh human muscle biopsy was collected from deltoid muscles and snap frozen by immersion in isopentane precooled in liquid nitrogen. Mouse or human tissues were embedded in Tissue-Tek O.C.T. and sliced into 20–50 μm sections on a cryostat and collected on Superfrost plus slides (Fisherbrand). Human muscle sections were prefixed with Acetone at 4 °C for 30 min. Tissue sections were dried for 30 min at room temperature and rinsed with washing buffer (PBS with Tween (PBST), 0.1% Tween in DPBS) three times (5 min each time) before incubation in blocking solution (10% normal donkey serum (NDS) + 0.3 or 1% Triton X-100 in PBST) for 1 h at room temperature to reduce non-specific binding. Tissue sections were then incubated in primary antibody solution (10% NDS + 0.3 or 1% Triton X-100 in PBST, mixed primary antibodies) at room temperature or 4 °C overnight. The next day, after washing with PBST three times, tissue sections were incubated in secondary antibody solution (10% NDS + 1% Triton X-100 in PBST + mixed secondary antibodies) for 1.5–2 h at room temperature. Tissue sections were washed three times and mounted with Prolong Glass Antifade mounting solution (Life Technologies) and coverslips. A list of the antibodies used in this study is provided in Supplementary Table 7.

### Microscopy and image analysis

Images were acquired on SP8 confocal microscope (Lecia) at ×20 or ×40 magnification and processed with the LAS-AM Lecia software (Lecia) and ImageJ. A minimum of four MS, NMS and NMJ areas were imaged per biological replicate with a minimum of three biological replicates. The number of macrophages and the area of MS, NMS and NMJ were counted and measured in ImageJ. Three-dimensional (3D) imaging was conducted in ImageJ using the 3D Viewer and 3D Project plugins.

### Laser-capture microdissection

To reduce RNase-induced RNA degradation, RNaseZap (Thermo Fisher) was used to clean the hood, microscope and all dissection tools. Mice were anaesthetized by intraperitoneally injecting ketamine (100 mg kg$^{-1}$) and xylazine (10 mg kg$^{-1}$) in filtered saline and then perfused with 20 ml of sterilized cold DPBS. Muscles were dissected and directly snap frozen in precooled 2-methylbutane for 15 s (for EDL and soleus muscles) or 35 s (for tibialis anterior and Gastro muscles) and embedded in Tissue-Tek O.C.T. Muscles were cut into 30 μm sections with CryoJane cryostat and attached to Lecia PEN-membrane slides (pretreated with RNaseZap and washed with nuclease-free water, exposed to ultraviolet light for 1 h). Slides were immediately fixed with cold 75% ethanol for 1 min and processed for staining protocol (washing with 75% ethanol: 30 s three times, staining with 4% Cresyl Violet: 1 min, washing and dehydrating with 75% ethanol: 30 s twice, 95% ethanol: 30 s, 100% ethanol: 30 s and Xylene: 5 min). Stained muscle sections were placed on Laser Microdissection with Zeiss Palm Microbeam microscopy. MS and NMS areas were selectively captured and collected into laser-capture microdissection collection tubes (with 150 μl of RLT lysis buffer, Qiagen). Collection tubes were then frozen with dry ice and kept at −80 °C for RNA extraction.

### FACS of macrophages

To sort cells in vivo, mice were anaesthetized with ketamine (80 mg kg$^{-1}$) and xylazine (10 mg kg$^{-1}$). Muscle, heart and lung were dissected and kept in cold DPBS on ice after cardiac perfusion (20 ml DPBS). For muscle macrophages, to reduce the contamination of NMS macrophages, fat, perimysium, nerves and tendons were carefully removed from dissected muscles. Muscle (EDL, soleus, tibialis anterior and gastro), heart and lung were transferred to digestion buffers and cut into small cubes. Digestion buffer contained Collagenase B (2 mg ml$^{-1}$), Dispase II (0.83 mg ml$^{-1}$), DNase I (0.25 mg ml$^{-1}$) and DNase I buffer (10 nM Tris Base, 2.5 mM MgCl$_2$, 0.1 mM CaCl$_2$), dissolved in RNase-free DPBS. Tissue suspensions were incubated in 15 ml of digestion buffer in a 37 °C water bath for 50 min. Tubes were shaken every 5 min. Then 10 ml of ice-cold MojoSort buffer (BioLegend) was added to stop digestion. Cell suspension was filtered through 70 and 40 μm cell strainers in series. Then, cells were washed with 5 ml of cold MojoSort buffer twice and resuspended with 800 μl of cold MojoSort buffer for FACS staining. Cell suspension was incubated with mixed antibodies at 4 °C for 20 min in the dark. The following antibodies were used: PE-conjugated anti-CD45 (Biolegend, catalogue no. 103106, 1:100), APC/Cyaine7-conjugated anti-CD11b (Biolegend, catalogue no. 101226, 1:100), APC-conjugated anti-F4/80 (Biolegend, catalogue no. 123115, 1:100), Brilliant Violet-conjugated anti-CX3CR1 (Biolegend, catalogue no. 149027, 1:100). LIVE/DEAD Fixable Aqua Dead Cell Stain Kit (Thermo Fisher Scientific, catalogue no. L34966, 1:200) was used to identify live or dead cells. Stained cells were washed twice with 1 ml of cold MojoSort, resuspended with 600 μl of MojoSort buffer (with RNase inhibitor, 1:500), filtered through a 40 μm cell strainer and kept on ice until sorting. Becton Dickinson FACS Aria Fusion Flow Cytometer with a 100 μm nozzle was set to 4 °C to protect RNA from degradation. The gating was set to FSC-A/SSC-A-Singlet/Doublet-Live/Dead$^-$-CD45$^+$-CD11b$^+$-F4/80$^+$-CX3CR1$^+$. Single stain, Fluorescence Minus One and negative controls were used for the gating boundaries. Target cells were directly collected into the cold collection buffer (DPBS with RNase inhibitor, 1:500). Around 80,000 cells were collected from each sample. Cell suspension was centrifuged at 500$g$ for 5 min, resuspended with RLT lysis buffer and processed for RNA extraction for bulk RNA-seq,

or resuspended with cold DPBS for single-cell RNA-seq or for cell culture. FlowJo was used for data analysis.

## RNA-seq preparation and analyses

Laser-capture micro-dissected muscle tissues or macrophages sorted from muscles, hearts and lungs were processed for RNA extraction. RNA was extracted by using RNeasy micro-kit and DNase on-column digestion (Qiagen) following the manufacturer's guidelines. RNA purity and concentration were verified by Agilent 2100 Bioanalyzer. RNA with an RNA integrity number above 8 was used for library preparation. Complementary DNA (cDNA) libraries for each sample were generated at the Source Bioscience Laboratory (Cambridge, UK) by using NEBNext Single Cell/Low Input Library Prep Kit (Ilumina). Sequencing was performed using Illumina HiSeq 4000 75 base pairs (bp) paired-end sequencing to obtain 60 million reads per sample. SNMP RNA-seq datasets (Fastq) were extracted from a published dataset GSE144708 and were processed with the same pipeline as the FACS-sorted macrophages from muscles, heart and lung. Sequences were demultiplexed and adapters trimmed with bcl2fastq-v.2.20 (Illumina). Read quality controls were carried out using FastQC-v.0.11.9, trimming reads less than Phred33 quality score 20 and removing remaining adapters with Trim-Galore (v.0.6.6). RNA-seq analysis was run using the COMBINE laboratory's Salmon-DESeq2 pipeline. Salmon-v.1.6.0 was used in mapping-based mode to quantify read sets. Reads were mapped to the M25 GENCODE reference mouse genome. Salmon's output files were imported and converted by Tximeta (v.1.12.4) for DESeq2. Differential expression analysis was performed using DESeq2 (v.1.34.0) in R (v.4.1.2). Normalized counts were generated using the median of ratios method in DESeq2. Differentially expressed genes were filtered to include genes with a fold change greater than 1.5. GO and Kyoto Encyclopedia of Genes and Genomes (KEGG) pathway analysis were carried out using the latest available DAVID (Database for Annotation, Visualization and Integrated Discovery) for all differentially expressed genes. GO and KEGG terms were filtered by $P < 0.05$. Graphs and dot-plots were made using GraphPad Prism v.9.4.0 or using ggplot2-v.3.3.6 in R, respectively.

## Loss of function experiments

**Cre-dependent perturbation.** We used a Cre-dependent viral strategy to selectively silence vesicle release from a CX3CR1[+] cell of the gastrocnemius muscles. We injected 25 µl of AAV9/2-DJ/2-FLEX-mCherry -2A-FLAG-TeTxLC (Viral Vector Facility, Universität Zürich) in both gastrocnemius muscles of CX3CR1[Cre]::ChR2[YFP] mice to prevent the release of SNARE vesicles. We injected AAV9/2-DJ/2-*flex*-mCherry (Viral Vector Facility, Universität Zürich) in another group of animals (the control group). Similarly, we also used a Cre-dependent depletion strategy as an alternative by injecting in the same muscles (gastrocnemius) AAV9/2-DJ/2-*Flex*-(pro)taCasp3 (Viral Vector Facility, Universität Zürich) in a group of mice defined as Cas3. In this way, the Cre[+] cells of the gastrocnemius muscle were depleted as previously shown for other types of cell including neurons and microglia[50,51]. Five weeks later, a cohort of mice from each group were euthanized and perfused. Muscles were dissected, fixed and processed for immunostaining. The control group, the TeLC group and the Cas3 group underwent behavioural testing 5 weeks after injection. Furthermore, some of the animals from the control and TeLC groups were tested with optical stimulation of MSMP in the muscle while recording tibialis and gastrocnemius.

**Systemic depletion of macrophages.** CSF1 inhibitor, BLZ945 (4 mg per mouse per day, S7725, Selleckchem) dissolved in sterile water with 20% 2-hydroxypropyl-β-cyclodextrin (H107, Sigma) or control with 20% 2-hydroxypropyl-β-cyclodextrin were delivered to the mice for four consecutive days by oral gavage. Clodronate liposomes or control liposomes (Liposoma) were injected into mice by intraperitoneal injection (200 µl per mouse per day) on days 1 and 3. Depleted or control mice were euthanized on day 4 (4 h after the last BLZ945 gavage) or day 11. Muscles together with spinal cords and sciatic nerves were dissected and freshly processed for FACS or fixed with 4% PFA and dehydrated with 30% sucrose for cryostat sectioning followed by immunostaining. Mice without or with macrophage depletion (oral gavage of control or BLZ945 in combination with control or clodronate liposomes) were used for behavioural assessments. Mouse hindlimbs were bilaterally assessed on day 11 after depletion. Animals were randomly and equally grouped before depletion treatments (15 biological replicates for each condition). All investigators were blind to the groups and treatments.

## Behavioural tests

**Grid walk.** Mice were trained to walk through the metal grid (50 × 5 cm) plastic grid (1 × 1 cm) placed between two vertical 40 cm high wood blocks. Three runs per session were carried out and measured. For each hind paw, the total number of steps and missteps per run were analysed by a blinded investigator.

**Treadmill.** Mice were trained to run on a motorized treadmill at two different speeds (20 and 40 cm s$^{-1}$). Mice were placed on the treadmill and their paw prints were captured by a video camera with 147 frames per second) mounted inside the treadmill chassis beneath the transparent belt (DigiGait Imaging system, Mouse Specifics Inc.). Each mouse, running at two different speeds, was recorded for a minimum of four consecutive strides. After processing all video frames, gait signals for each paw were filtered and 39 gait features were calculated to quantify the paw pattern alternation in mice. For each gait feature (variable)'s dataset, normality and equality of variance were assessed. Variables conforming to a normal distribution were analysed with unpaired Student's *t*-tests, and those from non-normally distributed datasets were analysed with the Mann–Whitney test. A *P* value less than 0.05 was considered statistically significant (IBM SPSS Statistics v.26.0). The Cohen's effect size quantified the size of the mean difference between two groups by comparing it to the variability of the dataset. A Cohen's value below 0.2 indicated a weak effect, a value between 0.2 and 1 indicated a moderate effect and a value greater than 1 signified a strong effect[52].

**Swimming.** The swimming ability of mice was evaluated using a transparent Plexiglas chamber (90 × 14.5 × 30 cm) filled with tap water, maintained at a constant temperature of 23 °C. Platforms were positioned at both ends of the chamber to allow the animals to rest before and after each trial. A single run spanned a unidirectional course from the start point to the endpoint, roughly 60 cm in length. A run was deemed complete if the animal moved from one platform to the other in less than 5 s. Runs not completed within this timeframe were stopped and recorded as incomplete. Each animal completed a minimum of three valid runs, with a 5 min rest period between runs. After testing, the animals were manually dried with tissue, then placed in a single cage with a heating pad at 37.2 °C for 20 min to ensure complete drying. The animals' movements were recorded at 200 FPS with two cameras (model ACA640-90UC, Basler) located outside the chamber, providing lateral and ventral views.

**Overground walking.** Locomotor performance was tested in a transparent corridor (120 cm long and 10 cm wide) with 20 cm high lateral walls. Animal movements were captured at 200 FPS using two cameras (model ACA640-90UC, Basler) to obtain ventral and lateral views, as previously demonstrated[53]. Each animal traversed the corridor from one end to the other, constituting a run. Each animal was required to complete a minimum of three runs without stops. To prevent fatigue and maintain consistency, particularly in speed, the same experimenter handled all animals throughout the testing process.

**Ladder.** The evaluation of skilled walking in mice was conducted using a horizontal ladder, 120 cm long and 10 cm wide, with evenly spaced rungs set 2 cm apart. This setup challenged the mice to execute precise and coordinated movements. The animals' movements were recorded from ventral and side views by two cameras. Each animal repeated the ladder walk three times, totalling a maximum of 180 steps.

**Locomotor analysis.** The swimming and walking abilities were assessed through video recordings, subsequently tracked using marker-less pose estimation technology based on deep-learning and convolutional neural networks[54]. This process involved identifying major key points on the mouse's anatomy, specifically two on the head, five on the trunk and four on the tail. Furthermore, kinematics of the limbs was detailed using four key points for each forelimb (shoulder, elbow, wrist, paw) and five for each hindlimb (hip, knee, ankle, paw, toes). These points on the limbs formed interconnected chains of segments. From these key points, kinematic analysis of the hindlimb and forelimb generated a comprehensive set of 37 gait parameters using custom MATLAB scripts, which parsed the pose estimation data. To discern differences across experimental conditions and identify relevant parameters explaining these differences, a multifactorial analysis was used. For each experimental dataset, PCA commenced by constructing the covariance matrix $X$ from all gait parameters, which were normalized to avoid bias towards parameters with larger values by subtracting their mean and dividing by the standard deviation. The eigenvectors and eigenvalues of $X$ were then calculated, representing the principal components and the variance each component explained, respectively. The principal components were sorted by decreasing order of variance, with the first few typically capturing the most substantial portion of the variance in the dataset. The significant principal components served as the dependent variables in subsequent analyses to assess differences between experimental conditions, using analysis of variance (ANOVA) and post hoc tests where appropriate.

### DRG neuronal nuclei sorting for single-nucleus RNA-seq

L4-6 DRG from control or macrophage-depleted mice were euthanized, dissected and snap frozen in liquid nitrogen on day 4. On the day of sorting, DRG were placed into homogenization buffer (0.25 M sucrose, 25 mM KCl, 5 mM MgCl$_2$, 20 mM tricine-KOH pH 7.8, 5 µg ml$^{-1}$ actinomycin, 1% BSA, 0.15 mM spermine, 0.5 mM spermidine, EDTA-free protease inhibitor, phosphatase inhibitor, RNase inhibitor). DRGs were homogenized by pelleting with a plastic pestle for 15 s. Trition-X-100 (Sigma) was added to the homogenization mixture to reach the final concentration of 0.1%. Another 15 s stroke was added to further homogenize the DRGs. Finally, DRG homogenization suspension was filtered through 70 µm followed by 40 µm cell strainers and incubated with NeuN-Alexa Fluor488 antibody (MAB377X Clone A60, 1:100) for 1 h in cold room with gentle rotation. After washing with washing buffer (homogenization buffer + 0.1% Triton X-100 + 2% BSA) three times, DRG nuclei were resuspended in 2 ml of washing buffer and incubated with DAPI (D5942, Sigma) for 10 min. DAPI$^+$/NeuN$^+$ DRG neuronal nuclei were sorted by Aria III sorter and collected in washing buffer for 10X single-nucleus RNA-seq according to 10X Chromium Next GEM Single Cell 3′ Reagent Kits v.3.1 (Dual Index) protocol.

### Optogenetic stimulation of neurons (Pv$^+$) or macrophages (CX3CR1$^+$)

To specifically activate ChR2 in either proprioceptive neurons or macrophages, we used mice expressing Channel rhodopsin (Ai32, Jackson Laboratories) under the parvalbumin (Pv$^{Cre}$, Jackson Laboratories) or the macrophage (Cx3cr1$^{Cre}$, Jackson Laboratories) promoters. Before surgery, the animals were anaesthetized with isoflurane (induction 4% and maintenance 2%) and an eye ointment was applied to prevent eyes dehydration. The animal was placed on a heating pad (Stoelting Warming Systems) and the anaesthesia's depth was verified through the absence of reflexes and tail pinch responses, ensuring the animals were in a controlled and unresponsive state throughout the whole procedure. A sizable incision was made to expose the sciatic nerve at the thigh level (for stimulation of sensory neurons) and to reveal the gastrocnemius and tibialis muscles (for MSMP optical stimulation). On exposure, an optical fibre (0.63 NA, two for muscle stimulation) was positioned roughly 1 mm away from the tissue surface of the sciatic nerve (or muscles) on left side of the mouse body. The right-side sciatic nerve (or muscles) was exposed without light treatment for contralateral control. This configuration enabled optical stimulation of the sciatic nerve (or muscle) for 1 h, using an optical fibre (core diameter of 200 µm) with 0.63 NA, Prizmatix connected to a dual LED optical source (Prizmatix) triggered by a pulse stimulator (Master 9). Two fibres were used for each muscle in the case of MSMP stimulation. Stimulation at a wavelength of 450–460 nm, with a pulse duration of 2 ms at a frequency of 1 Hz with a sufficient power (maximum power greater than 80 mW) to result in myogenic activity. In all case, the hindlimbs were securely immobilized to minimize movement and the exposed tissue was continually infused with warm saline to maintain its functional viability and avoid dehydration.

### MSMP sorting for single-cell RNA-seq

After 1 h of activation of Pv neurons, a cardiac perfusion was performed using 20 ml of cold DPBS. MSMP were dissected and FACS-sorted from both light-treated and control side of Pv-ChR2 mice with the previously described gating strategy. Sorted MSMP were collected in cold DPBS and processed for single-cell RNA-seq according to 10X Chromium Next GEM Single Cell 3′ Reagent Kits v.3.1 (Dual Index) protocol.

### Single-cell or single-nucleus RNA-seq data processing

The library of MSMP or DRG was sequenced by Illumina NovaSeq 6000 platform. Samples were demultiplexed into FASTQ reads and then aligned to the mouse GRCm39 genome reference. Sample demultiplexing, sequence alignment, barcode processing and single-cell 3′ unique molecular identifier counting were performed by using Cell Ranger Software Suite (v.7.1.0). Seurat objects for each sample were created by converting the count matrix and performing initial quality control. Cells were filtered on the basis of thresholds for read counts (greater than 500), gene features (greater than 250) and mitochondrial content (less than 15%). Genes expressed in fewer than ten cells were filtered out. Doublets were removed from the dataset by DoubletFinder package (doublet formation rate was set as 7.5%). The filtered datasets were normalized by using SCTransform. The count matrices from different samples were merged into a single Seurat object. The analysis then progressed to perform data integration using Seurat (v.4) integration function. Anchors were identified for dataset integration, and hierarchical clustering is used to perform dimensionality reduction. PCA and uniform manifold approximation and projection (UMAP) were used for dimensionality reduction and visualization. After PCA, 40 principal components were selected for cell clustering. Clusters were defined on the basis of a chosen resolution, and the resulting cell clusters were visualized through UMAP plots. Seurat FindAllMarkers function enabled the identification of marker genes associated with specific cell clusters. For DRG neurons, we annotated proprioceptors by using *Pth1r*, *Esrrg*, *Pvalb*, *Etv1*. C-LTMRs were annotated by using *Tafa4*, *Cacna1h*, *Zfp521*. Aδ-LTMR were annotated by using *Ntrk2*, *Cacna1h*, *Cadps2*, *Kcnq3*. Non-peptidergic C-fibre nociceptors (NPs) subsets were annotated using *Trpc3*, *Gfra2*, *Plxnc1* (NP1); *Osmr*, *Il31ra*, *Nppb* (NP2); *Tnr*, *Gfra1*, *Mrgpra3* (NP3). C-fibre peptidergic nociceptors (PEPs) subsets were annotated using *Tafa1* (PEP1); *Gfra3*, *Kcnmb2* (PEP2); *Tafa1*, *Kcnt2* (PEP3); *Ryr2*, *Ryr3*, *Cntn5*, *Gm10754* (PEP4). Cold thermoreceptor subsets were annotated using *Trpm8*, *Foxp2*.

Genes that were differentially expressed in each cell cluster between two conditions were identified by using the Seurat FindMarkers function with the MAST test. Differentially expressed genes between each

pair of clusters were determined with a false discovery rate (FDR) less than 0.05 and performed GO analysis. GO pathways were categorized by the DAVID Bioinformatics Resource with a $P$ value less than 0.05. GO categories highlighted in red and blue were upregulated or downregulated pathways, respectively.

## Electromyography and dorsal root recording with electric stimulation of the 'H reflex' or stimulation of stretch reflexes

The animals were deeply anaesthetized with isoflurane as previously mentioned (for optical stimulation of sensory neurons) and a shave at the level of the spinal columns and the two hindlimbs. Iodine and 70% ethanol were then applied to the shaved skin. After 10 min, another incision of the skin exposed specific muscles to allow bipolar recordings and monitor the activity of the tibialis, the gastrocnemius, and the semitendinosus muscles. Two dual core Teflon-coated platinum–iridium wires with a diameter of 125 µm (WPI, code number PTT0110) were threaded through a 30.5 gauge hypodermic needles. The Teflon cover was removed from about 0.5 mm of the tips and the two electrodes were then inserted 1–1.5 mm apart in the belly of each muscle. Successively, another skin incision at the level of the spinal column followed by a laminectomy exposed the thoraco-lumbar segments of the spinal cord with the respective dorsal roots. The dura mater was opened, and the dorsal roots were gently moved apart for easy identification of the belonging segment. Bleeding due to the laminectomy was blocked. A dorsal root between L2 and L6 was chosen and then recorded using a suction electrode. Electric activation of sensory neurons was obtained by stimulating the sciatic nerve using a bipolar stimulating electrode with a hook shape. Electric stimulation (50 ms, 50–150 mA, low threshold stimulation) elicit primarily the typical muscle activity obtained with activation of the proprioceptive afferents, so electromyography recordings that correspond in latencies and threshold to the 'M wave' and to the 'H reflex' in mice[55]. The M wave represents the direct stimulation of motor axons whereas the H reflex is generated by the monosynaptic activation of motor neurons by Ia afferent fibres in the muscles. The amplitude of the H reflex is proportional to the number of activated motor neurons. Electrical signals were acquired using a band pass filter (100–1,000 kHz) and all data were stored for off-line analysis. An NMDA blocker (AP-5, 5 mM) or AMPA blocker (CNQX, 5 mM) was injected into the muscle. Wash out of the drugs and restoration of responses were obtained by injecting saline.

To test the inactivation of MSMP (CX3CR1[+]) on the stretch reflexes, mice were anaesthetized and the hindlimb at the knee joint were immobilized to a horizontal plate. Then, the Achilles tendon of the gastrocnemius was cut and secured to a small diameter bar (diameter of 3 mm) with a suture wire (Mersilk, Ethicon). The bar was controlled by a programmable micromanipulator to (MCL3, WPI) executing movements to elongate the muscle from 0.5 to 4 mm s$^{-1}$ for a duration of 2 s. The elongations were randomly executed. The inhibitory opsin was activated by yellow laser light (590 nm) during the time of the stretch.

## MSMP glutamate assay

MSMP were isolated from mouse hindlimb skeletal muscles by FACS. Sorted macrophages were directly collected in the culture media (B27 (1:50), glucose (17.5 mM), glutamine (31.25 µM) and penicillin–streptomycin (1:100) in DMEM (no glucose, no glutamine, no phenol red)). After washing once with culture media, MSMP were resuspended in culture media with 20 ng ml$^{-1}$ M-CSF1 and cultured in a sterile 96-well plate (20,000 cells per well). On the next day, the original culture medium was collected and changed into the medium with or without glutamine (50 mM). Glutaminase inhibitor CB-839 (2 µM, Selleckchem) was added to the cultured MSMP 1 h before adding glutamine. Glutamine transporter inhibitors α-(Methylamino) isobutyric acid (MeAIB, 20 mM, Sigma) and V9302 (50 µM, Selleckchem), or calcium

chelators BAPTA-AM (10 µM, Thermo Fisher Scientific) and EGTA (10 mM, Sigma) were dissolved in glutamine-free culture medium and added to cultured MSMP for 10 min of preincubation, respectively. After washing with glutamine-free culture medium twice, MSMP were treated with 50 mM glutamine (dissolved in culture medium) together with or without dimethylsulfoxide (DMSO), MeAIB and V9302, or BAPTA-AM and EGTA. Control cells were treated with glutamine-free medium with or without DMSO. After 2 or 8 h of incubation, the MSMP-conditioning medium was collected and centrifuged with high speed to remove debris. Glutamate concentration was measured with the Glutamate-Glo Assay Kit (Promega) according to the manufacturer's instructions.

## MSMP calcium imaging

MSMP were isolated from mouse hindlimb skeletal muscles by FACS. Sorted macrophages were directly collected in the culture media (B27 (1:50), glucose (17.5 mM), glutamine (31.25 µM) and penicillin–streptomycin (1:100) in DMEM (no glucose, no glutamine, no phenol red)). After washing once with culture media, MSMP were resuspended in culture media with 20 ng ml$^{-1}$ M-CSF1 and cultured in a sterile 96-well plate (20,000 cells per well). On the next day, MSMP were washed with glutamine-free culture medium and incubate with Fluo-4-AM (2 µM, Thermo Fisher Scientific) for 30 min at 37 °C. Cells were washed three times with glutamine-free culture medium and preincubated with glutamine-free medium with or without DMSO, MeAIB (20 mM, Sigma) and V9302 (50 µM, Selleckchem), or BAPTA-AM (10 µM, Thermo Fisher Scientific) and EGTA (10 mM, Sigma) for 10 min. After washing with glutamine-free culture medium twice, MSMP were treated with 50 mM glutamine (dissolved in culture medium) together with or without DMSO, MeAIB and V9302, or BAPTA-AM and EGTA for 5 min. MSMP were immediately washed and fixed by 4% PFA for 30 min at room temperature and processed for immunostaining. Cells were blocked with 5% BSA for 1 h, incubated with F4/80 primary antibody and secondary antibody for 2 and 1 h, respectively.

## MSMP real-time qPCR

MSMP were isolated from mouse hindlimb skeletal muscles by FACS after 30 min of freely swimming or resting physiological conditions (control). FACS-sorted macrophages were collected in cold DPBS with RNase inhibitor and directly processed for cell lysis, reverse transcription and preamplification using a SuperScript IV Single Cell/Low Input cDNA PreAmp Kit (Invitrogen) according to the manufacturer's guidelines. Real-time quantitative PCR (qPCR) was performed on the Quant Studio 3 Flex System with 10 ng cDNA per reaction using KAPA SYBR FAST qPCR Master Mix kit (Sigma) according to the manufacturer's instructions. A list of the qPCR primers purchased and used in this study is provided in Supplementary Table 7. Expression values for each tested gene relative to a glyceraldehyde 3-phosphate dehydrogenase endogenous control were calculated using the ΔΔCt method.

## MSMP patch clamp

MSMP were isolated from mouse hindlimb skeletal muscles by FACS. Target cells were collected into culture media (1% sodium pyruvate, 1% penicillin–streptomycin, 10% FBS and 20 ng ml$^{-1}$ M-CSF1 in RPMI1640 media). After washing once with culture media, cells were seeded onto 8 mm diameter precoated 100 µg ml$^{-1}$ poly-D-lysine coverslips in sterile 48-well plates at a final density of 30,000 cells per well. Non-adherent cells were removed by changing medium before functional experiments. Patch-clamp experiments were performed not before 72–96 h postisolation. For this, cultures were positioned in a recording chamber on the stage of an upright BX51WI Olympus microscope, equipped with an ×40 water immersion objective and epifluorescence illumination. The chamber was continuously perfused with artificial cerebral spinal fluid (aCSF), composed of 125 mM NaCl, 2.5 mM KCl, 25 mM NaHCO$_3$, 2.5 mM CaCl$_2$, 1.3 mM MgCl$_2$, 1.25 mM NaH$_2$PO$_4$ and 25 mM D-glucose.

The aCSF was saturated with carbogen (95% $O_2$, 5% $CO_2$) and maintained at a flow rate of 1–2 ml min$^{-1}$.

Patch electrodes were fabricated from borosilicate glass using a P-1000 pipette puller (Sutter Instruments Co.) and showed resistances of 4–7 MΩ. Macrophages expressing either ChR2-YFP$^+$ or NpHR3-YFP$^+$ were identified under epifluorescence and subjected to whole-cell patch-clamp recordings in current-clamp mode. The intracellular recording solution contained 108 mM K-gluconate, 8 mM Na-gluconate, 2 mM MgCl$_2$, 8 mM KCl, 1 mM EGTA, 4 mM K2-ATP, 0.3 mM Na-GTP and 10 mM HEPES; pH was adjusted to 7.2 with KOH and osmolarity to roughly 285 mOsm. All recordings were conducted at room temperature.

After achieving whole-cell configuration, cells were allowed to stabilize for at least 5 min before measuring the resting membrane potential. Optogenetic activation of ChR2 and NpHR3 was performed using blue and red LED light, respectively, delivered through the recording objective by means of an X-Cite 120 LED boost lamp (Excelitas). Electrical signals were amplified and filtered using a MultiClamp 700B amplifier, then digitized at a 10 kHz sampling rate with a Digidata 1440A system (Molecular Devices). Cell capacitance and access resistance were monitored continuously; recordings were deemed acceptable if these parameters varied by less than 20%. Data analysis was performed off-line using the pClamp (v.11.2.1) software (Molecular Devices).

## Statistics and reproducibility

Statistical analysis was performed using GraphPad Prism v.8 software. For comparisons between two groups with normally distributed data and equal variances, we used a two-tailed unpaired Student's $t$-test. When variances were unequal, we used Welch's $t$-test or Wilcoxon. For non-normally distributed data comparing two groups, we applied the Mann–Whitney test. One-way ANOVA was used to compare more than two groups with normally distributed data, followed by appropriate post hoc tests to determine group-specific differences. In cases in which some groups of data were normally distributed and others were not, or when some groups showed unequal variances, we consistently applied the most conservative statistical test. Specifically, when assumptions of normality or homogeneity of variances were violated, we used non-parametric tests or tests designed to handle unequal variances (for example, Welch's $t$-test or Mann–Whitney test) to ensure robust and reliable results. A significance threshold of $P < 0.05$ was used, with statistical significance indicated as $*P < 0.05$, $**P < 0.01$, $***P < 0.001$ and $****P < 0.0001$. Non-significant results are marked as NS.

Sample size was determined on the basis of similar, previously established experimental designs or calculated using the Animal Experimentation Ethics Committee's power calculator (http://www.lasec.cuhk.edu.hk/sample-size-calculation.html), with parameters set to detect a difference of 1.5, with 80–90% power and a 5% significance level.

Each dataset was replicated and examined over two to three independent experiments.

## Replication

All attempts at replication were successful.

## Blinding

All analyses were conducted with blinding to the experimental groups. Data were obtained from two to three independent experiments.

## Reporting summary

Further information on research design is available in the Nature Portfolio Reporting Summary linked to this article.

## Data availability

All RNA-seq data are available from the NCBI GEO database under accession numbers GSE244351 and GSE246400. Source data are provided with this paper.

## Code availability

Code is available at Zenodo (https://doi.org/10.5281/zenodo.13902719)[56].

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

**Acknowledgements** This work was supported by the Adelson Medical Research Foundation (S.D.G.); The Rosetrees Trust (S.D.G.); the Imperial PhD Presidential Scholarship (J.C.) and the University of Copenhagen (C.B.). The research was supported by the National Institute for Health Research (NIHR) Imperial Biomedical Research Centre (S.D.G.). Illustrations in Figs. 1k, 2a, 4d and 5a and Extended Data Figs. 1a, 3a and 4a,c,e were created with BioRender. The views expressed are those of the author(s) and not necessarily those of the NHS, the NIHR or the Department of Health. We thank the FACS facility at Imperial and at University of Copenhagen.

**Author contributions** Y. Yan designed and performed experiments, performed data analysis and wrote the manuscript. N.A., L.Z., L.X., I.L.V., C.D.G.M., G.K., J.C., Y. Yang and J.F.B. performed experiments. F.M. performed data analysis. I.P., A.M.J.B. and C.L.C. provided experimental advice. G.P. and S.S. provided human biopsies. C.B. designed and performed experiments, performed data analysis, provided funding and wrote the manuscript. S.D.G. designed experiments, provided funding and wrote the manuscript.

**Competing interests** The authors declare no competing interests.

**Additional information**
**Correspondence and requests for materials** should be addressed to Carmelo Bellardita or Simone Di Giovanni.

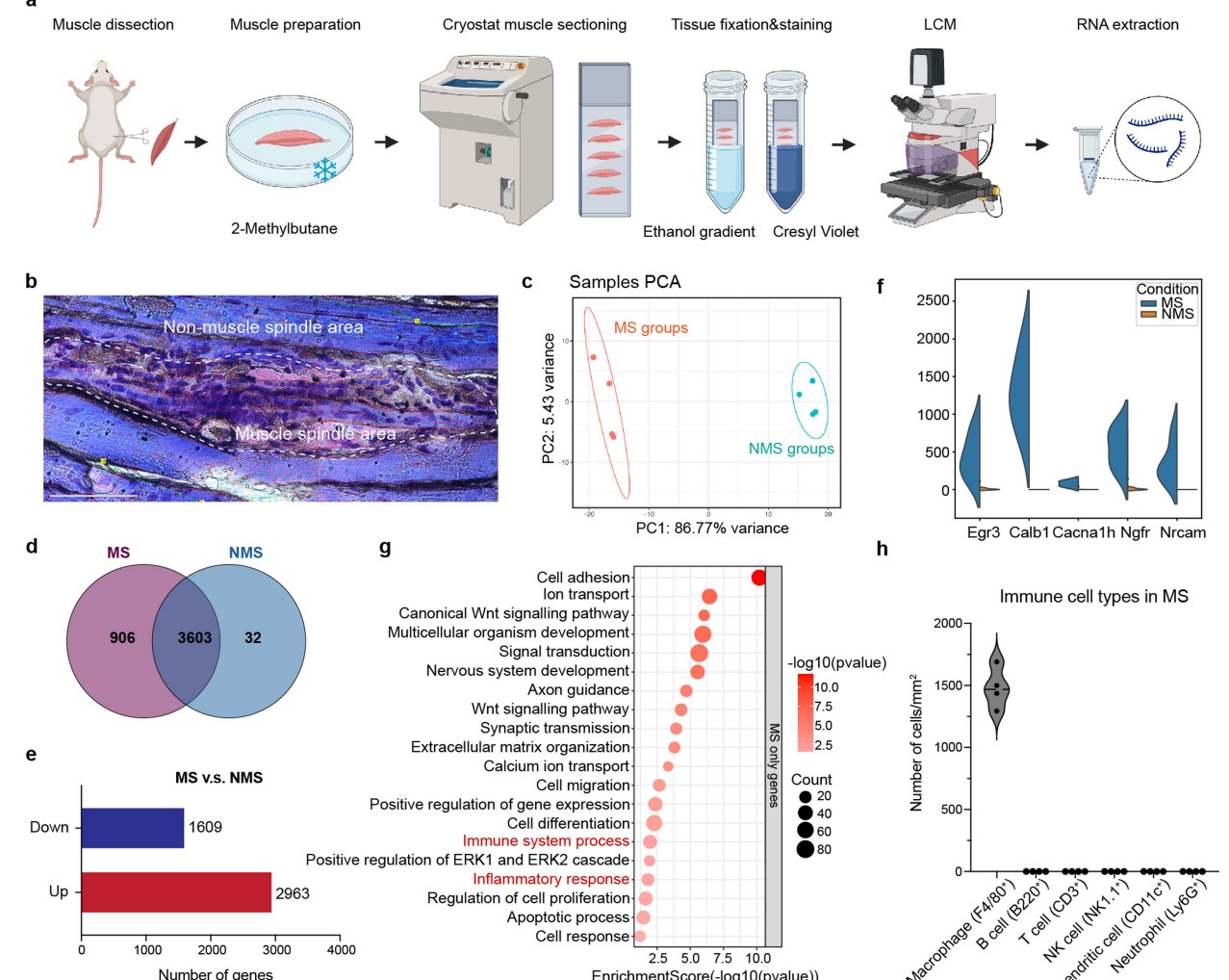

**Extended Data Fig. 1 | Molecular signatures of mouse muscle spindles (MS).**
**a**, Experimental flow chart of laser-capture microdissection (LCM) to dissect MS and non-MS (NMS) tissues from mouse hindlimb skeletal muscles.
**b**, Micrograph of laser-captured MS and NMS areas with cresyl violet staining. Scale bar = 50 μm. **c**, Principal component analysis (PCA) of MS and NMS RNA-seq datasets. **d**, Normalized genes expressed by MS and NMS areas (FDR < 0.1, normalized count >10). **e**, Summary of differential expression analysis of up- and downregulated genes in MS versus NMS (FDR < 0.1, fold change > 1.5). **f**, Exclusive expression of MS marker genes in MS areas compared with NMS areas. **g**, Functional analysis for genes only expressed by MS compared with NMS areas (with FDR < 0.1, normalized count >10). Shown are significantly upregulated GO pathways categorized by the Database for Annotation, Visualization, and Integrated Discovery (DAVID) Bioinformatics Resource with p < 0.05. **h**, Quantification of different types of immune cells in MS areas (n = 4 biological independent mice). Illustrations in **a** created using BioRender (https://biorender.com).

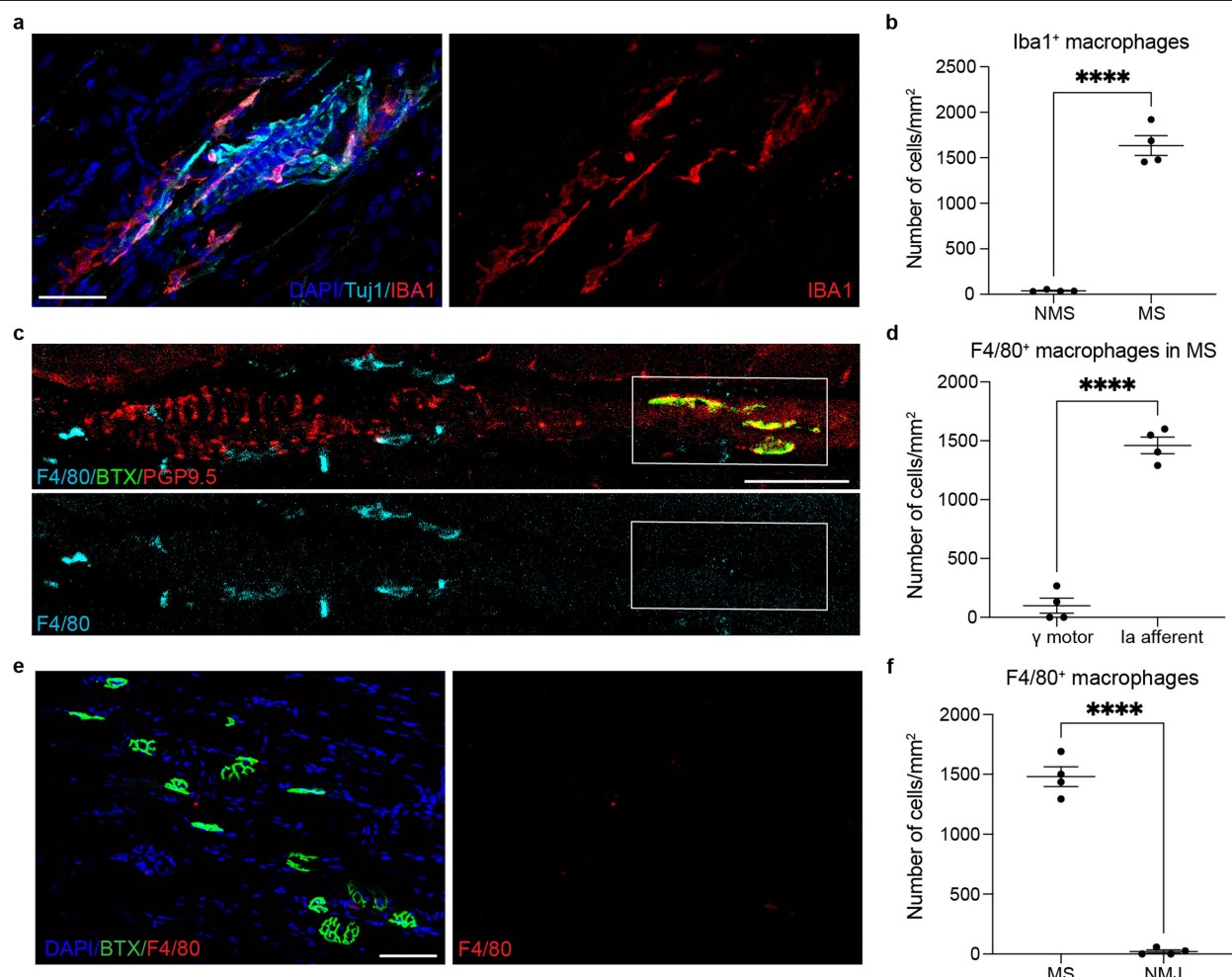

**Extended Data Fig. 2 | Localization of macrophages in muscle spindles and neuromuscular junctions. a**, Immunostaining for Iba1[+] macrophages in mouse hindlimb skeletal muscle. Muscle spindles (MS) are localized by the annulospiral axons labelled with Tuj1 and DAPI counterstaining. **b**, Quantification of the number of Iba1[+] macrophages in MS and non-MS (NMS) areas (n = 4 biological independent mice; two-tailed unpaired Student's t-test; mean ± s.e.m; ****p < 0.0001). Scale bar = 50 μm. **c**, Immunostaining showing F4/80[+] macrophages are found in the MS in proximity to sensory afferents (labelled by PGP9.5 and DAPI counterstaining) compared with **γ**-motor neurons (labelled by bungarotoxin; BTX). Scale bar = 100 μm. **d**, Quantification of the number of F4/80[+] macrophage close to MS sensory afferents and γ-motor neurons (n = 4 biological independent mice; two-tailed unpaired Student's t-test; mean ± s.e.m; ****p < 0.0001). **e**, Immunostaining indicating the lack of F4/80[+] macrophages around neuromuscular junctions (NMJ), labelled by bungarotoxin (BTX). **f**, Quantification of macrophage distribution in MS and NMJ (n = 4 biological independent mice; two-tailed unpaired Student's t-test; mean ± s.e.m; ****p < 0.0001). Scale bar = 50 μm.

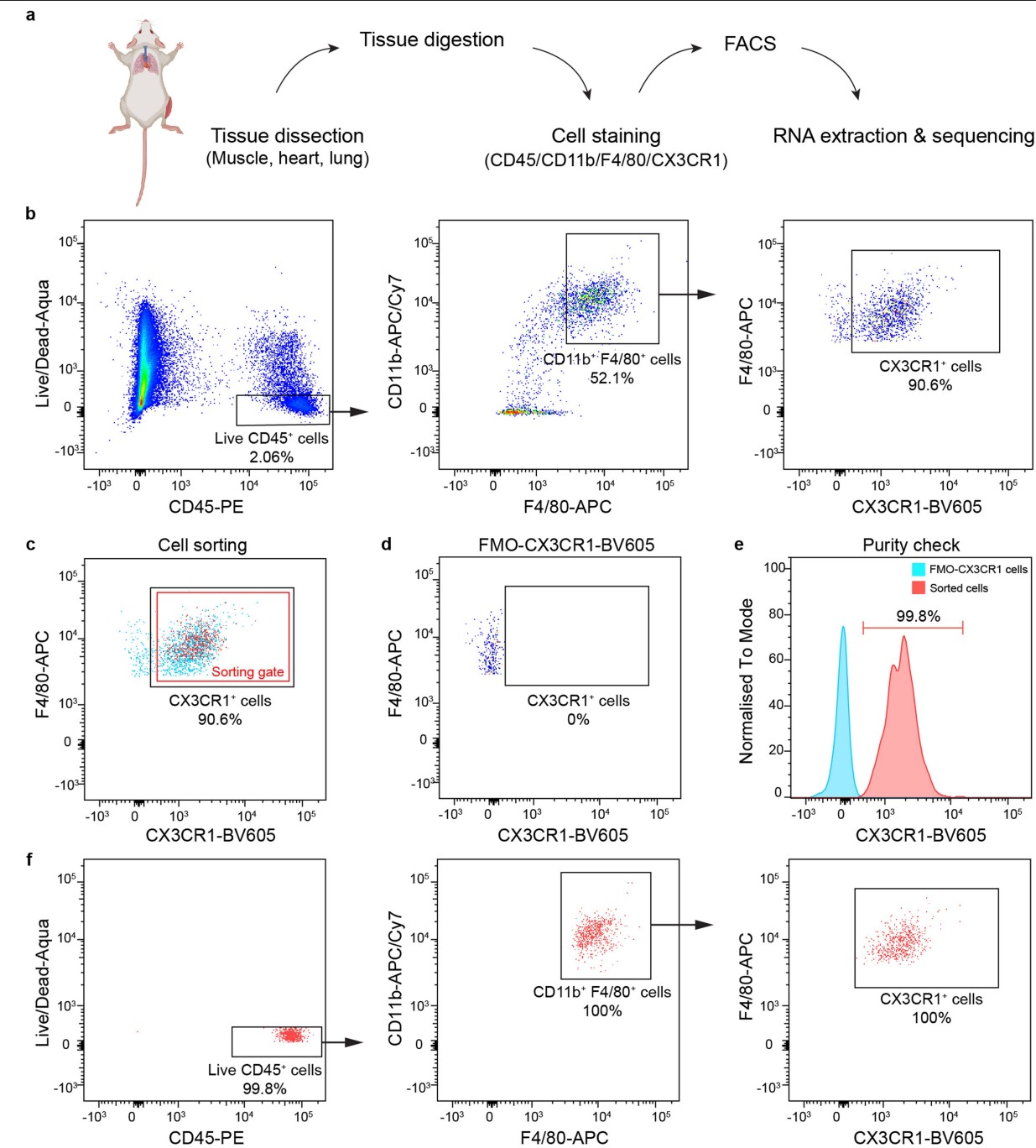

**Extended Data Fig. 3 | Macrophages isolated from various tissues via fluorescence-activated cell sorting. a**, Experimental flow chart of RNAseq of macrophages from dissected muscle, heart and lung, and isolated by fluorescence-activated cell sorting (FACS). **b**, FACS gating to isolate live CD45/CD11b/F4/80/CX3CR1+ macrophages. **c**, Cell sorting gate. **d**, Fluorescence Minus One (FMO)-CX3CR1 for CX3CR1 gating. **e**,**f**, Histogram and FACS showing the purity of sorted live CD45/CD11b/F4/80/CX3CR1+ macrophages. Illustrations in **a** created using BioRender (https://biorender.com).

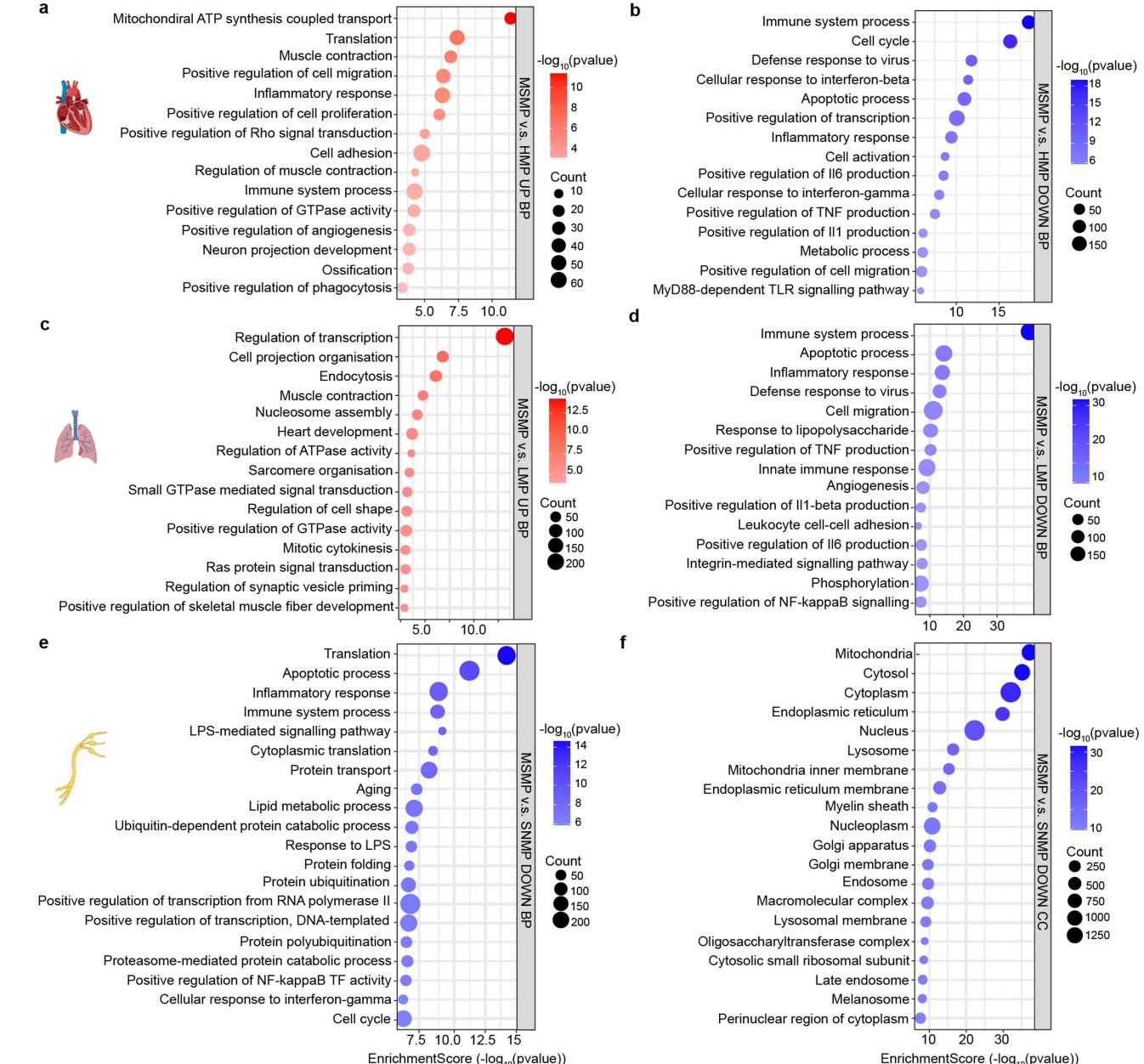

**Extended Data Fig. 4 | Differential expression analysis of genes higher and lower expressed in muscle spindle macrophages (MSMP) compared with heart, lung and sciatic nerve macrophages. a**,**b**, Functional analysis of biological process (BP) of upregulated (**a**) and downregulated (**b**) genes in MSMP versus heart macrophage (HMP) (FDR < 0.05, fold change > 1.5). **c**,**d**, Functional analysis of BP of upregulated (**c**) and downregulated (**d**) MSMP versus lung macrophage (LMP) (FDR < 0.05, fold change > 1.5). **e**, Functional analysis of BP of downregulated genes in MSMP versus sciatic nerve

macrophages (SNMP) (FDR < 0.05, fold change > 1.5). **f**, Functional analysis of cellular component (CC) of downregulated genes in MSMP versus SNMP (FDR < 0.05, fold change > 1.5). Shown are significantly upregulated and downregulated GO pathways categorised by the Database for Annotation, Visualization, and Integrated Discovery (DAVID) Bioinformatics Resource with p < 0.05. The statistically significant GO categories highlighted in red and blue are upregulated and downregulated pathways, respectively. Illustrations in **a**,**c**,**e** created using BioRender (https://biorender.com).

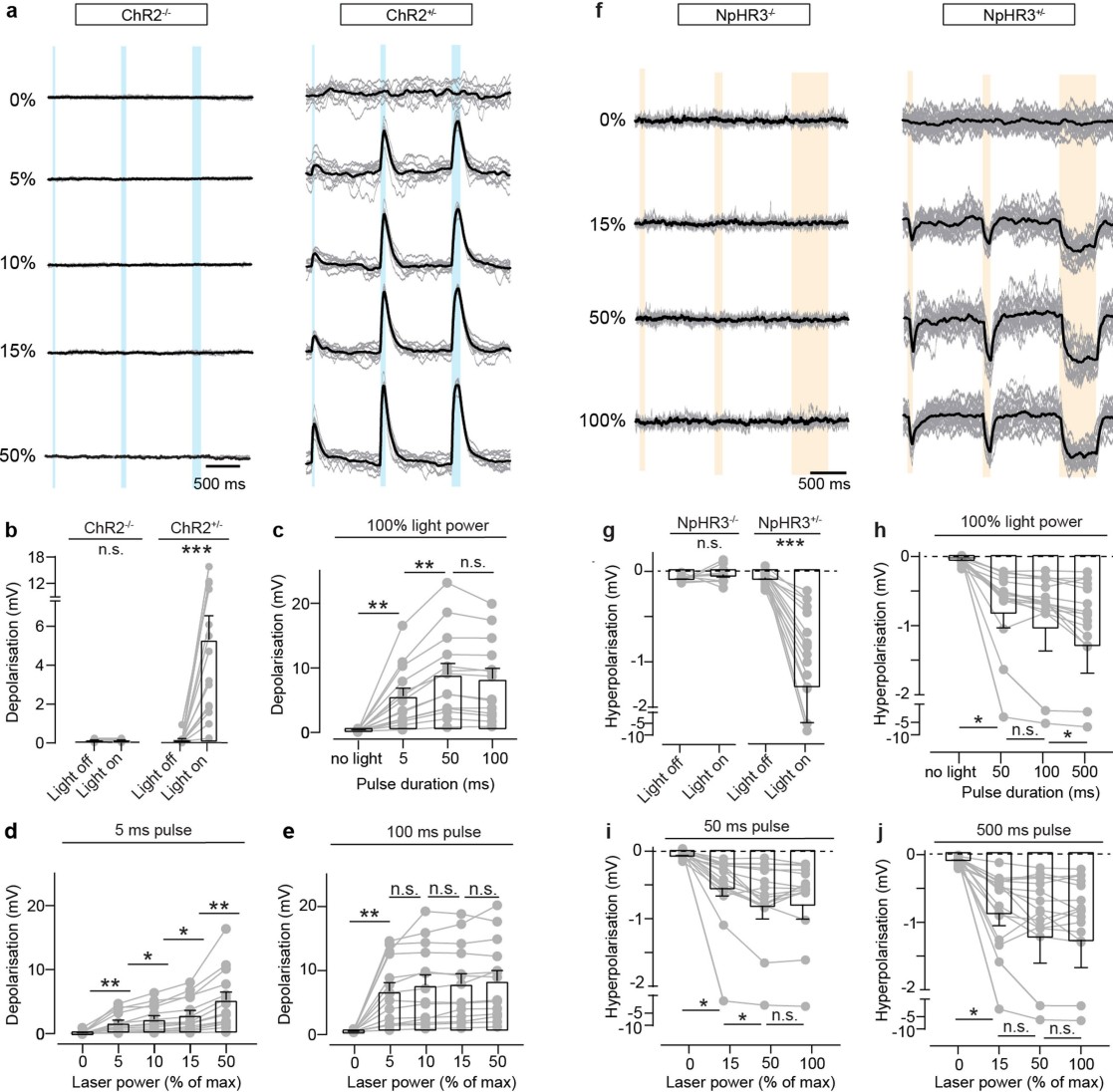

**Extended Data Fig. 5 | Electrophysiological characterisation of MSMP expressing ChR2 and NpHR3. a**,**f**, Shown are membrane potentials of cultured negative and positive cells for excitatory (**a**, CX3CR1^cre::ChR2^YFP, ChR2^− and ChR2^+) or inhibitory (**f**, CX3CR1^cre::NpHR3^YFP, NpHR3^− and NpHR3^+) opsins with the relative responses upon light stimulation (473 nm for ChR2 and 593 nm for NpHR3) at various light powers (0%, 5%, 10%, 15%, 50%, 100%) and various pulse durations (5, 50, 100, 500 ms). **b**,**g**, Quantification of depolarization in ChR2^− and ChR2^+ macrophages (n = 14 independent cells per group, three independent experiments) and hyperpolarization in NpHR3^− and NpHR3^+ macrophages (n = 16 independent cells per group, three independent experiments) with and without light stimulation (mean ± s.e.m.). **c**,**h**, Response amplitude in ChR2^+ cells (**c**) and NpHR3^+ (**h**) at different pulse durations with 100% light power. **d**,**e**, Response amplitude (depolarization) in ChR2^+ cells at various laser powers for 5 ms (**d**) and 100 ms (**e**) pulse durations. **i**,**j**, Response amplitude (hyperpolarization) in NpHR3^+ cells at various laser powers for 50 ms (**i**) and 500 ms (**j**) pulse durations. Data are given as mean ± s.e.m.; paired Student's t-test (**b**,**g**), RM One-way ANOVA with Tukey's (**c**–**e**,**h**), with Sidak (**j**) multiple comparisons test were used. Significance levels: n.s. = not significant, *p < 0.05, **p < 0.01, ***p < 0.001.

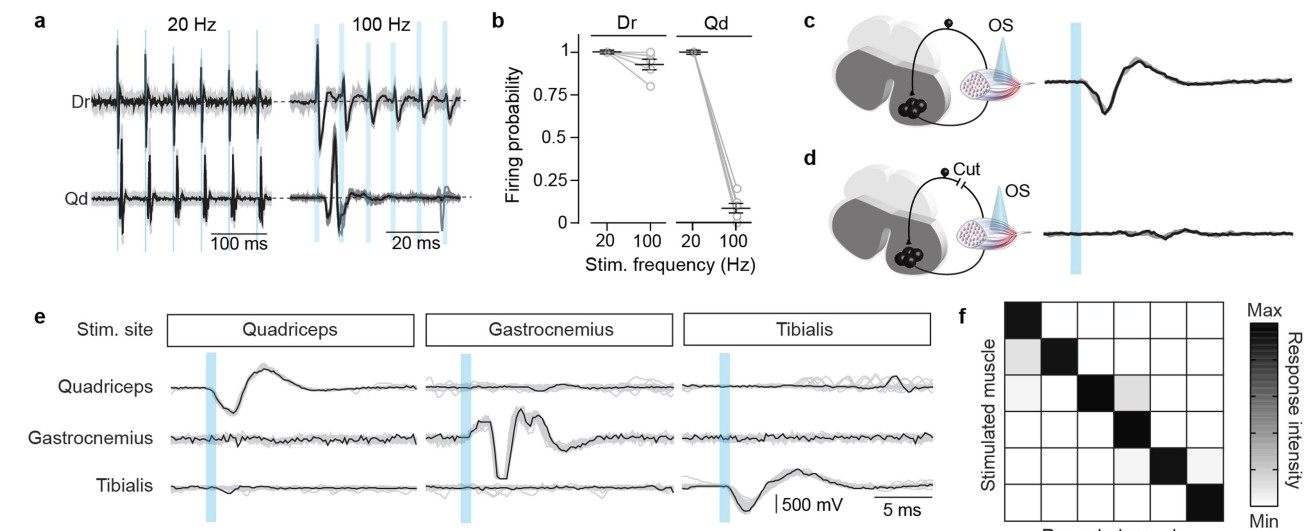

**Extended Data Fig. 6 | Optical stimulation of hindlimb muscle CX3CR1+ macrophages in CX3CR1^Cre^::ChR2^YFP^ animals modulates the reflex arc and muscle response. a,b,** Shown are traces (**a**) of sensory (Dr) and muscle responses (Qd) to optical stimulation (train of 15 pulses, 5 ms duration, 15% power) at low (20 HZ) and high frequency (100 Hz) in anaesthetized CX3CR1^Cre^::ChR2^YFP^ mice, and quantification of response probability (**b**, responses/number of pulses in the train, n = 5 biological independent mice). **c,d,** A short optical pulse (5 ms) in the gastrocnemius of anaesthetized CX3CR1^Cre^::ChR2^YFP^ mice evokes a muscle response (**c**) that is abolished when the homologues dorsal root is cut (**d**). **e,f,** Representative traces (**e**) and correlation matrix (**f**) showing the spatial specificity of MSMP modulation in CX3CR1^Cre^::ChR2^YFP^. Optical stimulation of specific muscles (stim. site) mice elicited the activation of the response only in that muscle (**e**), indicating the high spatial specificity of the MSMP modulation onto the neural activity and muscle contraction.

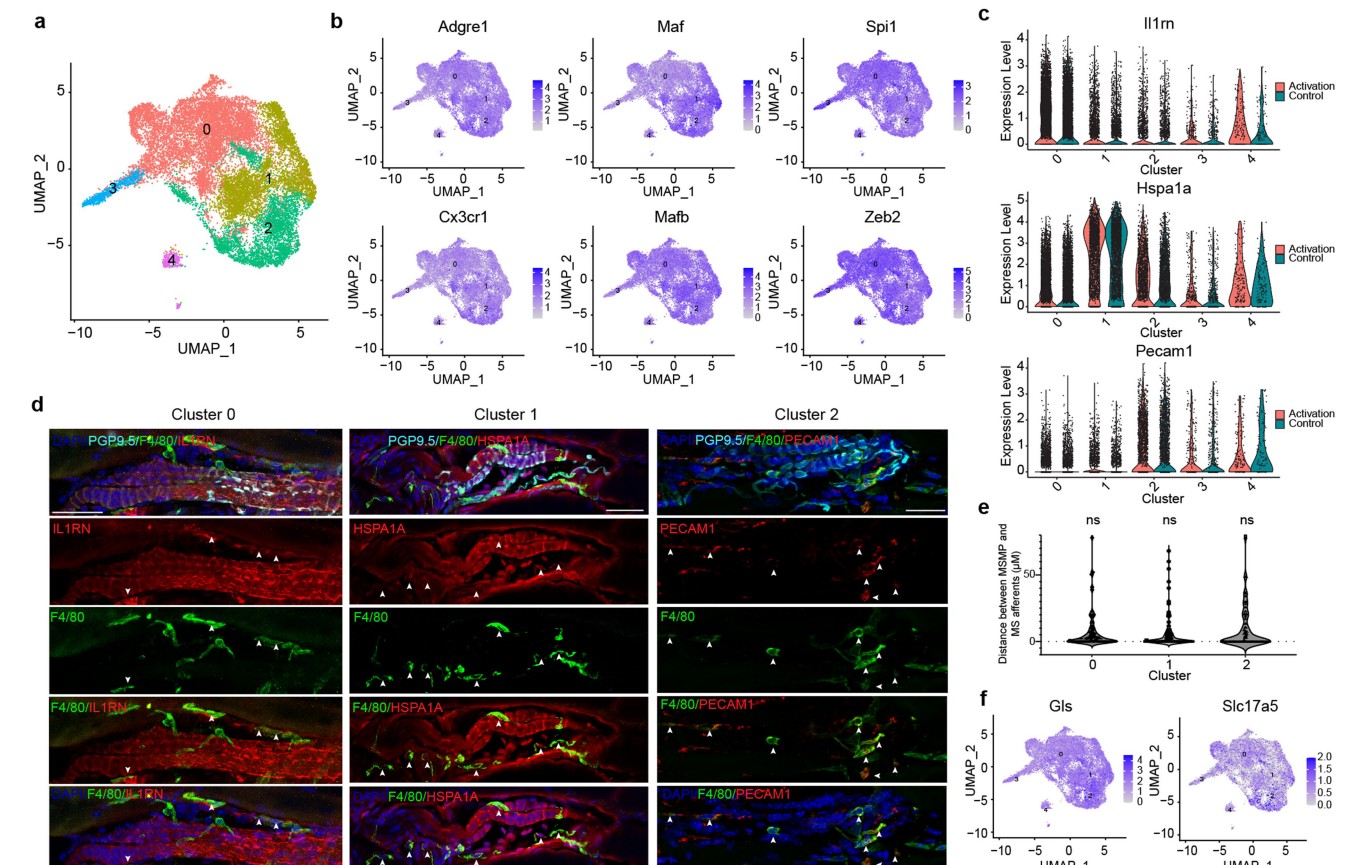

**Extended Data Fig. 7 | MSMP clusters express tissue resident markers and share similar sub-anatomical localization within MS. a**, Uniform manifold approximation and projection (UMAP) results show 5 clusters of MSMP from 18,538 F4/80/CX3CR1⁺ MSMP. **b**, Expression of MSMP markers and tissue-resident markers in MSMP. **c,d**, Violin plot (**c**) and representative immunostaining (**d**) for the expression of marker genes (*Il1rn*, *Hspa1a* and *Pecam1*) of cluster 0-2 MSMP (labelled by F4/80) in MS (labelled by PGP9.5).

Scale bar = 50 μm. **e**, Spatial localization (distance between MSMP and MS sensory fibre) of cluster 0-2 MSMP by using the sensory annular fibres as reference. Cluster 0, cluster 1 and cluster 2 MSMP are identified by the expression of Il1rn (n = 65 MSMP), Hspa1a (n = 69 MSMP) and Pecam1 (n = 69 MSMP), respectively (n = 3 biological independent mice; one-way ANOVA, Tukey's multiple comparisons test; ns = not significant). **f**, Expression of *Gls* (Glutaminase) and *Slc17a5* (Sialin) by MSMP.

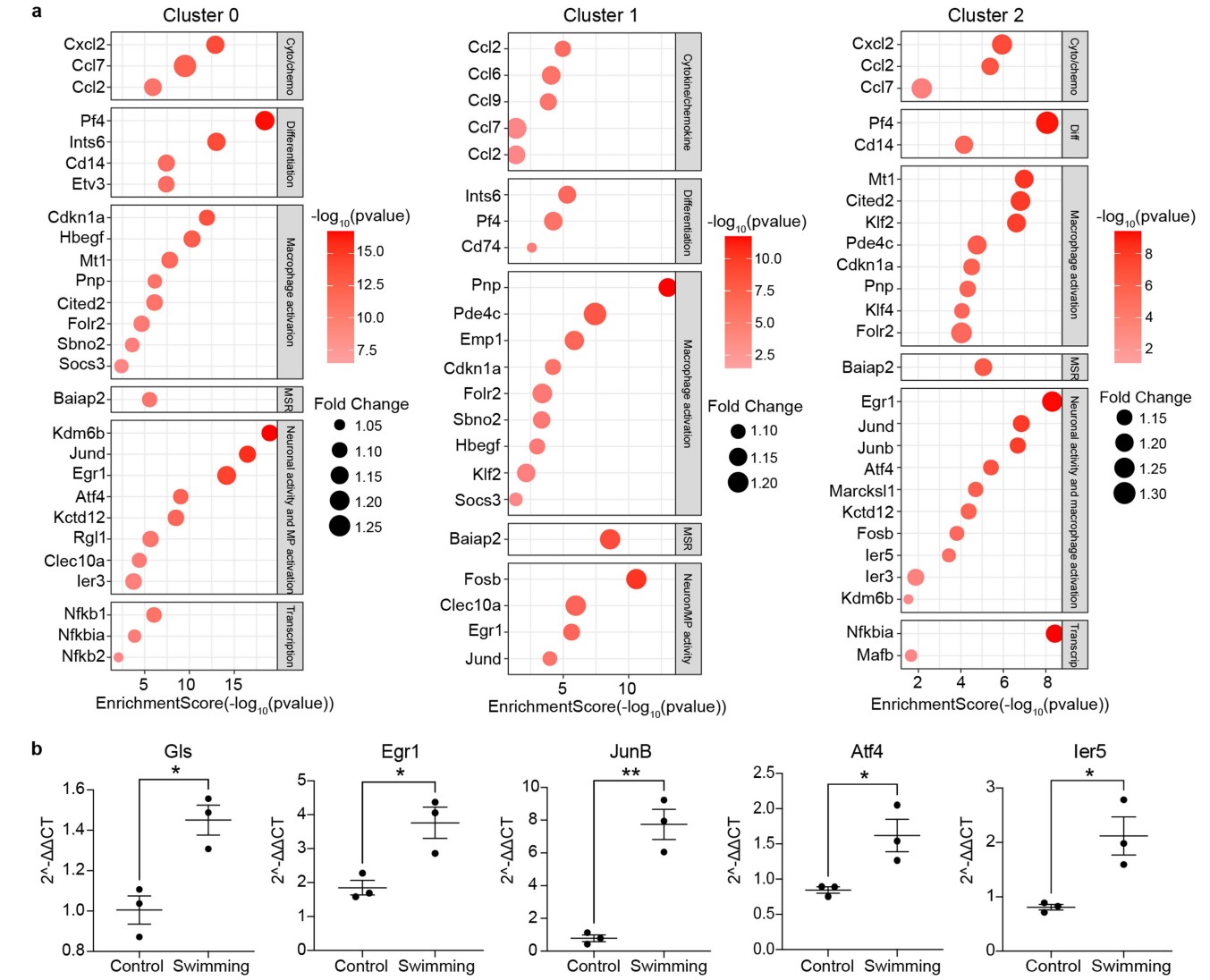

**Extended Data Fig. 8 | All muscle spindle macrophage (MSMP) clusters respond to optogenetic proprioceptive neuronal activation and swimming. a**, Differential expression (DE) analysis showing increased expression of genes involved in cytokine/chemokine signalling, macrophage/neuron activation, transcription, differentiation, and macrophage shape remodel (MSR) (FDR < 0.05, fold change > 1) of MSMP cluster 0 (a), cluster 1 (b) and cluster 2 (c) after optogenetic stimulation of Pv neurons. **b**, RT-qPCR results showing increased expression of glutaminase (*Gls*) and early response genes (*Egr1, JunB, Atf4* and *Ier5*) in sorted MSMP after swimming (n = 3 biological replicates; two-tailed unpaired Student's t-test; mean ± s.e.m.; *p < 0.05, **p < 0.01).

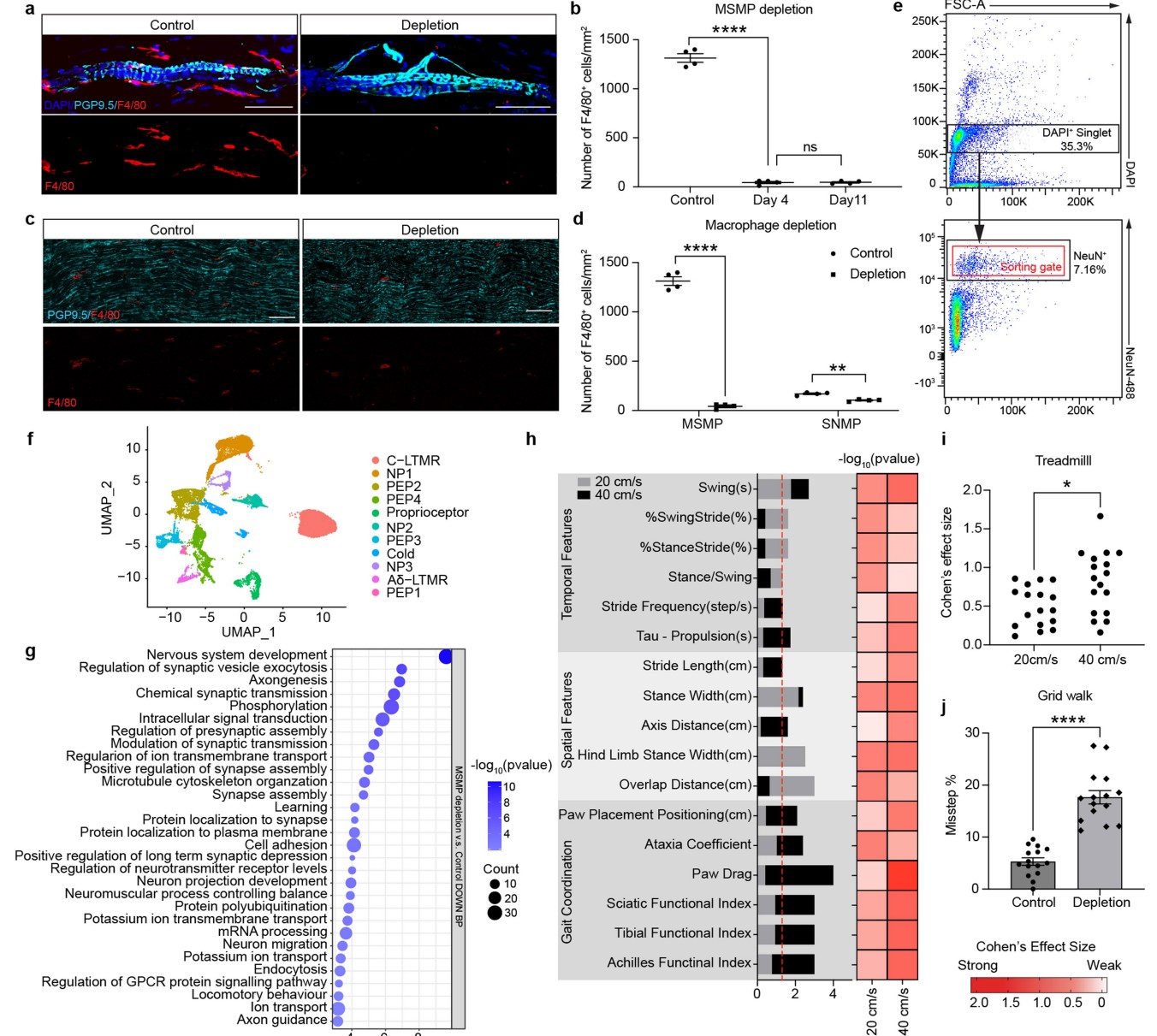

**Extended Data Fig. 9 | Macrophage depletion weakens the molecular signature of dorsal root ganglia proprioceptive neurons and impairs mouse locomotor behaviour. a,c,** Immunostaining for F4/80⁺ macrophages in MS (**a**) and sciatic nerve (**c**) (labelled by PGP9.5 and DAPI counterstaining) before and after MSMP depletion. **b,** Quantification of the number of CD45/F4/80⁺ macrophages in MS on day 0 (control), day 4 and day 11 (n = 4 biological independent mice per group; one-way ANOVA, Tukey's multiple comparisons test; mean ± s.e.m; ns, not significant; ****p < 0.0001). **d,** Quantification of the number of F4/80⁺ macrophages in MS and sciatic nerves before and after macrophage depletion treatment (n = 4 biological independent mice per group; two-tailed unpaired Student's t-test; mean ± s.e.m; **p < 0.01 ****p < 0.0001). **e,** FACS gating to isolate DAPI⁺/NeuN⁺ neuronal nuclei from mouse L4-6 DRGs. **f,** Uniform manifold approximation and projection (UMAP) results show 11 clusters of 19,156 DRG neuronal nuclei. LTMR, low threshold

mechanoreceptors; NP, non-peptidergic C-fibre nociceptors; PEP, C-fibre peptidergic nociceptors; Cold, cold thermoreceptor. **g,** Functional analysis of biological process (BP) analysis of downregulated genes in 1,352 proprioceptive neurons after macrophage depletion (FDR < 0.05, log₂(fold change) > 0.25). GO pathways categorized by the DAVID Bioinformatics Resource with p < 0.05. Scale bar = 50 μm. **h,** Treadmill result showing gait variables (spatiotemporal and coordination features) with statistically significant differences (p < 0.05) between conditions. Cohen's effect size was interpreted as weak (<0.2), moderate (0.2–1), or strong (>1). Red dash line represents p = 0.05. **i,** Comparison between Cohen's effect size of each gait variable listed in figure c at the speed of 40 cm/s and 20 cm/s (17 variables, paired Student's t-test, *p < 0.05). **j,** Grid walk results showing the increased percentage of misstep after macrophage depletion (n = 15 biological independent mice per group; two-tailed unpaired Student's t-test; mean ± s.e.m; ****p < 0.0001).

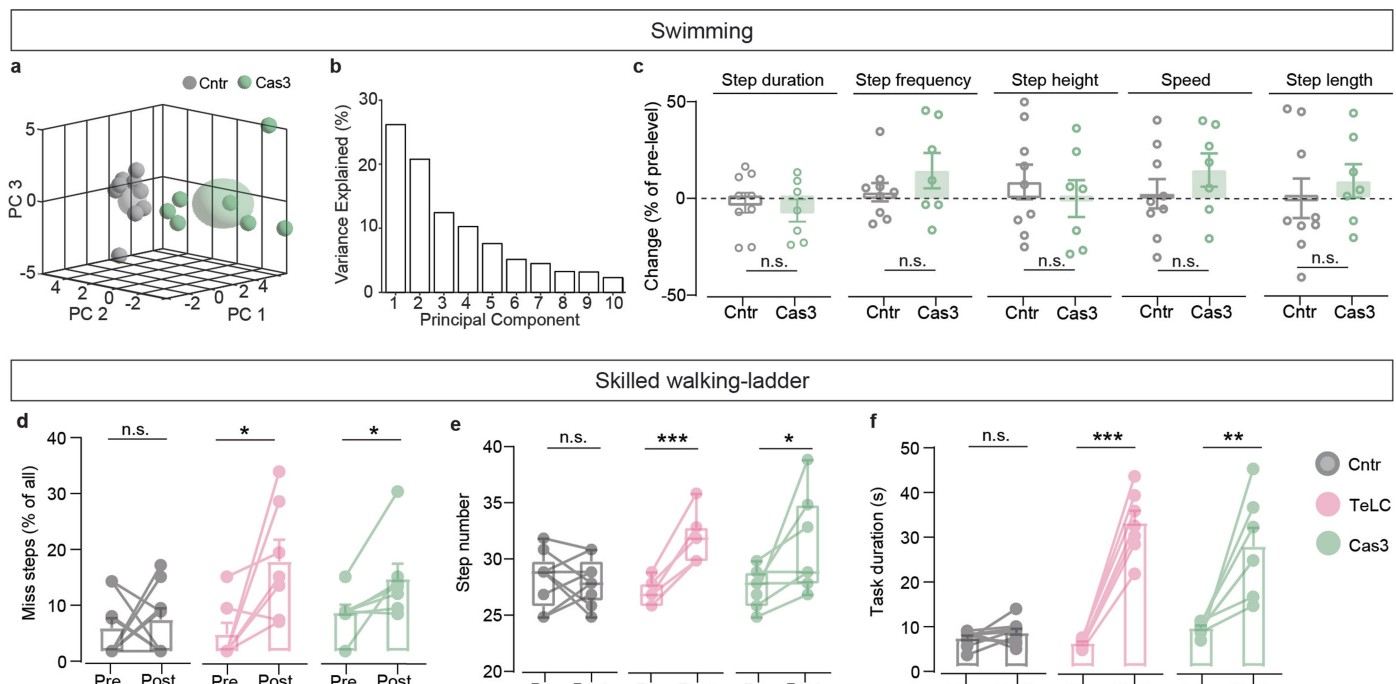

**Extended Data Fig. 10 | Analysis of gait dynamics after manipulations of MSMP in a free swimming or skilled locomotion paradigm. a–c,** PCA analysis clustering (**a**), variance explained by the first ten principal components (**b**) and changes in step duration, frequency, height, speed, and length (mean ± s.e.m, **c**) control (cntr, grey, n = 9 biological independent mice) and MSMP depleted (Cas3, green, n = 7 biological independent mice) CX3CR1$^{cre}$::ChR2$^{YFP}$ mice during swimming, referred to figure 5 (Mann-Whitney tests). **d–f,** Percentage of missed steps (**d**), step number (**e**) and time (**f**) to complete the task for CX3CR1$^{Cre}$::ChR2$^{YFP}$ mice walking on a ladder before and after the delivery of a cre-dependent virus to express a fluorophore (cntr, grey, n = 9 biological independent mice, mean ± s.e.m), and either TeLC (magenta, n = 7 biological independent mice, mean ± s.e.m) or Caspase 3 (Cas3, green, n = 7 biological independent mice, mean ± s.e.m). Statistical analysis was conducted using paired Student's t-test (**d**,**f**) and Wilcoxon test (**e**) to determine significance. Significance levels: n.s. = not significant, *p < 0.05, **p < 0.01, ***p < 0.001.

# Reporting Summary

## Statistics

For all statistical analyses, confirm that the following items are present in the figure legend, table legend, main text, or Methods section.

| n/a | Confirmed | |
|---|---|---|
| ☐ | ☒ | The exact sample size (*n*) for each experimental group/condition, given as a discrete number and unit of measurement |
| ☐ | ☒ | A statement on whether measurements were taken from distinct samples or whether the same sample was measured repeatedly |
| ☐ | ☒ | The statistical test(s) used AND whether they are one- or two-sided *Only common tests should be described solely by name; describe more complex techniques in the Methods section.* |
| ☒ | ☐ | A description of all covariates tested |
| ☐ | ☒ | A description of any assumptions or corrections, such as tests of normality and adjustment for multiple comparisons |
| ☐ | ☒ | A full description of the statistical parameters including central tendency (e.g. means) or other basic estimates (e.g. regression coefficient) AND variation (e.g. standard deviation) or associated estimates of uncertainty (e.g. confidence intervals) |
| ☐ | ☒ | For null hypothesis testing, the test statistic (e.g. *F*, *t*, *r*) with confidence intervals, effect sizes, degrees of freedom and *P* value noted *Give P values as exact values whenever suitable.* |
| ☒ | ☐ | For Bayesian analysis, information on the choice of priors and Markov chain Monte Carlo settings |
| ☒ | ☐ | For hierarchical and complex designs, identification of the appropriate level for tests and full reporting of outcomes |
| ☐ | ☒ | Estimates of effect sizes (e.g. Cohen's *d*, Pearson's *r*), indicating how they were calculated |

*Our web collection on statistics for biologists contains articles on many of the points above.*

## Software and code

Policy information about availability of computer code

| Data collection | Data collection, including image acquisition, laser capture microdissection, cell sorting, RNA sequencing, optogenetics, electrophysiological recording, locomotor tests were performed with the specific instrument software installed on the instruments, as detailed in the methods. |
|---|---|
| Data analysis | Image analysis: LAS X (v3.7,4), ImageJ-Fiji (v2.9.0). Flow cytometry: FlowJo (v10.9.0). Electrophysiology: pClamp (v11.2.1) Bulk RNA-seq: bcl2fastq (v2.20), FastQC (v0.11.9), Trim-Galore (v0.6.6), Salmon (v1.6.0), Tximeta (v1.12.4), DESeq2 (v1.34.0), R statistical environment (v4.1.2). scRNA-seq: Cell Ranger Software Suite (v7.1.0), Seurat (v4) in R 4.1.2. Locomotor test: DigiGait™ Imaging system (Mouse Specifics Inc., Boston, MA) Statistic, analysis and graphs: GraphPad Prism (v9.4.0), ggplot2 (v3.3.6), Adobe Illustrator (2022), Adobe photoshop (2024), Microsoft excel, DigiGait™ analysis system (Mouse Specifics Inc., Boston, MA), Matlab 2019b, IBM SPSS Statistics 26.0, online bioinformatics platform (https://www.bioinformatics.com.cn/)<br><br>All RNA sequencing data are available from the NCBI GEO database under accession number GSE244351 and GSE246400. Source data are provided with this paper.<br><br>Code is available at DOI: 10.5281/zenodo.13902719. |

For manuscripts utilizing custom algorithms or software that are central to the research but not yet described in published literature, software must be made available to editors and reviewers. We strongly encourage code deposition in a community repository (e.g. GitHub). See the Nature Portfolio guidelines for submitting code & software for further information.

## Data

Policy information about availability of data

All manuscripts must include a data availability statement. This statement should provide the following information, where applicable:
- Accession codes, unique identifiers, or web links for publicly available datasets
- A description of any restrictions on data availability
- For clinical datasets or third party data, please ensure that the statement adheres to our policy

All RNA sequencing data are available from the NCBI GEO database under accession number GSE244351 (https://www.ncbi.nlm.nih.gov/geo/query/acc.cgi?acc=GSE244351) and GSE246400 (https://www.ncbi.nlm.nih.gov/geo/query/acc.cgi?acc=GSE246400).
Sequences were demultiplexed and adapters trimmed with `bcl2fastq-v2.20 (Illumina). For bulk RNA-seq, read quality controls were carried out using FastQC-v0.11.9 and removing remaining adapters with Trim-Galore (v0.6.6). RNA-seq analysis was run using the COMBINE lab's Salmon (v1.6.0)-DESeq2 (v1.34.0) pipeline in R (v4.1.2). Reads were mapped to the M25 GENCODE reference mouse genome. For scRNA-seq, Samples were de-multiplexed into FASTQ reads and then aligned to the mouse GRCm39 genome reference. Sample de-multiplexing, sequence alignment, barcode processing and single cell 3' unique molecular identifier (UMI) counting were performed by using Cell Ranger Software Suite (v7.1.0) and quality control, data integration and further analysis were performed by using Seurat (v4).

## Research involving human participants, their data, or biological material

Policy information about studies with human participants or human data. See also policy information about sex, gender (identity/presentation), and sexual orientation and race, ethnicity and racism.

| | |
|---|---|
| Reporting on sex and gender | N/A |
| Reporting on race, ethnicity, or other socially relevant groupings | N/A |
| Population characteristics | N/A |
| Recruitment | N/A |
| Ethics oversight | N/A |

Note that full information on the approval of the study protocol must also be provided in the manuscript.

# Field-specific reporting

Please select the one below that is the best fit for your research. If you are not sure, read the appropriate sections before making your selection.

☒ Life sciences  ☐ Behavioural & social sciences  ☐ Ecological, evolutionary & environmental sciences

For a reference copy of the document with all sections, see nature.com/documents/nr-reporting-summary-flat.pdf

# Life sciences study design

All studies must disclose on these points even when the disclosure is negative.

| | |
|---|---|
| Sample size | The nature of the n is described for each experiment in the corresponding figure legends. Sample size determinations are based on previous experience (Serger et al., 2022) and standards in the field. |
| Data exclusions | No data was excluded |
| Replication | 3-15 independent biological replicates were used for each experiment. The exact number of animals were given in each figure legend for each experiment. |
| Randomization | The biological groups were homogeneous in terms of age (8-12 weeks) and mice. Randomization followed a computerized sequence. Specifically, allocation of mice to control or experimental group was randomised using a random number generator. |
| Blinding | All cell counting and measurements, behavioural tests, and fluorescence analysis were measured in blind. |

# Reporting for specific materials, systems and methods

We require information from authors about some types of materials, experimental systems and methods used in many studies. Here, indicate whether each material, system or method listed is relevant to your study. If you are not sure if a list item applies to your research, read the appropriate section before selecting a response.

## Materials & experimental systems

| n/a | Involved in the study |
|-----|----------------------|
| ☐ | ☒ Antibodies |
| ☒ | ☐ Eukaryotic cell lines |
| ☒ | ☐ Palaeontology and archaeology |
| ☐ | ☒ Animals and other organisms |
| ☒ | ☐ Clinical data |
| ☒ | ☐ Dual use research of concern |
| ☒ | ☐ Plants |

## Methods

| n/a | Involved in the study |
|-----|----------------------|
| ☒ | ☐ ChIP-seq |
| ☐ | ☒ Flow cytometry |
| ☒ | ☐ MRI-based neuroimaging |

## Antibodies

| | |
|---|---|
| Antibodies used | Anti-PGP9.5 (Proteintech, 14730-1-AP, 1:500)<br>Anti-PGP9.5, clone 1C9E11 (Proteintech, 66230-1-Ig, 1:500)<br>Anti-Tuj1, clone 5G8 (Promega, G7121, 1:500)<br>Anti-CD45 (R&D, AF114, 1:500)<br>Anti-F4/80, clone A3-1 (Bio-Rad, MCA497GA, 1:200)<br>Anti-Iba1, clone EPR16588 (Abcam, AB178846, 1:200)<br>Anti-CD68, cone PG-M1 (ThermoFisher, MA512507, 1:100)<br>Anti-B220-FITC, clone: RA3-6B2 (Biolegend, 103228, 1:100)<br>Anti-Nk1.1, clone: PK136 (Stemcell, 100-0459, 1:200)<br>Anti-CD3, clone SP7 (Abcam, ab16669, 1:200)<br>Anti-GFP (Abcam, AB13970, 1:500)<br>Anti-Collagen IV (Merk, AB769, 1:500)<br>Anti-Grin2a (Abcam, AB203197, 1:200)<br>Anti-Gria2 (Merk, AB1768, 1:200)<br>Alpha-Bungarotoxon Conjugates Alexa Fluor 488 (Invitrogen, B13422, 1:500)<br>Anti-IL1RA (ThermoFisher, PA5-21776, 1:200)<br>Anti-HSPA1A (ThermoFisher, PA5-34772, 1:200)<br>Anti-CD31, clone WM59 (ThermoFisher, MA1-26196, 1:200)<br>DAPI (Sigma, D5942, 1:1000)<br>Anti-NeuN-Alexa Fluor488, clone A60  (Merckmilip, MAB377X, 1:100)<br>Live/Dead-Aqua (Thermofisher, L34966A, 1:200)<br>Anti-CD45-PE, clone 30-F11 (Biolegend, 103106, 1:100)<br>Anti-Cd11b-APV/Cyaine7, clone M1/70 (Biolegend, 101226, 1:100)<br>Anti-F4/80-APC, clone BM8 (Biolegend, 123115, 1:100)<br>Anti-Cx3cr1-BV605, clone SA011F11 (Biolegend, 149027, 1:100)<br>Donkey anti-rabbit Alexa Fluor 488 (ThermoFisher Scientific, A21206, 1:500)<br>Donkey anti-chicken Alexa Fluor 488 (Jackson ImuRes, 703545155, 1:500)<br>Donkey anti-goat Alexa Fluor 568 (Life Technologies, A11057, 1:500)<br>Donkey anti-rabbit Alexa Fluor 568 (ThermoFisher Scientific, A10042, 1:500)<br>Donkey anti-mouse Alexa Fluor 594 (ThermoFisher Scientific, A21203, 1:500)<br>Donkey anti-rat Alexa Fluor 647 (Abcam, ab150155, 1:200) |
| Validation | All antibodies used in this study are from commercial suppliers (see notes above for each antibody) that have verified the specificity of the antibodies. All the antibodies have been previously used by various laboratories. All secondary antibodies are verified to not give a specific staining without the primary antibody. Primary antibodies:<br>Anti-PGP9.5 (Proteintech, 66230-1-Ig 1:500) was validated for immunofluorescent staining in mouse tissue (PMID: 32426489). We further validate this antibody in Figure2.<br>Anti-PGP9.5 (Proteintech, 14730-1-AP, 1:500) was validated for immunofluorescent staining in mouse tissue (PMID: 38437959). We further validate this antibody in Figure2.<br>Anti-Tuj1 (Promega, G7121, 1:500) was validated for immunofluorescent staining in mouse tissue (PMID: 34986324). We further validate this antibody in Extended Figure 1.<br>Anti-CD45 (R&D, AF11, 1:500) was validated for immunofluorescent staining in mouse tissue (PMID: 32221369). We further validate this antibody in Figure2.<br>F4/80 (Bio-Rad, MCA497GA, 1:200) was validated for immunofluorescent staining in mouse tissue (PMID: 36611965). We further validate this antibody in Figure2.<br>Anti-Iba1 (Abcam, AB178846, 1:200) was validated for immunoflourescent staining in mouse tissue (PMID: 32863210). We further validate this antibody in Extended Figure 1.<br>Anti-CD68 (ThermoFisher, Cone PG-M1, MA512507, 1:100). Validated for use in flow cytometry, IHC, IF, WB.<br>Anti-B220-FITC (Biolegend, clone: RA3-6B2, 103228, 1:200) was validated for immunofluorescent staining in mouse tissue (PMID: 29541074).<br>Anti-Nk1.1 (Stemcell, clone: PK136, 100-0459, 1:200). Validated for use in flow cytometry, IHC, IF, IP.<br>Anti-CD3 (Abcam, AB16669, 1:200). Validated for use in flow cytometry, IHC, IF, WB.<br>Anti-GFP (Abcam, AB13970, 1:500) was validated for immunofluorescent staining in mouse tissue (PMID: 32426489). We further validate this antibody in Figure2.<br>Collagen IV (Merk, AB769, 1:500) was validated for immunofluorescent staining in mouse tissue (PMID: 35730982). We further validate this antibody in Figure2.<br>Grin2a (Abcam, AB203197, 1:200). Validated for use in flow cytometry, IHC, ICC/IF, WB. We further validate this antibody in Figure3.<br>Gria2 (Merk, AB1768, 1:200) was validated for immunofluorescent staining in mouse tissue (PMID: 29941910). We further validate |

this antibody in Figure3.
Alpha-Bungarotoxon Conjugates Alexa Fluor 488 (Invitrogen, B13422, 1:500). Validated for use in flow cytometry, IHC, IF. We further validate this antibody in Figure2.
DAPI (Sigma, D5942, 1:1000)
Anti-NeuN Conjugates Alexa Fluor 488 (Merckmilip, MAB377X, 1:100). was validated for immunofluorescent staining in mouse tissue (PMID: 34582785).
Live/Dead-Aqua (Thermofisher, L34966A, 1:200) was validated for flow cytometry. We further validate this antibody in Extended Figure 2.

CD45-PE (Biolegend, 103106, 1:100) was validated for flow cytometry. We further validate this antibody in Extended Figure 2.
Cd11b-APV/Cyaine7 (Biolegend, 101226, 1:100) was validated for flow cytometry. We further validate this antibody in Extended Figure 2.
F4/80-APC (Biolegend, 123115, 1:100) was validated for flow cytometry. We further validate this antibody in Extended Figure 2.
Cx3cr1-BV605 (Biolegend, 149027, 1:100) was validated for flow cytometry. We further validate this antibody in Extended Figure 2.

# Animals and other research organisms

Policy information about studies involving animals; ARRIVE guidelines recommended for reporting animal research, and Sex and Gender in Research

| Laboratory animals | Original mouse lines used to generate the mouse lines used in this study:<br>• C57BL6/J mice: Charles River Laboratories.<br>• CX3CR1:GFP mice: was gifted by Marzia Malcangio. Details are described in PMID:24743146.<br>• PvCre : mice from Jakcson Laboratories (known as B6;129P2-Pvalbtm1(cre)Arbr/J), code: 008069<br>• Cx3Cr1Cre: from the Jackson Laboratories (known as B6J.B6N(Cg)-Cx3cr1tm1.1(cre)Jung/J), code 025524<br>• R26ChR2: from the Jackson Laboratories (known as B6;129S-Gt(ROSA)26Sortm32(CAG-COP4*H134R/EYFP)Hze/J), code: 012569<br>• R26NpHR3: from the Jackson Laboratories (known as B6;129S-Gt(ROSA)26Sortm39(CAG-hop/EYFP)Hze/J), code 014539<br><br>All male and female mice were used between 8 to 12 weeks of age were used for all experiments. Mice were maintained under standard housing conditions on a 12h light/dark cycle with food and water provided, at a constant room temperature (RT) and humidity (21-24°C and 45-65%, respectively). |
| --- | --- |
| Wild animals | No wild animals were used in this study. |
| Reporting on sex | Both male and female mice ranging from 8 to 12 weeks of age were used for all experiments. |
| Field-collected samples | No field collected samples were used in this study. |
| Ethics oversight | Animal work was conducted according to UK Home Office license legislation under the Animals (Scientific Procedures) Act 1986, with Local Ethical Review by the Imperial College London Animal Welfare and Ethical Review Body Standing Committee (AWERB) or, under the EU and Danish legislation to ensure the animals' well-being. |

Note that full information on the approval of the study protocol must also be provided in the manuscript.

# Flow Cytometry

## Plots

Confirm that:

☒ The axis labels state the marker and fluorochrome used (e.g. CD4-FITC).

☒ The axis scales are clearly visible. Include numbers along axes only for bottom left plot of group (a 'group' is an analysis of identical markers).

☒ All plots are contour plots with outliers or pseudocolor plots.

☒ A numerical value for number of cells or percentage (with statistics) is provided.

## Methodology

| Sample preparation | To sort cells in vivo, mice were anaesthetised with ketamine (80 mg/kg) and xylazine (10 mg/kg). Muscle, heart and lung were dissected and kept in cold DPBS on ice after cardiac perfusion (20 ml DPBS). For muscle macrophages, to reduce the contamination of NMS macrophages, fat, perimysium, nerves and tendons were carefully removed from dissected muscles. Muscle (EDL, soleus, TA and Gastro), heart and lung were transferred to digestion buffers and cut into small cubes. Digestion buffer contained Collagenase B (2 mg/ml), Dispase II (0.83 mg/ml), DNase I (0.25 mg/ml) and DNase I buffer (10 nM Tris Base, 2.5 mM MgCl2, 0.1 mM CaCl2), dissolved in RNase-free DPBS. Tissue suspensions were incubated in 15 ml digestion buffer in a 37 °C water bath for 50 minutes. Tubes were shaken every 5 minutes. 10 ml ice-cold MojoSort buffer (BioLegend) was added to stop digestion. Cell suspension was filtered through 70 μm and 40 μm cell strainers in series. Then, cells were washed with 5 ml cold MojoSort Buffer 2 times and resuspended with 800 μL cold MojoSort buffer for FACS staining. Cell suspension was incubated with mixed antibodies at 4 °C for 20 minutes in the dark. The following antibodies were used: PE-conjugated anti-CD45 (Biolegend, 103106, 1:100), APC/Cyaine7-conjugated anti-CD11b (Biolegend, 101226, 1:100), APC-conjugated anti-F4/80 (Biolegend, 123115, 1:100), Brilliant Violet-conjugated anti-CX3CR1 (Biolegend, 149027, 1:100). LIVE/ |
| --- | --- |

DEAD Fixable Aqua Dead Cell Stain Kit (Thermo Fisher Scientific, L34966, 1:200) was used to identify live/dead cells. Stained cells were washed for 2 times with 1 ml cold MojoSort, resuspended with 600 μL MojoSort buffer (with RNase inhibitor, 1:500), filtered through a 40 μm cell strainer and kept on ice until sorting. Becton Dickinson FACS Aria Fusion Flow Cytometer with 100 μm nozzle was set to 4 °C to protect RNA from degradation. Gating was set as FSC-A/SSC-A - Singlet/Doublet - Live/Dead--CD45+-CD11b+-F4/80+-CX3CR1+. Single stain, Fluorescence Minus One (FMO) and negative controls were used for the gating boundaries. Target cells were directly collected into cold collection buffer (DPBS with RNase inhibitor (1:500)).
L4-6 dorsal root ganglia (DRG) from control or macrophage depleted mice were sacrificed, dissected, and snap frozen in liquid nitrogen on day 4. On the day of sorting, DRG were placed into homogenisation buffer (0.25 M sucrose, 25 mM KCl, 5 mM MgCl2, 20 mM tricine-KOH pH7.8, 5 μg/ml actinomycin, 1% BSA, 0.15 mM spermine, 0.5 mM spermidine, EDTA-free protease inhibitor, phosphatase inhibitor, RNase inhibitor). DRGs were homogenised by pelleting with a plastic pestle for 15 seconds. Trition-X100 (Sigma) was added to the homogenisation mixture to reach the final concentration of 0.1%. Another 15-second stroke was added to further homogenise the DRGs. Finally, DRG homogenisation suspension was filtered through 70 μm followed by 40 μm cell strainers and incubated with NeuN-Alexa Fluor488 antibody (MAB377X Clone A60, 1:100) for 1 hour in cold room with gentle rotation. After washing with washing buffer (homogenisation buffer + 0.1% Triton-X100 + 2% BSA) 3 times, DRG nuclei were resuspended in 2 ml washing buffer and incubated with DAPI for 10 minutes. DAPI+/NeuN+ DRG neuronal nuclei were sorted by Aria III sorter and collected in washing buffer for 10x single nucleus RNA sequencing according to 10X Chromium Next GEM Single Cell 3' Reagent Kits v3.1 (Dual Index) protocol.

| | |
|---|---|
| Instrument | BD FACS Aria Fusion or BD FACS Aria III |
| Software | FlowJo (v10.9.0) |
| Cell population abundance | Purity of sorted cells or nuclei fraction was above 90%. |
| Gating strategy | Cells were gated as FSC-A/SSC-A - Singlet/Doublet - Live/Dead--CD45+-CD11b+-F4/80+-CX3CR1+ (Extended Figure 2). Neuronal nuclei were gated as FSC-A/SSC-A - Singlet/Doublet - DAPI+- NeuN+ (Extended Data Figure 9). |

☒ Tick this box to confirm that a figure exemplifying the gating strategy is provided in the Supplementary Information.

