## [Peer Review File · Nature]

Macrophages excite muscle spindles with glutamate to bolster locomotion

Corresponding Author: Professor Simone Di Giovanni

Version 1:

Reviewer comments:

Referee #1

(Remarks to the Author)
2023-07-12035A -

In this study Yan et al, propose a new role for macrophages in controlling nerve activity and contraction of skeletal muscles at the neuro-muscular synapse. The authors propose that macrophages are integral part of the neuromuscular circuitry. They suggest that muscle spindle macrophages regulate nerve activity using glutamate mediated signaling. The results are novel. If true, these results would improve our understanding of the neuro-muscular synapse in the control of skeletal muscle activity and should be of broad interest for the biomedical community.

The authors use innovative approaches to address the function of muscle spindle macrophages (MSMP). Microdissection was used to sort macrophages associated with the muscle spindle and perform RNA-seq analysis. Conditional genetic targeting and in-vivo optogenetics are used to experimentally study macrophage function.

The conclusion that MSMP cells control over neural activity is based on a 'gain of function approach where they ectopically express the Chr2 channel (a light-driven proton pump that depolarize neurons) under CX3cr1-cre to activate macrophages optogenetically, and a 'loss of function' approaches where they ectopically express the halorhodopsin (NpHR3 which hyperpolarize neurons) to inhibit macrophage activation, and a macrophage 'depletion treatment' with a combination of Clodronates and kinase inhibitors. Controls are missing, and there are methodological issues that should be addressed before the conclusion is made.

Although the data presented may be compatible with the hypothesis, they fall short of demonstrating the roles of macrophages at the neuro-muscular synapse.

Specific comments

1- To my knowledge, induction of Chr2 expression to depolarize/activate macrophages was used in two study, both using Cx3cr1 ERT2-Cre (Hulsmans, Cell 2017, Yi M-H, Plos Biology, 2021).

Assuming that the authors find that CX3cr1-Cre allows relatively specific labeling of MSMP in muscle (see point 4,5), the authors should document the expression of Chr2 and the activation of Cx3cr1-cre expressing cells in the muscle in CX3cr1-cre x RosaChr2 mice, for example by histology and RNAseq. How do this activation relate to macrophage activation observed in wt mice or in pv-cre mice? All these experiments should include littermates cre-neg controls.

More importantly, induction of NpHR3 expression to hyperpolarize/de-activate macrophages was not reported before. The authors do not provide a citation from the literature where this model was use for macrophages. Some more details on the model would be useful, and the authors should also document the 'inactivation' of 'hyperpolarized' macrophages expressing NpHR3 by in their CX3cr1-cre x NpHR3 mice and in littermates cre-neg controls that should be included in the experiments. How do this relate to macrophage activation observed in wt mice?

2- The specificity of the macrophage 'depletion treatment' with a combination of Clodronates and kinase inhibitors is questionable (off target effects are very likely) and thus do not add much to the manuscript. To support the mechanism of action proposed by the authors and study the impact of MSMP on locomotor behaviors, the authors could use the CX3cr1-cre model crossed with a Rosa DTR mouse for depletion, or the conditional ko of glutamine synthase using a Glu-synthase floxed allele.

3- A large part of the manuscript is dedicated to the interpretation of RNA-seq results. They suggest that muscle spindle macrophages have an enrichment in neuronal genes, synaptic molecular pathways as well as glutamate related genes. Did the author consider the possibility that phagocytosis of neighboring neural cell may be involved? More validation would be needed before concluding these are macrophage genes. Single nuclei RNAseq analysis of the unsorted muscle spindle followed by nuclei annotation could help. Single nuclei RNAseq analysis of sorted macrophages nuclei (e.g. based on Pu.1 staining) could also be useful.

4- The authors use CX3CR1-gfp mice, where GFP reflects expression of CX3CR1 to identify and analyze MSMP. The comparison of F4/80+ and CX3CR1-gfp+ cells in the spindles (Figure 2) by immunofluorescence is methodologically questionable because the experiment is performed in 15 areas from 4 mice, and dot plots do not distinguish individual mice. For the statistical analysis it would be better for example to show mean results per mice. Of note the authors describe the Cx3cr1 CreERT2 (and Pv CreERT2) mice in the method section, while the results and legends indicate Cx3cr1 Cre and Pv Cre mice, and do not mention the use of Tamoxifen. This should be precised/corrected.

5- With the caveat above, the data suggest that there are more Cx3cr1+ cells than F4/80+ cells, with ~80% of CX3CR1+ cells expressing F4/80. This suggests that ~20% Cx3cr1-expressing cells are not macrophages. Importantly, in the following figures, the authors use a CX3cr1-Cre mice bred to 'activating' and 'inhibitory' alleles. Cells targeted in CX3cr1-Cre mice include cells currently expressing CX3CR1 (as in CX3CR1-gfp mice) plus all cells that have expressed CX3CR1 at any time during their development (in the immune system this mean most myeloid cells, See Yona et al 2013, Immunity). Therefore, the analysis in figure 2 would be more accurately done in CX3cr1-Cre x Rosa LSL-YFP mice, as they may be more than 20% non MSMP cells labeled in this model. If it is the case, the author may have to reconsider the cellular specificity of their gain of function models. The use of alternative models to target macrophages could be necessary.

Referee #2

(Remarks to the Author)

Yan et al. identified a new cellular component within muscle spindles called muscle spindle macrophages (MSMP). These MSMP exhibit distinct molecular characteristics and possess the machinery to produce and release glutamate. Through elegant experiments using transgenic mice, ex vivo culture studies and single-cell sequencing, the authors demonstrate that activation of MSMP triggers firing in proprioceptive sensory neurons, influencing spinal circuits, motor neurons, and muscles. The data suggest that this is happening via a glutamate-dependent mechanism. Moreover, MSMP respond to stretch reflex activation by increasing glutaminase expression, which converts glutamine into glutamate during muscle contraction.

The exciting findings of this manuscript highlight MSMP's homeostatic role in regulating both neural activity and muscle contraction, providing a novel perspective on sensation and motor action. However, from an immune-phenotyping perspective, the study should be improved. Furthermore, the strategy used to deplete macrophages in vivo is not state-of-the-art so that the results observed in behaviour differences may stem from off-target effects.

Major points:

1. The major weakness of the study is the chosen approach to deplete macrophages as this is not specific for muscles. Also depletion of other macrophages, immune activation due to clodronate injections (often activates IL1a) etc. may have an influence on behaviour. Further, clodronate has additional side effects that do not allow to conclude anything regarding specific macrophage functions (see PMID 36976180). Therefore, the conclusions drawn regarding the impact of MSMP on locomotion are preliminary. Instead, the authors should use a Cx3cr1-CreER^{td}/DTR + tamoxifen injection approach (or similar) to deplete e.g. glutaminase in MSMP. Possibly, they could also use a local macrophage depletion using their Cx3cr1-ChR2 model by extending the stimulation with blue light (see also PMID 32718989).
2. Fig. 5: analysis of scRNA-seq data not clearly explained. What is being compared? A typical comparison should be a sham-operated Pv-ChR2 mouse without optogenetic stimulation. Alternatively, the clusters should be identified in situ using the described markers and compared to non-stimulated mice via immunofluorescence
3. Macrophage heterogeneity observed by scRNA-seq after neuronal activity should be validated by in situ/flow cytometry analyses.
4. lines 214-217: Overinterpretation of results since release of glutamine has not been shown in vivo. Tune down the statement or use conditional knockout mouse models to show this mechanism in vivo.
5. Fig. 2A, B: the F4/80 and Cx3cr1 signals show an unusual shape of macrophages, which does not correspond to what is shown in 2G. Can the authors improve visualization, e.g. by including DAPI and performing a whole mount staining, or at least a thick section. 3D rendering of Iba1 signal in mouse and human samples will help the reader to appreciate the location of these macrophages

Minor points:

- a. flow cytometry gates and purity of sorted cells used for assays should be included in the extended data
- b. It is unclear from the methods part how the published RNAseq dataset was integrated into the data

- c. include the line number from Jackson since there can be several lines with a similar locus
- d. Fig. 5c: do the authors mean Zeb2? This figure is rather supplemental information. please show the top expressed genes in the main figure instead of showing differences between clusters
- e. F4/80 is not always expressed highly on all macrophages. Iba1 should be used to quantify macrophages in different regions.
- f. Fig. 2: what are the scale bars in IF pictures?

Referee #3

(Remarks to the Author)

This is a provocative study implicating macrophages within muscle spindles as mediators of the classic myotatic stretch reflect. Some interesting findings are presented, but in the end the authors' model is unclear. Are MSMPs mechanically sensitive, and does stretch of MSMPs lead to Glu release? Are MSMPs attached to the muscle spindles, and do they stretch when the spindles stretch? Some key experiments to test these or related ideas would be welcome.

Interestingly, GO analysis of gene expression experiments revealed that MSMPs not enriched for transcripts implicated in immune responses, as compared to lung and heart macrophages. On the other hand, MSMPs express genes associated with muscle contraction, neuronal projection and synaptic activity, and glutamate transport, synthesis, and release. This begs the question, are MSMPs macrophages? Are they phagocytic?

The optogenetic experiments are critical for the authors' conclusions. An important observation is that MSMP optogenetic stimulation elicits sensory neuron excitation within milliseconds and with low jitter, and subsequent muscle contraction. Some controls are needed here, including non-transgenic controls. It seems critical that the authors report the expression patterns of the opsins in the CX3CR1-ChR2 and CX3CR1Cre-NpHR3 mice in muscles, to provide confidence that these actuators are restricted in their expression and in the expected cell types (MSMPs and not other cells).

The most crucial experiments are the loss-of-function - CX3CR1Cre-NpHR3 and genetic depletion experiments. Ideally, recordings in the MSPC cells would be done to assess their excitability and the efficacy of optogenetic responses. Why do Glutamate receptor antagonists completely block the evoked response while NpHR3 activation/hyperpolarization leads to a modest, at most 35% inhibition? Is this because NpHR3 is ineffective in silencing MSMPs? More importantly, electrophysiological measurements of proprioceptor responses and excitation of muscle cells as well as the stretch reflex should all be assessed in the macrophage depleted mice, since the depletion appears complete (Extended figure 5), to complement the behavioral measurements. This would be the most robust test of necessity, since the CX3CR1Cre-NpHR3 mediated hyperpolarization experiments may never provide complete inhibition (Figure 3).

Version 2:

Reviewer comments:

Referee #1

(Remarks to the Author)

The authors report novel and important findings using innovative methodological approaches. In the revised manuscripts they have addressed most of our comments.

They improve the imaging of MSMP, and they now show specific expression of the channel rhodopsin in MSMP, via immunofluorescence. They also show that depolarization and hyperpolarization happens in the macrophages using a patch clamp technic on sorted macrophages in vitro with their optogenetic model.

They came up with 2 additional macrophages depletion strategies, using cre dependent TeLC delivery on the gastrocnemius muscle of Cx3cr1Cre-Chr2 mice to block vesicles release from macrophages. They also use cre dependent AAV-DIO-Caspase 3 injection in the muscle to deplete macrophages, which seems to work for depletion and results in impairment of locomotor behavior.

Overall, however, the mechanism by which MSMP promote muscle contraction is still not clear. The authors suggest that neuronal genes expressed by macrophage may play a role, they propose a key role for glutamine but this is not demonstrated in vivo and macrophages express many genes, including cytokines, which could also influence nerves stimulation

On balance, this is a very interesting paper describing a novel type of macrophage function, and which would deserve to be published, with the caveat that it lacks a clear biological mechanism.

Referee #2

(Remarks to the Author)

The revised manuscript has addressed the most important points. The models chosen to deplete macrophages locally are elegant and make a strong point for MSMPs controlling locomotion.

I would still insist on exploiting the scRNA-seq data so that it becomes clear to the reader what can be actually observed between activated and control MSMPs besides the one violin plot shown for the glutaminase. ○ The UMAPs showing expression of genes in almost 100% of cells doesn't help. This should be moved into supplement, also the scheme explaining what was done is somewhat self-explanatory and can be supplementary, as it should be found in the text and methods part.

A dot plot/violin plots showing what is defining the identity of the macrophage clusters 0-3 would be a first step (it is clear that cluster 4 are proliferating macrophages). The conditions should be split, as shown in Figure 5f for this representation. The authors write that the cells have similar identities, but still, something is determining their clustering and should be shown. For the macrophage community it would be of interest to define/validate these subpopulations also on a protein level. Do they have distinct subanatomical locations? Cluster 3 seems to not upregulate glutaminase, so it may not be involved in the described process. This needs to be addressed to some degree.

Referee #3

(Remarks to the Author)

This is a re-review of a manuscript that reports a series of experiments that address the role of resident macrophages in muscle and their role in muscle stretch evoked proprioceptor signaling and function. The revision includes many new experiments, which have improved the study.

I had previously questioned the overall model, and in particular whether MSMP are mechanically sensitive and whether stretch of MSMP leads to Glu release. This issue has not been fully addressed, and how macrophages respond to stimuli and contribute to proprioceptor signaling remains unclear.

Version 3:

Reviewer comments:

Referee #1

(Remarks to the Author)

In this second revision, the author performed additional experiments which supporting the role of macrophages in muscle activity and identify candidate genes to play roles in this process. The swimming test is a nice addition. The author have also clarified the limitation of their experiments. Overall, this very interesting, conceptually and technically innovative study, should be of great interest for biologists in general and in particular for the study of the neuromuscular junction/synapse.

Referee #2

(Remarks to the Author)

The second revision of this manuscript addresses all mentioned points.

Please check that all subfigures have the used statistics indicated in the figure legend. It is for instance not clear from the legend in Figure 4 and 6, which test is used for which subfigure.

Regarding the used statistics:

The methods part regarding statistics should distinguish between the different purposes of each statistical test, e.g. the Student's t-test is typically used for comparing two groups. The same is, however, true for the Welch's t-test when variances are unequal; Mann-Whitney test is a non-parametric test and should be used for non-normally distributed data when comparing two groups (not more).

Only One-way ANOVA is for comparing more than two groups with normally distributed data.

The authors should check which test is suitable for the data they display and adjust them.

Referee #3

(Remarks to the Author)

The new findings implicating a mechanism involving glutamine release by muscle, its uptake and conversion to glutamate in MSMPs, and glutamate then acting on afferent terminals leading to excitation are interesting.

Referees' comments:

Referee #1 (Remarks to the Author):

2023-07-12035A -

In this study Yan et al, propose a new role for macrophages in controlling nerve activity and contraction of skeletal muscles at the neuro-muscular synapse. The authors propose that macrophages are integral part of the neuromuscular circuitry. They suggest that muscle spindle macrophages regulate nerve activity using glutamate mediated signaling. The results are novel. If true, these results would improve our understanding of the neuro-muscular synapse in the control of skeletal muscle activity and should be of broad interest for the biomedical community.

The authors use innovative approaches to address the function of muscle spindle macrophages (MSMP). Microdissection was used to sort macrophages associated with the muscle spindle and perform RNA-seq analysis. Conditional genetic targeting and in-vivo optogenetics are used to experimentally study macrophage function.

The conclusion that MSMP cells control over neural activity is based on a 'gain of function' approach where they ectopically express the Chr2 channel (a light-driven proton pump that depolarize neurons) under CX3cr1-cre to activate macrophages optogenetically, and a 'loss of function' approaches where they ectopically express the halorhodopsin (NpHR3 which hyperpolarize neurons) to inhibit macrophage activation, and a macrophage 'depletion treatment' with a combination of Clodronates and kinase inhibitors. Controls are missing, and there are methodological issues that should be addressed before the conclusion is made.

Although the data presented may be compatible with the hypothesis, they fall short of demonstrating the roles of macrophages at the neuro-muscular synapse.

Specific comments

We would like to thank this reviewer for their excellent constructive comments.

1-To my knowledge, induction of Chr2 expression to depolarize/activate macrophages was used in two study, both using Cx3cr1 ERT2-Cre (Hulsmans, Cell 2017, Yi M-H, Plos Biology, 2021). Assuming that the authors find that CX3cr1-Cre allows relatively specific labeling of MSMP in muscle (see point 4,5), the authors should document the expression of Chr2 and the activation of Cx3cr1-cre expressing cells in the muscle in CX3cr1-cre x RosaChr2 mice, for example by histology and RNAseq. How do this activation relate to macrophage activation observed in wt mice or in pv-cre mice? All these experiments should include littermates cre-neg controls. More importantly, induction of NpHR3 expression to hyperpolarize/de-activate macrophages was not reported before. The authors do not provide a citation from the literature where this model was used for macrophages. Some more details on the model would be useful, and the authors should also document the 'inactivation' of 'hyperpolarized' macrophages expressing NpHR3 by in their CX3cr1-cre x NpHR3 mice and in littermates cre-neg controls that should be included in the experiments.

Thank you for your insightful comments. We have undertaken several targeted experiments to address your concerns regarding the specificity and functionality of our optogenetic manipulations in muscle spindle macrophages, employing both CX3CR1^{Cre::ChR2^{YFP}} and CX3CR1^{Cre::NpHR3^{YFP}} mouse models.

1. Validation of expression:

We have confirmed the specific expression of ChR2 and NpHR3 in MSMP using CX3CR1^{Cre::ChR2/NpHR3^{YFP}} mice. Immunostaining of muscle sections showed ChR2 and NpHR3-YFP

specific expression in MSMP (macrophages labelled by F4/80 and MS labelled by PGP9.5), validating our Cre-driven expression (see revised Figure 3).

2. Functional validation of optogenetic modulation via patch-clamp:

To directly prove that CX3CR1-ChR2 MSMP are depolarised and that CX3CR1-NpHR3 MSMP are hyperpolarised after light stimulation, we utilized patch-clamp, which is the gold standard for measuring ion channel activity and membrane potential, especially in response to optical manipulation. Patch-clamp analysis of CD45/CD11b/F4/80/CX3CR1⁺ sorted from CX3CR1^{Cre::ChR2}^{YFP} and CX3CR1^{Cre::NpHR3}^{YFP} skeletal muscles demonstrated that light stimulation induced depolarisation and hyperpolarisation in ChR2 and NpHR3 models respectively (see Extended Data Fig.4).

More specifically, light stimulation on cultured cells induced the expected depolarisation in CX3CR1^{Cre::ChR2}^{YFP} cells (see Extended Data Fig.4a-e). The amount of depolarisation was dependent on the stimulus features such as pulse duration and light power, in accordance with the study from Yi et al., (Yi et al., 2021). We specifically tested the short duration pulse with low light power (5 ms, 15% power) that we have used for the in vivo experiments. This stimulation was able to induce sustained depolarisation of the cells.

Similarly, light stimulation on cultured MSMP from CX3CR1^{Cre::NpHR3}^{YFP} mice induced a hyperpolarisation that was dependent on the stimulus parameters as pulse duration and light power (Extended Data Fig.4f-j). As expected, halorhodopsin stimulation required higher light power. We specifically tested stimulus features that we have used for in vivo experiments (high power light and prolonged duration). YFP positive cells responded to this stimulation with sustained hyperpolarisation independently from the length of the stimulation (Extended Data Fig.4h-j). We have now demonstrated that the light-induced activation of halorhodopsin in CX3CR1⁺ cells lead to a prolonged membrane hyperpolarisation.

3. Littermate controls

We have included littermate controls lacking ChR2 (CX3CR1^{Cre+/-::ChR2}^{YFP-/-}) or NpHR3 (CX3CR1^{Cre+/-::NpHR3}^{YFP-/-}) in all experiments to ensure the specificity of our findings and rule out non-specific effects of genetic background or experimental handling (see revised Figure 3, 4 and 6, Extended Data Fig.4).

4. Optogenetic activation in relation to PV-induced activation of MSMP

While the experiments are not directly comparable, conceptually, direct optogenetic macrophage activation via depolarisation relates to the expression of activity-dependent genes after optogenetic stimulation of PV DRG neurons as seen in our RNAseq data (Figure 5). In fact, optical activation of MSMP investigated how MSMP modulate neurons; however, the PV stimulation experiments aimed to investigate how neurons modulate MSMP. These two sets of experiments were done to examine whether there was a cross-talk between macrophages and sensory neurons, generating a reciprocal functional regulatory loop.

5. Optogenetic activation in relation to WT activation

To relate MSMP optogenetic activation to the role of MSMP in physiological settings, we optogenetically inhibited MSMP in NpHR3-YFP positive mice (CX3CR1^{Cre+/-::NpHR3}^{YFP+/-}) and their negative littermate (CX3CR1^{Cre+/-::NpHR3}^{YFP-/-}) during progressive muscle elongation when EMG responses were recorded.

Results showed that the reduction in the amplitude of the muscle action potential in CX3CR1-NpHR3 mice was commensurate to the degree of muscle elongation, reaching about 35% reduction in muscle action potential compared to control condition with the greatest stretch (4 mm) (Figure 3i-m). This supports the idea that the activity of MSMP is required for optimal muscle contraction in response to increasing degrees of muscle elongation.

We have also depleted MSMP for a loss of function experiment and performed locomotor tasks. These experiments showed a role for MSMP in locomotor behaviour, however, as indicated by the reviewers, the depletion is not specific to MSMP complicating data interpretation. Therefore, we employed a cre-dependent strategy for two new loss of function experiments where we perturbed MSMP activity (Cre-dependent expression of tetanus toxin light chain -TeLC) or MSMP number (Cre-dependent expression of Caspase 3) in the gastrocnemius muscle respectively (Figure 4 and 6).

These two new experiments also directly address the second important point raised by this reviewer (see next point).

2- The specificity of the macrophage 'depletion treatment' with a combination of Clodronates and kinase inhibitors is questionable (off target effects are very likely) and thus do not add much to the manuscript. To support the mechanism of action proposed by the authors and study the impact of MSMP on locomotor behaviors, the authors could use the CX3cr1- cre model crossed with a Rosa DTR mouse for depletion, or the conditional ko of glutamine synthase using a Glu-synthase floxed allele.

Specifically, we leveraged the knowledge that TeLC degrades SNARE proteins VAMP1-3 in neurons and other cell types (Sweeney et al., 1995, Pitzurra et al., 1996, Mendez et al., 2011, Bouvier et al., 2015, Hoogstraaten et al., 2020, Calvigioni et al., 2023) to block synaptic vesicle release from MSMP. Importantly, VAMPs are expressed in MSMP (Figure 2O) and previous evidence showed that macrophages express VAMP-related proteins that are sensitive to TeTx proteolytic cleavage (Pitzurra et al., 1996). Therefore, we used viral delivery of TeLC in the gastrocnemius muscle of CX3CR1^{Cre}::ChR2^{YFP} mice to block vesicle release specifically in MSMP.

Additionally, in a separate set of experiments we injected AAV-DIO-Caspase 3 to selectively deplete MSMP (a similar approach had been used for microglia depletion (Jia et al., 2023)). AAV-mCherry was used in controls.

We have perturbed MSMP in the gastrocnemius because of the important function of this muscle in generating the power stroke in swimming and walking on a ladder in mice and because the stretch reflex circuits of this muscle are highly engaged during these tasks.

Both approaches provided consistent results, demonstrating significant and highly muscle specific alterations in spatiotemporal gait parameters and angular excursion variability when MSMP were disrupted. By using these advanced genetic models, we have strengthened the evidence for the critical role of MSMP in locomotor behaviour, thereby enhancing the robustness and validity of our findings.

Expression of TeLC in MSMP and MSMP depletion (about 75%) were confirmed by immunostaining in the gastrocnemius muscle. The results of these experiments are summarised in Figure 4 and 6 and Extended Data Fig.9 and 10.

3-A large part of the manuscript is dedicated to the interpretation of RNA-seq results. They suggest that muscle spindle macrophages have an enrichment in neuronal genes, synaptic molecular pathways as well as glutamate related genes. Did the author consider the possibility that phagocytosis of neighboring neural cell may be involved? More validation would be needed before concluding these are macrophage genes. Single nuclei RNAseq analysis of the unsorted muscle spindle followed by nuclei annotation could help. Single nuclei RNAseq analysis of sorted macrophages nuclei (e.g. based on Pu.1 staining) could also be useful.

Thank you for allowing a clarification here. There are three main lines of evidence that reassure us about the origin of the enrichment of neuronal genes in MSMP.

1. In this study, initially, we performed two independent RNA sequencing experiments: the first was RNAseq after laser capture microdissection (LCM) of MS versus non-MS areas. This LCM-MS RNAseq dataset showed the expression of classical MS-component-related genes, including genes expressed by intrafusal myofibers and sensory afferents. However, we surprisingly also detected the expression of immune-related genes. To prove the presence of immune cells in MS, we performed immunostaining, which showed the presence of macrophages in the MS. To further investigate the molecular identity and signatures of these macrophages, we sorted MSMP by using the immune cell and macrophage markers CD45, CD11b, F4/80 and CX3CR1 that we found highly enriched in the MS and performed RNA sequencing. This RNAseq dataset of sorted MSMP showed the significant expression of neuron-related genes (Figure 2). There were therefore no neuronal components being sequenced in this dataset.
2. In the MS there are no neuronal nuclei, or cell bodies but rather only sensory fibers. The neuronal nuclei and cell bodies of sensory neurons are located in the dorsal root ganglia, far away from the muscle spindles. Furthermore, no other cell of neuronal origin to our knowledge is located in the muscle spindles.
3. Lastly, in our study muscle spindles appear intact and healthy and there is no suggestion that MSMP in proximity to MS have a direct phagocytic activity on the axons as can be seen by additional 3D reconstructions (Extended Movie 1-3). Furthermore, we searched for the expression of axonal mRNA in MSMP, and we could not find any expression above background. This includes the highly abundant axonal mRNAs Nefl, Gap43, Map2, and Uchl1 (this last is actually PGP9.5, which we used to identify the sensory afferents).

4- The authors use CX3CR1-gfp mice, where GFP reflects expression of CX3CR1 to identify and analyze MSMP. The comparison of F4/80+ and CX3CR1-gfp+ cells in the spindles (Figure 2) by immunofluorescence is methodologically questionable because the experiment is performed in 15 areas from 4 mice, and dot plots do not distinguish individual mice. For the statistical analysis it would be better for example to show mean results per mice.

Thanks very much for this important comment, we have followed the advice and changed all quantification figures by replacing with mean results per mice (Figure 2 and Extended Data Fig.1 and 7).

Of note the authors describe the Cx3cr1 CreERT2 (and Pv CreERT2) mice in the method section, while the results and legends indicate Cx3cr1 Cre and Pv Cre mice, and do not mention the use of Tamoxifen. This should be precised/corrected.

Thanks for allowing a clarification here. We did not use the tamoxifen inducible version of the two mice. We have used the CX3CR1^{Cre} (Strain #:025524, Jackson laboratories) and the PV^{Cre} (Strain #:008069 from Jackson laboratories). We realised that in one point of the method there was a mistake with the name of both mice (inadvertently we add ER to the name of the mice). We are sorry for the mistake and the confusion. We have now corrected this and make it clear along the paper.

5- With the caveat above, the data suggest that there are more Cx3cr1+ cells than F4/80+ cells, with ~80% of CX3CR1+ cells expressing F4/80. This suggests that ~20% Cx3cr1-expressing cells are not macrophages. Importantly, in the following figures, the authors use a CX3cr1-Cre mice bred to 'activating' and 'inhibitory' alleles. Cells targeted in CX3cr1-Cre mice include cells currently expressing CX3CR1 (as inCX3CR1-gfp mice) plus all cells that have expressed CX3CR1 at any time during their development (in the immune system this mean most myeloid cells, See Yona et al 2013, Immunity). Therefore, the analysis in figure 2 would be more accurately done in CX3cr1-Cre x Rosa LSL-YFP mice, as they may be more than 20% non MSMP cells labelled in this model. If it is the case, the author may have to reconsider the

cellular specificity of their gain of function models. The use of alternative models to target macrophages could be necessary.

Thanks for allowing to clarify this point. The graph the reviewer is referring to (Figure 2f) showed that approximately 90% of F4/80 positive macrophages are also CX3CR1 positive (CX3CR1⁺-F4/80⁺) and not the opposite. We have renamed the figure labelling to make it clearer (Figure 2f). This implies that our manipulation of the CX3CR1 allele targets F4/80 positive cells specifically.

Referee #2 (Remarks to the Author):

Yan et al. identified a new cellular component within muscle spindles called muscle spindle macrophages (MSMP). These MSMP exhibit distinct molecular characteristics and possess the machinery to produce and release glutamate. Through elegant experiments using transgenic mice, ex vivo culture studies and single-cell sequencing, the authors demonstrate that activation of MSMP triggers firing in proprioceptive sensory neurons, influencing spinal circuits, motor neurons, and muscles. The data suggest that this is happening via a glutamate-dependent mechanism. Moreover, MSMP respond to stretch reflex activation by increasing glutaminase expression, which converts glutamine into glutamate during muscle contraction.

The exciting findings of this manuscript highlight MSMP's homeostatic role in regulating both neural activity and muscle contraction, providing a novel perspective on sensation and motor action. However, from an immune-phenotyping perspective, the study should be improved. Furthermore, the strategy used to deplete macrophages in vivo is not state-of-the-art so that the results observed in behaviour differences may stem from off-target effects.

We would like to thank this reviewer for their excellent constructive comments.

Major points:

1. The major weakness of the study is the chosen approach to deplete macrophages as this is not specific for muscles. Also depletion of other macrophages, immune activation due to clodronate injections (often activates IL1a) etc. may have an influence on behaviour. Further, clodronate has additional side effects that do not allow to conclude anything regarding specific macrophage functions (see PMID 36976180). Therefore, the conclusions drawn regarding the impact of MSMP on locomotion are preliminary. Instead, the authors should use a Cx3cr1-CreER/iDTR + tamoxifen injection approach (or similar) to deplete e.g. glutaminase in MSMP. Possibly, they could also use a local macrophage depletion using their Cx3cr1-ChR2 model by extending the stimulation with blue light (see also PMID 32718989).

Thanks for this comment and suggestion. We agree that this is an important point. Therefore, we decided to design and execute two complementary experiments to conclusively prove the need for MSMP activity and presence in muscle contraction during motor tasks.

To this end, we employed a Cre-dependent strategy for two new loss of function experiments where we perturbed MSMP activity (Cre-dependent expression of tetanus toxin light chain - TeLC) or MSMP number (Cre-dependent expression of Caspase 3) in the gastrocnemius muscles of CX3CR1^{Cre::}ChR2^{YFP} mice (Figure 4 and 6).

Specifically, we leveraged the knowledge that TeLC degrades SNARE proteins VAMP1-3 in neurons and other cell types (Sweeney et al., 1995, Pitzurra et al., 1996, Mendez et al., 2011, Bouvier et al., 2015, Hoogstraaten et al., 2020, Calvigioni et al., 2023) to block synaptic vesicle release from MSMP. Importantly, VAMPs are expressed in MSMP (Figure 2O) and previous evidence showed that macrophages express VAMP-related proteins that are sensitive to TeTx proteolytic cleavage (Pitzurra et al., 1996).

Therefore, we used viral delivery of TeLC in the gastrocnemius muscle of CX3CR1^{Cre+}:ChR2^{YFP} mice to block vesicle release specifically in MSMP.

Additionally, in a separate set of experiments we injected AAV-DIO-Caspase 3 (Cas3) to selectively deplete MSMP (a similar approach had been used for microglia depletion (Jia et al., 2023)). AAV-mCherry was used in controls.

We have perturbed MSMP in the gastrocnemius because of the important function of this muscle in generating the power stroke in swimming and walking on a ladder for mice and because the stretch reflex circuits of this muscle are highly engaged during these tasks.

Both approaches provided consistent results, demonstrating significant and highly muscle specific alterations in spatiotemporal gait parameters and angular excursion variability when MSMP were disrupted. By using these advanced genetic models, we have strengthened the evidence for the critical role of MSMP in locomotor behaviour, thereby enhancing the robustness and validity of our findings.

In conclusion, both the TeLC and Cas3 experiments show the selective role of MSMP in locomotor behaviour.

Expression of TeLC in MSMP and MSMP depletion (about 75%) were confirmed by immunostaining in the gastrocnemius muscle. The results of these experiments are summarised in Figure 4 and 6 and Extended Data Fig.9 and 10.

2. Fig. 5: analysis of scRNA-seq data not clearly explained. What is being compared? A typical comparison should be a sham-operated Pv-ChR2 mouse without optogenetic stimulation. Alternatively, the clusters should be identified in situ using the described markers and compared to non-stimulated mice via immunofluorescence.

Thanks for allowing this clarification. We used contralateral control, with sham-operated without optogenetic stimulation from the same mouse. This is made clearer in the methods section now 'Method: Optogenetic stimulation of neurons (Pv+) or macrophages (CX3CR1+): The right-side sciatic nerve (or muscles) was exposed without light treatment for contralateral control.'

3. Macrophage heterogeneity observed by scRNA-seq after neuronal activity should be validated by in situ/flow cytometry analyses.

d. Fig. 5c: do the authors mean Zeb2? This figure is rather supplemental information. please show the top expressed genes in the main figure instead of showing differences between clusters:

Thanks for this question that allows an important clarification. The aim of this scRNA-seq experiment (a population of CD45-Cd11b-F4/80-CX3CR1 positive cells was selected here) was to prove whether MSMP broadly or selectively respond to the activation of sensory neurons (Pv-ChR2 optogenetic activation). Indeed, all cell clusters express similar classical macrophage markers, with similar cellular identity and the clustering is given by the different level of gene expression changes in activity-responsive genes after neuronal activation. Therefore, each cluster has similar identity with some variable degrees of responses to activation of sensory neurons. Indeed, one of the clusters show Zeb2 as suggested by the reviewer (Figure 5c), apologies for the typo. In conclusion the message here is that the response of MSMP to sensory neuronal optogenetic activation, despite some fluctuations in gene expression, is rather homogenous across the entire cell population.

4. lines 214-217: Overinterpretation of results since release of glutamine has not been shown in vivo. Tune down the statement or use conditional knockout mouse models to show this mechanism in vivo.

Thanks for pointing this out. We agree and we have removed the related statement.

5. Fig. 2A, B: the F4/80 and Cx3cr1 signals show an unusual shape of macrophages, which does not correspond to what is shown in 2G. Can the authors improve visualization, e.g. by including DAPI and performing a whole mount staining, or at least a thick section. 3D rendering of Iba1 signal in mouse and human samples will help the reader to appreciate the location of these macrophages.

Thanks for this comment, we have followed the reviewer's advice and added Iba1 staining in mouse muscles with thicker sections (50 μm Z-stack) (Revised Figure 2g, Extended Data Fig.1a). We also added 3D rendering of Iba1 (Extended 1,2).

We would also like to point out that depending on the orientation of how a given muscle spindle has been sectioned macrophages might appear with a different morphology.

Minor points:

a. flow cytometry gates and purity of sorted cells used for assays should be included in the extended data:

Thanks for allowing clarification here, we have added the flow cytometry gates (Extended Data Fig.2b, c), FMO-CX3CR1 check (Extended Data Fig.2d) and purity check (Extended Fig. 2e-f) for the sorted cells. This has also been added in the manuscript (Extended Data Fig.2).

b. It is unclear from the methods part how the published RNAseq dataset was integrated into the data:

Thanks for allowing clarification, we extracted the Fastq format of the raw RNAseq data (GSE144708) from the published paper (Ydens et al., 2020). And this dataset was integrated to our data analysis by processing together with the Fastq data of our own MSMP, HMP and lung macrophage datasets, with the same pipeline.

c. include the line number from Jackson since there can be several lines with a similar locus

Thanks for the comment. We have included the detailed information about the transgenic mouse line we employed.

e. F4/80 is not always expressed highly on all macrophages. Iba1 should be used to quantify macrophages in different regions.

As requested, we have added the quantification of Iba1⁺ macrophages in both MS and non-MS areas (Extended Data Fig.1a and 1b). Similar to F4/80 immunostaining results, there are significant more Iba1⁺ macrophages in MS areas compared with non-MS areas (only rarely found), suggesting the enrichment of Iba1⁺ macrophages in MS.

f. Fig. 2: what are the scale bars in IF pictures?

We have added the scale bars for ALL IF pictures, apologies for this oversight.

Referee #3 (Remarks to the Author):

We would like to thank this reviewer for their excellent constructive comments.

This is a provocative study implicating macrophages within muscle spindles as mediators of the classic myotatic stretch reflect. Some interesting findings are presented, but in the end the authors' model is unclear.

Are MSMP mechanically sensitive, and does stretch of MSMP lead to Glu release? Are MSMP attached to the muscle spindles, and do they stretch when the spindles stretch? Some key experiments to test these or related ideas would be welcome.

Thanks for this excellent point!

In the present manuscript we focused on the novel role of MSMP in releasing glutamate to modulate neuronal activity and muscle contraction. The additional functional muscle specific experiments in this revision further strengthen the model. Our data suggests that MSMP couple increasing metabolic demands associated with muscle contraction with neuronal activity by converting glutamine to glutamate that is used to promote the activity of the MS sensory afferents ultimately supporting muscle contraction.

In line with the reviewer's thinking, in addition to chemical and biological stimuli, there is evidence suggesting that macrophages can sense physical cues such as matrix architecture, tissue stiffness and mechanical stimulation ((Previtera and Sengupta, 2015, Hsieh et al., 2017, Jain et al., 2019, Atcha et al., 2021). As the diversity of macrophage functions is built up by their capability to dynamically respond to environmental cues, MSMP localising in MS could have the ability to sense the mechanical signals coming from the muscle stretch. Based on both bulk and single-cell RNA sequencing data, MSMP express Piezo1, a piezo-type mechanosensitive ion channel (mechanoreceptor). Piezo1 can detect membrane tension and transduce the mechanical signal into intracellular chemical and electrical signals by allowing Ca²⁺ influx (Liu et al., 2022). This suggests that MSMP could potentially be mechanically sensitive, which links mechanical stimuli to the activation of MSMP, and possibly to the release of glutamate from MSMP.

Here, we have performed further immunostaining experiments and 3D rendering that clearly showed that MSMP are in close proximity to MS and their shape can vary between been very oblong to been less stretched, suggesting that they might modify their shape in response to muscle stretch (Figure 2, Extended Movie 1-3).

Therefore, we are also very interested to investigate the hypothesis suggested by this reviewer that MSMP might be sensing muscle stretch in future work. In fact, providing hard functional evidence in support of the hypothesis would require extensive and highly complex in vivo experimentation.

Interestingly, GO analysis of gene expression experiments revealed that MSMP not enriched for transcripts implicated in immune responses, as compared to lung and heart macrophages. On the other hand, MSMP express genes associated with muscle contraction, neuronal projection and synaptic activity, and glutamate transport, synthesis, and release. This begs the question, are MSMP macrophages? Are they phagocytic?

The RNA seq results of sorted MSMP showing the significant expression of neuron-related genes (Figure 2). There was no neuronal component being sequenced in this sorted MSMP RNAseq dataset as we used a set of macrophage markers to sort out pure CD45/CD11b/F4/80/CX3CR1⁺ cells without neuronal/muscle cell contamination.

We also need to clarify that MSMP do express immune modulatory genes, although at lower levels compared with other types of macrophages. All data we have generated suggest that

MSMP play a novel role in regulating MS function and activity by releasing the neurotransmitter glutamate. However, we do not argue that they do not also have an immune modulatory role similarly to all other types of macrophages. The comparison between MSMP, HMP, LMP and SNMP also show the expression of genes related to inflammatory responses and phagocytosis-related gene (Extended Data Fig.3). We hypothesise that in addition to the new physiological role we identified and characterised here (release of glutamate in this instance), MSMP might also participate in immune modulation and possibly in a phagocytic/defensive role in response to MS abnormalities, tissue damage or other pro-inflammatory pathological conditions.

MSMP express all canonical macrophage marker genes including Iba1, CD11b, CD68 and F4/80, as well as immune cell marker CD45, suggesting MSMP are macrophages.

In the MS there are no neuronal nuclei, or cell bodies but rather only sensory fibers. The neuronal nuclei and cell bodies of sensory neurons are located in the dorsal root ganglia, far away from the muscle spindles. Furthermore, no other cell of neuronal origin to our knowledge is located in the muscle spindles.

Lastly, in our study muscle spindles appear intact and healthy and there is no suggestion that MSMP in proximity to MS have a direct phagocytic activity on the axons as can be seen by additional 3D reconstructions that we have now provided (Extended Movie 1-3). Furthermore, we searched for the expression of axonal mRNA in MSMP, and we could not find any expression above background. This includes the highly abundant axonal mRNAs Nefl, Gap43, Map2, and Uchl1 (which is actually PGP9.5, which we used to identify the sensory afferents).

The optogenetic experiments are critical for the authors' conclusions. An important observation is that MSMP optogenetic stimulation elicits sensory neuron excitation within milliseconds and with low jitter, and subsequent muscle contraction. Some controls are needed here, including non-transgenic controls. It seems critical that the authors report the expression patterns of the opsins in the CX3CR1-ChR2 and CX3CR1^{Cre}-NpHR3 mice in muscles, to provide confidence that these actuators are restricted in their expression and in the expected cell types (MSMP and not other cells).

We thank the reviewer for these valid comments and suggestions.

Aiming to specifically modulate MSMP activity via optogenetic stimulation, we crossed CX3CR1^{Cre/+} mice with ChR2^{YFP+/-} mice to generate CX3CR1^{Cre+/-} ChR2^{YFP+/-} (ChR2 positive mice) and their negative littermate (CX3CR1^{Cre+/-}::ChR2^{YFP-/-}). Similarly, the same cross has been set for the NpHR3 experiments to obtain CX3CR1^{Cre+/-}::NpHR3^{YFP+/-} (NpHR3 positive) and their negative controls (CX3CR1^{Cre+/-}::NpHR3^{YFP-/-}). All these groups of animals have been used for the in vivo experiments in Figure 3, Extended Data Fig.5., and the new patch-clamp experiments in Extended Data Fig.4. ChR2 positive mice (CX3CR1^{Cre+/-}::ChR2^{YFP+/-}) have been also used for all loss of function experiments with a viral approach (Figure 4 and 6, Extended Data Fig.9 and 10).

Specifically:

1. Validation of expression:

We have confirmed the specific expression of ChR2 and NpHR3 in MSMP using CX3CR1^{Cre}::ChR2/NpHR3^{YFP} mice. Immunostaining of muscle sections showed ChR2 and NpHR3-YFP specific expression in MSMP (macrophages labelled by F4/80 and MS labelled by PGP9.5), validating our Cre-driven expression (see revised Figure 3).

2. Functional Validation of Optogenetic Modulation via Patch-Clamp:

To directly prove that CX3CR1-ChR2 MSMP are depolarised and that CX3CR1- NpHR3 MSMP are hyperpolarised after light stimulation, we utilized patch-clamp, which is the gold standard for measuring ion channel activity and membrane potential, especially in response to optical manipulation. Patch-clamp analysis of CD45/CD11b/F4/80/CX3CR1⁺ sorted from CX3CR1^{Cre::}ChR2^{YFP} and CX3CR1^{Cre::}NpHR3^{YFP} skeletal muscles demonstrated the light stimulation induced depolarisation and hyperpolarisation in ChR2 and NpHR3 models respectively (see Extended Data Fig.4).

More specifically, light stimulation (480 nanometres) on cultured cells induced the expected depolarisation in CX3CR1^{Cre::}ChR2^{YFP} cells (see Extended Data Fig.4a-e). The amount of depolarisation was dependent on the stimulus features such as pulse duration and light power, in accordance with the study of Yi et al., 'Optogenetic activation of spinal microglia triggers chronic pain in mice (Yi et al., 2021). We specifically tested the short duration pulse with low light power (5 ms, 15% power) that we have typically used for the in vivo experiments. This stimulation was able to induce sustained depolarisation of the cells.

Similarly, light stimulation (593 nanometres) on cultured MSMP from CX3CR1^{Cre::}NpHR3^{YFP} induced a hyperpolarisation that was dependent on the stimulus parameters as pulse duration and light power (Extended Data Fig.4f-j). As expected, halorhodopsin stimulation required higher light power. We specifically tested stimulus features that we have used for in vivo experiments (high power light and prolonged duration). YFP positive cells responded to this stimulation with sustained hyperpolarisation independent from the length of the stimulation (Extended Data Fig.4h-j). We have now demonstrated that the light-induced activation of halorhodopsin in CX3CR1⁺ cells lead to membrane hyperpolarisation.

3. Littermate controls

We have included littermate controls lacking ChR2 or NpHR3 in all experiments to ensure the specificity of our findings and rule out non-specific effects of genetic background or experimental handling (see revised Figure 4, 6, Extended Data Fig.9 and 10).

Why do Glutamate receptor antagonists completely block the evoked response while NpHR3 activation/hyperpolarisation leads to a modest, at most 35% inhibition? Is this because NpHR3 is ineffective in silencing MSMP? More importantly, electrophysiological measurements of proprioceptor responses and excitation of muscle cells as well as the stretch reflex should all be assessed in the macrophage depleted mice, since the depletion appears complete (Extended Data Fig.5), to complement the behavioral measurements. This would be the most robust test of necessity, since the CX3CR1^{Cre}-NpHR3 mediated hyperpolarisation experiments may never provide complete inhibition (Figure 3).

Thanks for allowing a clarification here. Glutamate receptor antagonism was performed with the optogenetic activation of MSMP eliciting muscle action potential that were very significantly impaired by glutamate receptor antagonism, suggesting that the effect of MSMP upon muscle contraction require glutamate signalling.

However, NpHR3-mediated MSMP inhibition was performed in a very different and not directly comparable experimental paradigm due to the different question. Here we wanted to test whether the activity of MSMP was required to allow optimal muscle evoked responses at progressively increasing muscle elongations that are expected to require higher neuron activity and thus (our hypothesis), more support from MSMP. Our model in fact suggests that MSMP support neural function when metabolic demands are greater such as with increased muscle elongation/stretch. Specifically, we optogenetically inhibited MSMP in NpHR3-YFP positive mice (CX3CR1^{Cre+/-::} NpHR3^{YFP+/-}) and their negative littermate (CX3CR1^{Cre+/-::} NpHR3^{YFP-/-}) during progressive muscle elongation when EMG responses were recorded. Results showed that the reduction in the amplitude of the muscle action potential in CX3CR1- NpHR3

mice was commensurate to the degree of muscle elongation, reaching about 35% reduction in muscle activity compared to control condition with the greatest stretch (4 mm) (Figure 3i-m). This supports the idea that the activity of MSMP is required for optimal muscle contraction in response to increasing degrees of muscle elongation. The fact that a 35% reduction was obtained as opposed to complete obliteration of the response is consistent with the discovery that MSMP are an additional component of the sensorimotor circuitry to optimise performance (this is also evident by additional experiments with MSMP specific loss of function now in Figure 6, Extended Data Fig. 9 and 10); thus their absence is not expected to result in the complete absence of the stretch reflexes. Although it cannot be fully ruled out, less likely is the possibility of ineffective silencing given the results of the new patch clamp experiments proving the efficiency of halorhodopsin-dependent membrane hyperpolarisation (Extended Data Fig.4).

However, we have now added a new set of experiments that allow to directly compare the MSMP gain function (ChR2) with a MSMP loss of function approach (TeLC) in the same experimental setting.

To this end, we employed a cre-dependent strategy for loss of function experiments where we perturbed MSMP activity in the gastrocnemius muscle (Figure 4 and 6).

Specifically, we leveraged the knowledge that TeLC degrades SNARE proteins VAMP1-3 in neurons and other cell types (Sweeney et al., 1995, Pitzurra et al., 1996, Mendez et al., 2011, Bouvier et al., 2015, Hoogstraaten et al., 2020, Calvigioni et al., 2023) to block synaptic vesicle release from MSMP. Importantly, VAMPs are expressed in MSMP (Figure 2O) and previous evidence showed that macrophages express VAMP-related proteins that are sensitive to TeTx proteolytic cleavage (Pitzurra et al., 1996).

Therefore, we used viral delivery of TeLC in the gastrocnemius muscles of CX3CR1^{Cre::}ChR2^{YFP} mice to block vesicle release specifically in MSMP and performed EMG recordings. AAV-mCherry was used in controls. We have perturbed MSMP in the gastrocnemius because of the important function of this muscle in generating the power stroke in swimming and walking on a ladder for mice and because the stretch reflex circuits of this muscle are highly engaged during these tasks.

Optical stimulation of MSMP in controls led to the expected muscle evoked potential selectively in the stimulated muscle (Figure 4f-g, Extended Data Fig.5e). In contrast, the muscle response was selectively abolished in the TeLC infected muscle (Figure 4f-g), demonstrating that synaptic vesicle release is needed for MSMP-mediated muscle contraction. This allowed to demonstrate that MSMP depolarisation resulting from ChR2 stimulation need the integrity of the vesicle release machinery to drive muscle activity.

Additionally, in a separate set of experiments we injected AAV-DIO-caspase 3 to selectively deplete MSMP (a similar approach had been used for microglia depletion: (Jia et al., 2023)). Still, AAV-mCherry was used in controls mice (CX3CR1^{Cre::}ChR2^{YFP} mice).

Both approaches provided consistent results, demonstrating significant and highly muscle specific alterations in spatiotemporal gait parameters and angular excursion variability when MSMP were disrupted (Figure 4 and 6 and Extended Data Fig.9 and 10).

Expression of TeLC in MSMP and MSMP depletion (about 75%) were confirmed by immunostaining in the gastrocnemius muscle. The results of these experiments are summarised in Figure 4 and 6 and Extended Data Fig.9 and 10.

By using these advanced genetic models, we have strengthened the evidence for the critical role of MSMP in locomotor behaviour, thereby enhancing the robustness and validity of our findings.

In conclusion, both the TeLC and Caspase 3 experiments show the selective role of MSMP in locomotor behaviour.

Reference list:

- ATCHA, H., JAIRAMAN, A., HOLT, J. R., MELI, V. S., NAGALLA, R. R., VEERASUBRAMANIAN, P. K., BRUMM, K. T., LIM, H. E., OTHY, S., CAHALAN, M. D., PATHAK, M. M. & LIU, W. F. 2021. Mechanically activated ion channel Piezo1 modulates macrophage polarization and stiffness sensing. *Nat Commun*, 12, 3256.
- BOUVIER, J., CAGGIANO, V., LEIRAS, R., CALDEIRA, V., BELLARDITA, C., BALUEVA, K., FUCHS, A. & KIEHN, O. 2015. Descending Command Neurons in the Brainstem that Halt Locomotion. *Cell*, 163, 1191-1203.
- CALVIGIONI, D., FUZIK, J., LE MERRE, P., SLASHCHEVA, M., JUNG, F., ORTIZ, C., LENTINI, A., CSILLAG, V., GRAZIANO, M., NIKOLAKOPOULOU, I., WEGELAGE, M., LAZARIDIS, I., KIM, H., LENZI, I., PARK, H., REINIUS, B., CARLEN, M. & MELETIS, K. 2023. Esr1(+) hypothalamic-habenula neurons shape aversive states. *Nat Neurosci*, 26, 1245-1255.
- HOOGSTRAATEN, R. I., VAN KEIMPEMA, L., TOONEN, R. F. & VERHAGE, M. 2020. Tetanus insensitive VAMP2 differentially restores synaptic and dense core vesicle fusion in tetanus neurotoxin treated neurons. *Sci Rep*, 10, 10913.
- HSIEH, J. Y., SMITH, T. D., MELI, V. S., TRAN, T. N., BOTVINICK, E. L. & LIU, W. F. 2017. Differential regulation of macrophage inflammatory activation by fibrin and fibrinogen. *Acta biomaterialia*, 47, 14-24.
- JAIN, N., MOELLER, J. & VOGEL, V. 2019. Mechanobiology of Macrophages: How Physical Factors Coregulate Macrophage Plasticity and Phagocytosis. *Annu Rev Biomed Eng*, 21, 267-297.
- JIA, J., ZHENG, L., YE, L., CHEN, J., SHU, S., XU, S., BAO, X., XIA, S., LIU, R., XU, Y. & ZHANG, M. 2023. CD11c(+) microglia promote white matter repair after ischemic stroke. *Cell Death Dis*, 14, 156.
- LIU, H., HU, J., ZHENG, Q., FENG, X., ZHAN, F., WANG, X., XU, G. & HUA, F. 2022. Piezo1 channels as force sensors in mechanical force-related chronic inflammation. *Frontiers in Immunology*, 13, 816149.
- MENDEZ, M., GROSS, K. W., GLENN, S. T., GARVIN, J. L. & CARRETERO, O. A. 2011. Vesicle-associated membrane protein-2 (VAMP2) mediates cAMP-stimulated renin release in mouse juxtaglomerular cells. *J Biol Chem*, 286, 28608-18.
- PITZURRA, L., ROSSETTO, O., CHIMIENTI, A. R., BLASI, E. & BISTONI, F. 1996. Tetanus toxin-sensitive VAMP-related proteins are present in murine macrophages. *Cell Immunol*, 169, 113-6.
- PREVITERA, M. L. & SENGUPTA, A. 2015. Substrate stiffness regulates proinflammatory mediator production through TLR4 activity in macrophages. *PloS one*, 10, e0145813.
- SWEENEY, S. T., BROADIE, K., KEANE, J., NIEMANN, H. & O'KANE, C. J. 1995. Targeted expression of tetanus toxin light chain in Drosophila specifically eliminates synaptic transmission and causes behavioral defects. *Neuron*, 14, 341-51.
- YDENS, E., AMANN, L., ASSELBERGH, B., SCOTT, C. L., MARTENS, L., SICHEN, D., MOSSAD, O., BLANK, T., DE PRIJCK, S., LOW, D., MASUDA, T., SAEYS, Y., TIMMERMAN, V., STUMM, R., GINHOUX, F., PRINZ, M., JANSSENS, S. & GUILLIAMS, M. 2020. Profiling peripheral nerve macrophages reveals two macrophage subsets with distinct localization, transcriptome and response to injury. *Nat Neurosci*, 23, 676-689.
- YI, M. H., LIU, Y. U., UMPIERRE, A. D., CHEN, T. J., YING, Y. L., ZHENG, J. Y., DHEER, A., BOSCO, D. B., DONG, H. L. & WU, L. J. 2021. Optogenetic activation of spinal microglia triggers chronic pain in mice. *Plos Biology*, 19.

Referee #1 (Remarks to the Author):

The authors report novel and important findings using innovative methodological approaches. In the revised manuscripts they have addressed most of our comments.

They improve the imaging of MSMP, and they now show specific expression of the channel rhodopsin in MSMP, via immunofluorescence. They also show that depolarization and hyperpolarization happens in the macrophages using a patch clamp technic on sorted macrophages in vitro with their optogenetic model.

They came up with 2 additional macrophages depletion strategies, using cre dependent TeLC delivery on the gastrocnemius muscle of Cx3cr1Cre-Chr2 mice to block vesicles release from macrophages. They also use cre dependent AAV-DIO-Caspase 3 injection in the muscle to deplete macrophages, which seems to work for depletion and results in impairment of locomotor behavior.

Overall, however, the mechanism by which MSMP promote muscle contraction is still not clear. The authors suggest that neuronal genes expressed by macrophage may play a role, they propose a key role for glutamine but this is not demonstrated in vivo and macrophages express many genes, including cytokines, which could also influence nerves stimulation

On balance, this is a very interesting paper describing a novel type of macrophage function, and which would deserve to be published, with the caveat that it lacks a clear biological mechanism.

We would like to thank this reviewer for appreciating our manuscript, our improvement with the revision that addressed the extremely useful comments, and for encouraging publication.

The reviewer also mentioned that MSMP may express other genes that affect nerve stimulation including cytokines. While we are agree with this assumption (as been shown from multiple studies), we also believe the short time frame in which MSMP modulate neural activity in our study (5 to 10 milliseconds) and the fact that we abolish neural responses upon MSMP stimulation by using neurotransmitter blockers indicate that neural responses, muscle contraction and behavioural outcome are regulated by a fast, glutamatergic transmission between MSMP and sensory afferents.

We fully agree that our proposed model showing that macrophages can release glutamate via glutaminase-dependent conversion of glutamine is supported by experiments in cultured MSMP. In fact, in response to reviewer 2 comments after the first revision we have added a statement highlighting this limitation.

To address the final reviewer's comment at least partially, we decided to provide additional evidence of the physiological in vivo relevance of the mechanism by investigating whether swimming enhanced glutaminase expression in MSMP. Briefly, as we showed in figure 5, glutaminase and other early response genes (as markers of MSMP activation) were upregulated after optogenetic PV neuronal stimulation of the stretch reflex. We have now measured them again from MSMP sorted after a physiologically relevant task such as swimming vs resting mice. Swimming induced a significant upregulation of glutaminase (*Gls*), *Egr1*, *JunB*, *Atf4*, and *Ier5*, adding evidence that MSMP respond to a physiologically relevant task such as swimming. These data have been added to the manuscript as Extended Data Figure 8.

Extended Data Figure 8. Muscle spindle macrophages (MSMP) respond to swimming. RT-qPCR results showing increased expression of glutaminase (*Gls*) and early response genes (*Egr1*, *JunB*, *Atf4* and *Ier5*) in sorted MSMP after 30-minute swimming vs resting physiological conditions (N = 3 biological replicates; unpaired t-test; mean \pm s.e.m; *p < 0.05, **p < 0.01).

Please see also an additional video (**Extended Movie 4**) showing how mice whose MSMP were transduced with AAV-TeLC in the gastrocnemius muscle display impairment in angular excursions during swimming.

This reviewer is also kindly invited to read the response to reviewer 3 where we performed some additional experiments in cultured MSMP to link glutamine uptake to glutamate release.

Finally, we have also added a specific limitation statement to the manuscript as follows: “This manuscript describes the discovery of MSMP and their functional characterisation in contributing to the stretch reflex, muscle contraction and locomotion by a fast glutamatergic transmission between MSMP and sensory afferents. However, whether MSMP convert glutaminase into glutamate in vivo and whether they respond to mechanical stretch to support this or other mechanisms in vivo remains to be determined”.

Referee #2 (Remarks to the Author):

The revised manuscript has addressed the most important points. The models chosen to deplete macrophages locally are elegant and make a strong point for MSMPs controlling locomotion. I would still insist on exploiting the scRNA-seq data so that it becomes clear to the reader what can be actually observed between activated and control MSMPs besides the one violin plot shown for the glutaminase. The UMAPs showing expression of genes in almost 100% of cells doesn't help. This should be moved into supplementary, also the scheme explaining what was done is somewhat self-explanatory and can be supplementary, as it should be found in the text and methods part. A dot plot/violin plots showing what is defining the identity of the macrophage clusters 0-3 would be a first step (it is clear that cluster 4 are proliferating macrophages). The conditions should be split, as shown in Figure 5f for this representation. The authors write that the cells have similar identities, but still, something is determining their clustering and should be shown. For the macrophage community it would be of interest to define/validate these subpopulations also on a protein level. Do they have distinct subanatomical locations? Cluster 3 seems to not upregulate glutaminase, so it may not be involved in the described process. This needs to be addressed to some degree.

We would like to thank this reviewer for the positive assessment of our revised manuscript, for the previous suggestions that have helped crafting an improved story and for the further specific question that we have now addressed.

We have reworked the scRNA seq data and figures as requested and moved the panels in Figure 5 to supplementary and replaced them with new ones. As suggested, we have also performed protein validation of the clusters by using representative markers with well working antibodies by immunofluorescence. This has also allowed us to create violin plots to show spatial localisation within the MS capsule by using the sensory annular fibres as reference (see revised Figure 5 and Extended Data Figure 6).

Revised Figure 5. Muscle spindle macrophages (MSMP) response to proprioceptive neuronal activation and convert glutamate into glutamine via glutaminase.

a. Uniform Manifold Approximation and Projection (UMAP) results show the expression of marker gene for 5 clusters of 18,538 FACS-sorted MSMP. **b.** DE analysis from identified macrophage clusters shows increased expression of genes involved in cytokine/chemokine signalling, macrophage/neuron activation, transcription, differentiation, and macrophage shape remodel (MSR) (FDR < 0.05, fold change > 1). **c.** DE analysis shows significantly increased expression of glutaminase in response to Pv neuron activation in MSMP cluster 0, 1 and 2 (FDR < 0.05, fold change > 1; *p < 0.05, **p < 0.01, ****p < 0.0001). **d.** Experimental design of *in vitro* MSMP culture medium glutamate assay with FACS-sorted MSMP treated with vehicle or glutamine or glutaminase inhibitor (CB-839) or glutamine transporter inhibitors (MeAIB and V9302) or Ca²⁺ chelators (BAPTA-AM and EGTA). **e.** Glutamate was detected in MSMP culture medium 2/8 hours after 50 mM glutamine treatment. The release of glutamate was diminished by glutaminase inhibitor (CB-839) (N=6 biological replicates; One-way ANOVA; mean ± s.e.m; ****p < 0.0001). **f.** Expression of glutamine transporters SNAT1 (*Slc38a1*), SNAT2 (*Slc38a2*) and ASCT2 (*Slc1a5*) by MSMP. **g-h.** Immunostaining and quantification results showing elevated MSMP intracellular Ca²⁺ level (Fluo-4-AM signal) induced by 50 mM glutamine treatment and was significantly reduced by glutamine transporter inhibitors (MeAIB+V9302) and Ca²⁺ chelators (BAPTA-AM and EGTA) (N = 3 biological replicates; One-way ANOVA; mean ± s.e.m; **p < 0.01, ****p < 0.0001, Scale bar = 50 µm). **i.** Glutamate was detected in MSMP culture medium after 2-hour 50 mM glutamine treatment. The medium glutamate level was significantly reduced by glutamine transporter inhibitors (MeAIB+V9302) and Ca²⁺ chelators (BAPTA-AM and EGTA) (N = 3 biological replicates; One-way ANOVA; mean ± s.e.m; ***p < 0.001, ****p < 0.0001).

Extended Data Figure 6. MSMP clusters express tissue resident markers and share similar sub-anatomical localisation within MS.

a. Uniform Manifold Approximation and Projection (UMAP) results show 5 clusters of MSMP from 18,538 F4/80/CX3CR1⁺ MSMP. **b.** Expression of MSMP markers and tissue-resident markers in MSMP. **c-d.** Violin plot (c) and representative immunostaining (d) for the expression of marker genes (*Il1rn*, *Hspa1a* and *Pecam1*) of cluster 0-2 MSMP (labelled by F4/80) in MS (labelled by PGP9.5). Scale bar = 50 μ m. **e.** Spatial localisation (distance between MSMP and MS sensory fiber) of cluster 0-2 MSMP by using the sensory annular fibres as reference. Cluster 0, cluster 1 and cluster 2 MSMP are identified by the expression of *Il1rn* (n = 65 MSMP), *Hspa1a* (n = 69 MSMP) and *Pecam1* (n = 69 MSMP), respectively (N = 3 biological independent mice; One-way ANOVA; ns = not significant). **f.** Expression of *Glis* (Glutaminase) and *Slc17a5* (Sialin) by MSMP.

Referee #3 (Remarks to the Author):

This is a re-review of a manuscript that reports a series of experiments that address the role of resident macrophages in muscle and their role in muscle stretch evoked proprioceptor signaling and function. The revision includes many new experiments, which have improved the study.

I had previously questioned the overall model, and in particular whether MSMP are mechanically sensitive and whether stretch of MSMP leads to Glu release. This issue has not been fully addressed, and how macrophages respond to stimuli and contribute to proprioceptor signaling remains unclear.

- a.
- b. We would like to thank this reviewer's appreciation of the substantial work we have carried out in response to the first round of reviews.
- c. While we agree with the reviewer that we have not investigated "whether MSMP are mechanically sensitive and whether stretch of MSMP leads to Glu release", we believe that we have clearly showed the MSMP promote neural modulation of the sensory afferents and muscle contraction through a fast glutamatergic signalling. Additionally, we have provided robust in vivo experimental evidence of the physiological relevance of MSMP by performing MSMP inhibition, silencing and depletion in selected muscles in the context of the stretch reflex and physiological locomotor tasks (skilled overground locomotion and swimming).
- d. To recapitulate:

- 1) **The spatio-temporal features of MSMP makes them an appropriate candidate to modulate the stretch reflex in a behaviourally relevant time scale in physiological conditions.** The particular location of the MSMP in the muscle spindle, juxtaposed to the Ia sensory fibers, and the possibility to modulate the sensory fibers in a ms time scale provide initial support for the possibility of the MSMP to be an active component of the stretch reflex (that works also in a ms time scale) supporting muscle contraction
- 2) **The reduction of the stretch reflex while inhibiting the MSMP indicate a direct and temporally fast action of the MSMP on the sensorimotor component of the reflex arc.** The experiments with halorhodopsin (NpHR3) silencing of MSMP activity after we evoked the stretch reflex of the gastrocnemius showed a reduction of the stretch response. The stretch reflex is a natural response that requires about 15 ms to generate a response with phasic and tonic components in the gastrocnemius muscle of a mouse in physiological conditions. The temporal features of the response (milliseconds) and the temporal feature of the optical stimulation (milliseconds) exclude any other communication between MSMP, sensory and motor components if not electrochemical.
- 3) **Independently from any optogenetic manipulation, the impaired locomotion in mice with silenced or depleted MSMP indicate the physiological relevance of MSMP in natural motor tasks.** The MSMP silencing experiments with TeLC and depletion with caspase 3 in a specific muscle such as the gastrocnemius, showed how MSMP are required for the optimal execution of ethologically relevant tasks as swimming and walking (we have now found that glutaminase expression is induced after swimming, Figure 6, Extended Data Figure 8, 11-12). Please see an additional video (**Extended Movie 4** showing how mice whose MSMP were transduced with TeLC in the gastrocnemius muscle display impairment in angular excursions during swimming).

To further prove the physiological relevance, we decided to provide additional evidence of the physiological in vivo relevance of the mechanism by investigating whether swimming enhanced glutaminase expression in MSMP. Briefly, as we showed in figure 5, glutaminase (*Gls*) and other early response genes (as markers of MSMP activation) were upregulated after optogenetic PV neuronal stimulation of the stretch reflex. We have now measured them again from MSMP sorted after a physiologically relevant task such as swimming vs resting mice. Swimming induced a significant upregulation of glutaminase (*Gls*), *Egr1*, *JunB*, *Atf4*, and *Ier5*, adding evidence that MSMP respond to a physiologically relevant task such as swimming. These data have been added to the manuscript as Extended Data Figure 8.

Extended Data Figure 8. Muscle spindle macrophages (MSMP) respond to swimming. RT-qPCR results showing increased expression of glutaminase (*Gls*) and early response genes (*Egr1*, *JunB*, *Atf4* and *Ier5*) in sorted

MSMP after 30-minute swimming vs control physiological conditions (N = 3 biological replicates; unpaired t-test; mean \pm s.e.m; *p < 0.05, **p < 0.01).

[TEXT REDACTED]

[FIGURE REDACTED]

In alternative to this model, and by using the same experimental paradigm, we further explored another mechanism that may mediate glutamate release to support our main hypothesis: MSMP might release glutamate directly in response to glutamine.

Previous studies have shown that glutamine transporters may induce membrane depolarisation, calcium mobilisation and glutamate release (Sherman, 1991, Chaudhry et al., 2002, Bacci et al., 2002, Billups et al., 2013, Kolbaev and Draguhn, 2008). Our RNA sequencing data have shown the expression of glutamine transporters in MSMP (Fig. 2o, Fig. 5f). Therefore, we investigated whether glutamine itself can induce MSMP membrane depolarisation which leads to glutamate release from cultured MSMP.

As in the previous experiment and in previous studies (Gross et al., 2014, Yoo et al., 2020, Zhang et al., 2020, Schulte et al., 2018, Raposo et al., 2015), we cultured MSMP adding the calcium Fluo-4 sensor to use at proxy of MSMP activation. To the basal medium, we added glutamine that resulted in an strong increase in Fluo-4 signal (Figure 5g and h) and a concurrent increase in glutamate concentration in the medium (vehicle + glutamine, figure 5i).

When glutamine transporters inhibitors MeAIB and V9302 were used (blocking the transporters SNAT1, SNAT2 and ASCT2 encoded in MSMP by *Slc38a1*, *Slc38a2* and *Slc1a5*, these resulted in both a marked reduction in MSMP activation (Figure 5h) and glutamate concentration (Figure 5i).

Lastly, the addition of calcium chelators BAPTA-AM and EGTA to the cultured MSMP completely abolished MSMP activity (Figure 5h) and fully inhibited glutamate release (Figure 5i).

These results clearly demonstrate that glutamine induces MSMP activity and glutamate release via glutamine transporters. Furthermore, inhibiting these transporters reduces both MSMP activation and glutamate concentration, and calcium chelation completely blocks these processes. Together, this supports the idea that glutamine transporters play a key role in glutamate release from MSMP, driven by calcium-dependent mechanisms.

The new Figure 5 has been pasted here below.

Revised Figure 5. Muscle spindle macrophages (MSMP) response to proprioceptive neuronal activation and convert glutamine into glutamate via glutaminase.

Uniform Manifold Approximation and Projection (UMAP) results show the expression of marker gene for 5 clusters of 18,538 FACS-sorted MSMP. **b**. DE analysis from identified macrophage clusters shows increased expression of genes involved in cytokine/chemokine signalling, macrophage/neuron activation, transcription, differentiation, and macrophage shape remodel (MSR) (FDR < 0.05, fold change > 1). **c**. DE analysis shows significantly increased expression of glutaminase in response to Pv neuron activation in MSMP cluster 0, 1 and 2 (FDR < 0.05, fold change > 1; *p < 0.05, **p < 0.01, ****p < 0.0001). **d**. Experimental design of *in vitro* MSMP culture medium glutamate assay with FACS-sorted MSMP treated with vehicle or glutamine or glutaminase inhibitor (CB-839) or glutamine transporter inhibitors (MeAIB and V9302) or Ca²⁺ chelators (BAPTA-AM and EGTA). **e**. Glutamate was detected in MSMP culture medium 2/8 hours after 50 mM glutamine treatment. The release of glutamate was diminished by glutaminase inhibitor (CB-839) (N=6 biological replicates; One-way ANOVA; mean ± s.e.m; ****p < 0.0001). **f**. Expression of glutamine transporters SNAT1 (*Slc38a1*), SNAT2 (*Slc38a2*) and ASCT2 (*Slc1a5*) by MSMP. **g-h**. Immunostaining and quantification results showing elevated MSMP intracellular Ca²⁺ level (Fluo-4-AM signal) induced by 50 mM glutamine treatment and was significantly reduced by glutamine transporter inhibitors (MeAIB+V9302) and Ca²⁺ chelators (BAPTA-AM and EGTA) (N = 3 biological replicates; One-way ANOVA; mean ± s.e.m; ****p < 0.0001). **i**. Glutamate concentration was significantly reduced by glutamine transporter inhibitors (MeAIB+V9302) and Ca²⁺ chelators (BAPTA-AM and EGTA) (N = 3 biological replicates; One-way ANOVA; mean ± s.e.m; ****p < 0.0001).

± s.e.m; **p < 0.01, ****p < 0.0001, Scale bar = 50 µm). i. Glutamate was detected in MSMP culture medium after 2-hour 50 mM glutamine treatment. The medium glutamate level was significantly reduced by glutamine transporter inhibitors (MeAIB+V9302) and Ca²⁺ chelators (BAPTA-AM and EGTA) (N = 3 biological replicates; One-way ANOVA; mean ± s.e.m; ***p < 0.001, ****p < 0.0001).

These additional data and the overall experiments suggest a model where sensory afferents communicate with MSMP indirectly, while MSMP directly communicate with Ia fibers: Ia fibers sense changes in intrafusal muscle tension to elicit, via motoneuron activation, muscle contraction, which increases muscle metabolism that leads to the release of glutamine that is uptaken by MSMP to convert it via glutaminase in glutamate, which in turn activates the Ia afferents.

This provides a previously unknown layer of metabolic control in MASP (conversion of glutamine into glutamate) to facilitate sensory afferent activity to enhance muscle contraction in response to muscle activity on top of the known regulation that relies on the mechanically sensitive stretch receptors that are the muscle spindles.

In summary, our model suggests that MSMP play an active role by converting glutamine into glutamate allowing the type Ia afferents to take advantage of MSMP-dependent “metabolic boost” during muscle activity, ultimately contributing to the degree of Ia activity and muscle contraction by sensing the muscular metabolic demand.

Finally, while we hope that this reviewer will agree that we have discovered an exciting new mechanism that support physiological function, we are aware that radical new findings require caution to avoid overinterpretation and we remain humble and open to the future development of additional roles for MSMP, including in response to mechanical stretch. To reflect the limitations of our study, we have also added a specific limitation statement to the manuscript as follows: “This manuscript describes the discovery of MSMP and their functional characterisation in contributing to the stretch reflex, muscle contraction and locomotion by a fast glutamatergic transmission between MSMP and sensory afferents. However, whether MSMP convert glutaminase into glutamate in vivo and whether they respond to mechanical stretch to support this or other mechanisms in vivo remains to be determined”.

References

- BACCI, A., SANCINI, G., VERDERIO, C., ARMANO, S., PRAVETTONI, E., FESCE, R., FRANCESCHETTI, S. & MATTEOLI, M. 2002. Block of glutamate-glutamine cycle between astrocytes and neurons inhibits epileptiform activity in hippocampus. *J Neurophysiol*, 88, 2302-10.
- BILLUPS, D., MARX, M. C., MELA, I. & BILLUPS, B. 2013. Inducible presynaptic glutamine transport supports glutamatergic transmission at the calyx of Held synapse. *J Neurosci*, 33, 17429-34.
- CAI, G., LU, Y., ZHONG, W., WANG, T., LI, Y., RUAN, X., CHEN, H., SUN, L., GUAN, Z., LI, G., ZHANG, H., SUN, W., CHEN, M., ZHANG, W. B. & WANG, H. 2023. Piezo1-mediated M2 macrophage mechanotransduction enhances bone formation through secretion and activation of transforming growth factor-beta1. *Cell Prolif*, 56, e13440.
- CHAUDHRY, F. A., SCHMITZ, D., REIMER, R. J., LARSSON, P., GRAY, A. T., NICOLL, R., KAVANAUGH, M. & EDWARDS, R. H. 2002. Glutamine uptake by neurons: interaction of protons with system a transporters. *J Neurosci*, 22, 62-72.
- COSTE, B., MATHUR, J., SCHMIDT, M., EARLEY, T. J., RANADE, S., PETRUS, M. J., DUBIN, A. E. & PATAPOUTIAN, A. 2010. Piezo1 and Piezo2 are essential components of distinct mechanically activated cation channels. *Science*, 330, 55-60.
- COSTE, B., XIAO, B., SANTOS, J. S., SYEDA, R., GRANDL, J., SPENCER, K. S., KIM, S. E., SCHMIDT, M., MATHUR, J., DUBIN, A. E., MONTAL, M. & PATAPOUTIAN, A. 2012. Piezo proteins are pore-forming subunits of mechanically activated channels. *Nature*, 483, 176-81.
- GROSS, M. I., DEMO, S. D., DENNISON, J. B., CHEN, L., CHERNOV-ROGAN, T., GOYAL, B., JANES, J. R., LAIDIG, G. J., LEWIS, E. R., LI, J., MACKINNON, A. L., PARLATI, F., RODRIGUEZ, M. L., SHWONEK, P. J., SJOGREN, E. B., STANTON, T. F., WANG, T., YANG, J., ZHAO, F. & BENNETT, M. K. 2014. Antitumor activity of the glutaminase inhibitor CB-839 in triple-negative breast cancer. *Mol Cancer Ther*, 13, 890-901.
- KOLBAEV, S. & DRAGUHN, A. 2008. Glutamine-induced membrane currents in cultured rat hippocampal neurons. *Eur J Neurosci*, 28, 535-45.
- LIN, Y. C., GUO, Y. R., MIYAGI, A., LEVRING, J., MACKINNON, R. & SCHEURING, S. 2019. Force-induced conformational changes in PIEZO1. *Nature*, 573, 230-234.
- RAPOSO, B., VAARTJES, D., AHLQVIST, E., NANDAKUMAR, K. S. & HOLMDAHL, R. 2015. System A amino acid transporters regulate glutamine uptake and attenuate antibody-mediated arthritis. *Immunology*, 146, 607-17.
- SCHULTE, M. L., FU, A., ZHAO, P., LI, J., GENG, L., SMITH, S. T., KONDO, J., COFFEY, R. J., JOHNSON, M. O., RATHMELL, J. C., SHARICK, J. T., SKALA, M. C., SMITH, J. A., BERLIN, J., WASHINGTON, M. K., NICKELS, M. L. & MANNING, H. C. 2018. Pharmacological blockade of ASCT2-dependent glutamine transport leads to antitumor efficacy in preclinical models. *Nat Med*, 24, 194-202.
- SHERMAN, A. D. 1991. Depolarization and synaptosomal glutamine utilization. *Neurochem Res*, 16, 501-4.
- SOLIS, A. G., BIELECKI, P., STEACH, H. R., SHARMA, L., HARMAN, C. C. D., YUN, S., DE ZOETE, M. R., WARNOCK, J. N., TO, S. D. F., YORK, A. G., MACK, M., SCHWARTZ, M. A., DELA CRUZ, C. S.,

- PALM, N. W., JACKSON, R. & FLAVELL, R. A. 2019. Mechanosensation of cyclical force by PIEZO1 is essential for innate immunity. *Nature*, 573, 69-74.
- TU, P. C., PAN, Y. L., LIANG, Z. Q., YANG, G. L., WU, C. J., ZENG, L., WANG, L. N., SUN, J., LIU, M. M., YUAN, Y. F., GUO, Y. & MA, Y. 2022. Mechanical Stretch Promotes Macrophage Polarization and Inflammation via the RhoA-ROCK/NF-kappaB Pathway. *Biomed Res Int*, 2022, 6871269.
- YOO, H. C., YU, Y. C., SUNG, Y. & HAN, J. M. 2020. Glutamine reliance in cell metabolism. *Exp Mol Med*, 52, 1496-1516.
- ZHANG, Z., LIU, R., SHUAI, Y., HUANG, Y., JIN, R., WANG, X. & LUO, J. 2020. ASCT2 (SLC1A5)-dependent glutamine uptake is involved in the progression of head and neck squamous cell carcinoma. *Br J Cancer*, 122, 82-93.